# Reversible conjugation of a CBASS nucleotide cyclase regulates bacterial immune response to phage infection

Larissa Krüger [1] ✉, Laura Gaskell-Mew[1], Shirley Graham[1], Sally Shirran[1], Robert Hertel [2] & Malcolm F. White [1] ✉

Prokaryotic antiviral defence systems are frequently toxic for host cells and stringent regulation is required to ensure survival and fitness. These systems must be readily available in case of infection but tightly controlled to prevent activation of an unnecessary cellular response. Here we investigate how the bacterial cyclic oligonucleotide-based antiphage signalling system (CBASS) uses its intrinsic protein modification system to regulate the nucleotide cyclase. By integrating a type II CBASS system from *Bacillus cereus* into the model organism *Bacillus subtilis*, we show that the protein-conjugating Cap2 (CBASS associated protein 2) enzyme links the cyclase exclusively to the conserved phage shock protein A (PspA) in the absence of phage. The cyclase–PspA conjugation is reversed by the deconjugating isopeptidase Cap3 (CBASS associated protein 3). We propose a model in which the cyclase is held in an inactive state by conjugation to PspA in the absence of phage, with conjugation released upon infection, priming the cyclase for activation.

The evolution of bacteria and their viruses (phages) has been driven by the constant arms race between them. While bacteria developed more potent defence systems, phages evolved to overcome these systems. Many eukaryotic defence systems have their ancestral roots in the prokaryotic world. Thus, the eukaryotic cGAS enzyme of the cGAS-STING pathway is a homologue of the bacterial cyclic nucleotidyl-transferase (CD-NTase; hereafter, 'cyclase'), which is part of the cyclic-oligonucleotide-based antiphage signalling system (CBASS)[1,2]. Both pathways function via the detection of viral infection and activation of a nucleotide cyclase which generates a cyclic nucleotide second messenger that in turn activates an antiviral downstream response. cGAMP is used as the second messenger by the eukaryotic and some bacterial defence systems, but recent studies have revealed a large diversity of bacterial cyclase enzymes that are all related to cGAS and synthesize a diverse array of cyclic nucleotides in response to phage infection[1–4]. These signals in turn activate a wide range of downstream effectors including nucleases,

hydrolases and membrane disruption proteins that modulate the bacterial immune response[5–8].

Key details of CBASS immunity remain unexplored; in particular, the signal that activates cyclases is in many cases still unknown. The eukaryotic homologue cGAS is stimulated by binding to double-stranded DNA or RNA in the cytosol, indicative of viral infection[9–11]. Similarly, it has recently been observed that cyclases from type I CBASS systems in *Staphylococcus schleiferi* bind viral RNA, thereby triggering cyclic dinucleotide synthesis[12]. Phage capsid proteins have been implicated in triggering type II CBASS defence in *Pseudomonas aeruginosa*[13], while the cyclase DncV from *Vibrio cholerae* binds folate-like molecules and alterations in their levels during phage replication activates the enzyme[14–16]. In type III CBASS systems, binding of certain phage peptides by HORMA-domain containing proteins changes their state and allows them to bind and activate the cyclase[17].

Type II CBASS operons contain genes encoding homologues of the ubiquitin ligation machinery[4]. Two studies have reported that the

¹School of Biology, University of St Andrews, St Andrews, UK. ²Genomic and Applied Microbiology, Göttingen Centre for Molecular Biosciences, Georg-August-University Göttingen, Göttingen, Germany. ✉e-mail: lmak1@st-andrews.ac.uk; mfw2@st-andrews.ac.uk

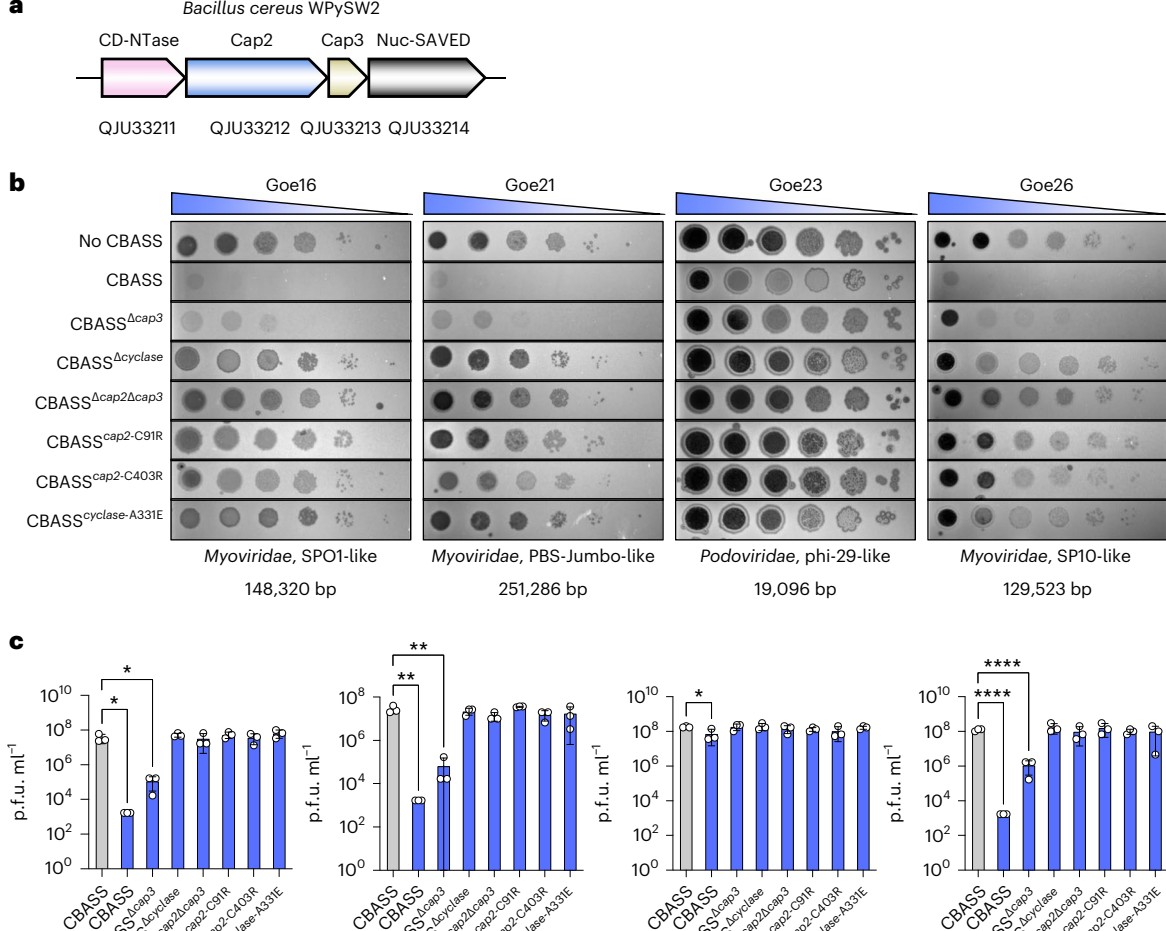

**Fig. 1 | CBASS protects against phage infection. a**, CBASS operon organization in *B. cereus* WPySW2 with GenBank accession numbers. **b**, Variants of the CBASS operon were integrated into the genome of *B. subtilis* Δ6. Strains used: LK06 (CBASS), LK10 (CBASS$^{Δcap2Δcap3}$), LK18 (no CBASS), LK20 (CBASS$^{Δcap3}$), LK64 (CBASS$^{cap2-C91R}$), LK65 (CBASS$^{cap2-C403R}$), LK66 (CBASS$^{cyclase-A331E}$). Exponentially growing cells were infected with dilutions of the respective phage. Representative plates of 3 biological replicates are shown. The morphotype and the genome size of each phage are indicated. **c**, Plaque-forming units (p.f.u. ml$^{-1}$) of infection drop assays in **b**. Data are presented as mean ± s.d. Statistical analysis was performed using one-way analysis of variance (ANOVA), followed by Dunnett's multiple comparisons test ($P$ values listed from left to right on the graph: Goe16 *$P$ = 0.0153, *$P$ = 0.0155; Goe21 **$P$ = 0.0027, **$P$ = 0.0028; Goe23 *$P$ = 0.03; Goe26 ****$P$ < 0.0001, ****$P$ < 0.0001). Detailed information on the genotypes can be found in Supplementary Table 2.

carboxy terminus of the cyclase is activated by ADP ribosylation and transiently linked to Cap2 before conjugation to its target protein[18,19]. The nature of the cellular target, the role of the regulation by conjugation of cyclase for immunity, as well as its link to phage infection remain largely unexplored.

Here we studied the type II CBASS system from *Bacillus cereus* strain WPySW2. The operon consists of four genes encoding a cyclase, Cap2, Cap3 and a Nuc-SAVED effector (Fig. 1a). We transferred the system from *B. cereus* into the cognate host *Bacillus subtilis* to study this CBASS system in vivo. Using an unbiased cell separation approach, we observe that the cyclase is in part membrane associated and, in the presence of the Cap2 enzyme, conjugated to phage shock protein A (PspA), the main effector of the phage shock protein (Psp) response, a universally conserved bacterial response to envelope stress[20].

## Results

### The *B. cereus* CBASS system is functional in *B. subtilis*

The *B. cereus* CBASS operon (Fig. 1a) was introduced into the genome of *B. subtilis* Δ6 and expressed under a strong, constitutive promoter ($P_{degQ-H}$)[21,22]. *B. subtilis* Δ6 was used as it is deficient of the two prophages

(SPβ and PBSX), the three prophage-like elements (prophage 1, prophage 3 and *skin*) and the polyketide synthase operon[23]. The protective effect of CBASS against phage infection was tested by infecting the *B. subtilis* Δ6 parental strain (hereafter referred to as 'no CBASS') and strains containing full or partial CBASS operons with four novel *B. subtilis* phages. We compared the genomes of the four phages to known *B. subtilis* phages and classified them accordingly (Extended Data Fig. 1 and Supplementary Table 1). The phages Goe16 and Goe26 belong to two different genera within the *Herelleviridae* family and the *Spounavirinae* subfamily. Goe21 is a jumbo virus and belongs to the Takahashivirus genus. Goe23 is related to phage phi29 and is a Salasvirus within the *Herelleviridae* family. Morphologically, Goe16, Goe21 and Goe26 are myoviruses and Goe23 is a podovirus (Supplementary Table 1).

The type II CBASS system protected bacteria from infection with all four phages (Fig. 1b,c and Extended Data Fig. 1e–g). CBASS conferred 10$^4$-fold defence compared with the parental strain for phages Goe16, Goe21 and Goe26. There was no reduction in plaque-forming units (p.f.u.) count for Goe23 (Fig. 1b,c); however, the morphology of the plaques changed from a clear to a much more turbid plaque (Fig. 1b)

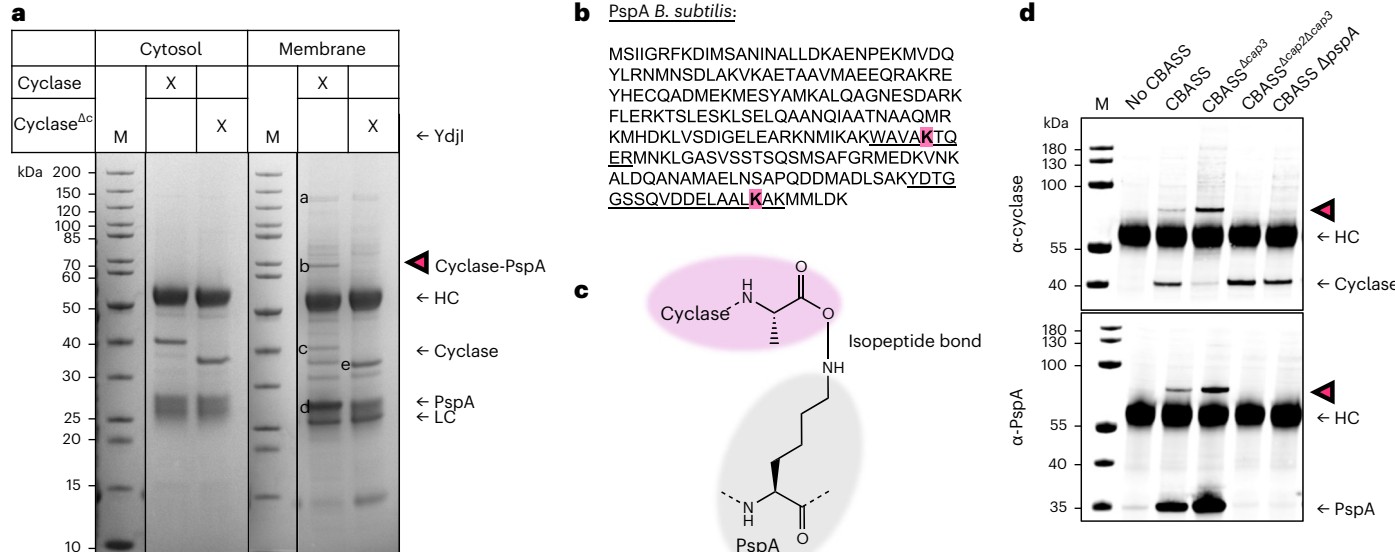

**Fig. 2 | The cyclase partitions between the cytoplasm and membranes, and is conjugated to PspA in vivo. a**, The full-length or truncated (1–300) cyclase was overexpressed from a plasmid in *B. subtilis* in the absence of Cap3 (strain LK40). Cultures were collected at exponential growth phase and cell lysates separated into cytosolic and membrane fractions. The cyclase was purified from both fractions using protein A-coupled beads and a cyclase-specific antibody, and analysed by SDS–PAGE. In the membrane fraction, additional higher-molecular-weight bands were detected for the full-length cyclase, with a prominent band below the 70 kDa band (magenta arrowhead). HC, heavy chain; LC, light chain; ΔC, deletion of C terminus of cyclase (aa 301–331). **b**, Amino acid sequence of

PspA from *B. subtilis* with the modified lysine residues highlighted (magenta) and the respective tryptic peptides identifed by MS underlined. **c**, Depiction of the conjugation site between PspA(K147/K220) and the cyclase (A331). **d**, The cyclase was purified from *B. subtilis* cell extracts expressing genomically integrated versions of CBASS (strains LK18 (control), LK06 (CBASS), LK20 (CBASS$^{Δcap3}$), LK10 (CBASS$^{Δcap2Δcap3}$), LK51 (CBASS ΔpspA)), separated by SDS–PAGE and analysed by western blot with specific antibodies (α-cyclase, α-PspA$^{Bsu}$). All experiments were conducted with at least 2 biological replicates. Detailed information on the genotypes of all strains can be found in Supplementary Table 2.

and we could observe protection in liquid medium (Extended Data Fig. 1e). Thus, the CBASS$^{Bce}$ system confers immunity against four different classes of phages (Fig. 1b).

To assess the essential components of CBASS, we deleted parts of the operon, or introduced point mutations in relevant residues to generate protein variants. The double deletion of the *cap2* and *cap3* genes led to a complete loss of immunity, as did deletion of the *cyclase* gene. In contrast, deletion of *cap3* reduced but did not abolish immunity (Fig. 1b,c). However, mutation of the catalytic residues of either the E2 (C91R) or E1 (C403R) domain of Cap2 led to loss of immunity (Fig. 1b,c). The C-terminal tails of cyclases are highly conserved even among distantly related type II CBASS systems and the last amino acid is typically an Ala or Gly residue that is conjugated to a lysine on cellular targets[18]. To investigate the importance of the C-terminal Ala, we mutated Ala331 (A331E) and observed loss of immunity (Fig. 1b,c).

### Bce cyclase is conjugated to PspA

Eukaryotic cGAS is regulated partly by association with the plasma membrane[24]. The subcellular localization of the bacterial cyclases could thus be an important aspect of their regulation. We addressed this by separating the cytosolic and membrane fractions of a CBASS expression strain and purified the cyclase by immuno-affinity chromatography. While most of the overexpressed cyclase was cytosolic, there was a substantial amount of the protein in the membrane fractions, irrespective of the presence or absence of the C-terminal tail (Fig. 2a and Extended Data Fig. 3a).

The membrane fractions contained additional bands (at 28, 65 and >200 kDa) associated with the full-length cyclase (Fig. 2a and Extended Data Fig. 2a). The 65 kDa band (magenta arrowhead) contained equivalent amounts of the cyclase and PspA, suggestive of a 1:1 conjugated species (Extended Data Fig. 2a). In *B. subtilis*, the *pspA-ydjG-ydjH-ydjI*

operon is activated by alkali stress and infection with phage SPP1 (ref. 25). YdjI, which was identified in the 200 kDa band (Fig. 2a and Extended Data Fig. 2b), is essential for membrane localization of PspA in *B. subtilis*[26,27]. The prominent band at 27 kDa was confirmed as PspA. Comparing the intensity of the band in the SDS gel of both the PspA monomer and the cyclase suggested that PspA was present in high molar excess (Fig. 2a and Extended Data Fig. 2b). PspA proteins assemble in helical rods with an ESCRT-III-like fold[28,29]. Thus, we hypothesize that the cyclase associates with and is conjugated to PspA rods in vivo.

To confirm that the cyclase and PspA proteins were indeed conjugated by an isopeptide bond and identify the sites on each protein, we searched the mass spectrometry (MS) data for PspA peptides carrying the last tryptic peptide of the cyclase, KPGGFA, which results in a predicted mass increase of 557.3 Da. We observed this mass shift on two residues, K147 and K220, of PspA (Fig. 2b and Extended Data Fig. 2b). This indicates that the C-terminal carboxyl group of the cyclase forms an isopeptide bond with the ε-amino group of a lysine residue in PspA (Fig. 2c).

To investigate which components of CBASS were required for cyclase conjugation under more realistic conditions in vivo, we purified cyclase from *B. subtilis* expressing genomically integrated CBASS and performed western blot analysis to determine the level of conjugation in the different strains. The cyclase–PspA conjugate only formed in the presence of cyclase, Cap2 and PspA, indicating that Cap2 is indeed the enzyme responsible for cyclase ligation to PspA (Fig. 2d). In the absence of PspA, we did not observe conjugation to an alternative target (Fig. 2d). The observation that Cap3 prevented formation of the cyclase–PspA conjugate strongly indicates that Cap3 cleaves the cyclase–PspA conjugate in vivo, as previously observed[18,30]. We then separated the cell extract of the same strains into membrane and cytosolic fractions, and purified the cyclase from the cytosolic fraction to concentrate the sample to the same volume

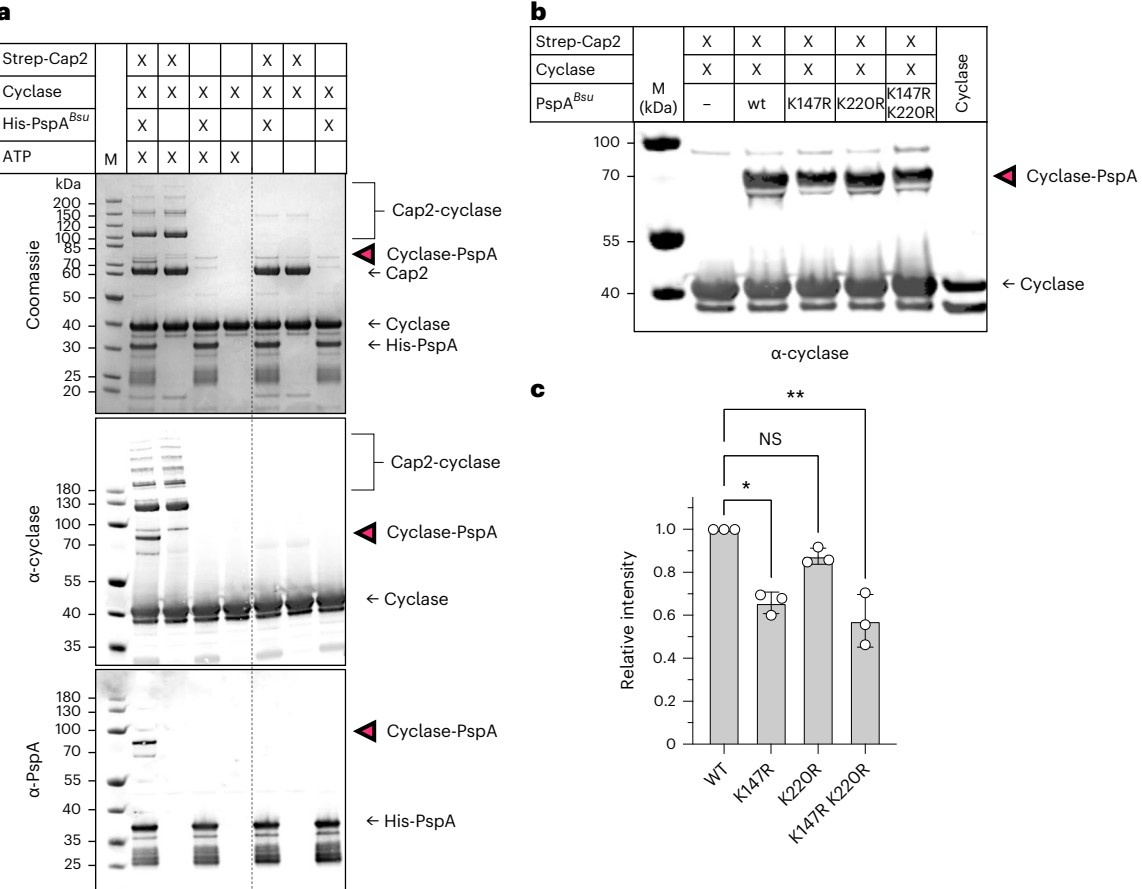

**Fig. 3 | Cap2 conjugates the cyclase to PspA. a**, In vitro conjugation of the cyclase to PspA. Reaction mixture contained 5 μM Strep-Cap2, 5 μM cyclase and 10 μM PspA, 10 mM MgCl₂, 1 mM ATP, 100 mM Tris-HCl pH 8 and 150 mM NaCl. Samples were incubated for 30 min at 25 °C, separated by SDS–PAGE and analysed by western blot with specific antibodies (α-cyclase, α-PspA). The cyclase–PspA conjugate is marked with magenta arrowheads. Strep-Cap2, 67.8 kDa; cyclase, 37.8 kDa; cyclase$^{ΔC}$, 34.2 kDa; 6xHis-V5-PspA, 30.2 kDa.

**b**, In vitro conjugation with PspA lysine mutants (reaction mix as in **a**). **c**, Quantification of **b**. Data are presented as mean ± s.d. WT, wild type. Statistical analysis was performed using one-way ANOVA, followed by Dunnett's multiple comparisons test (**$P$ = 0.0037; *$P$ = 0.0151; NS, not significant). All experiments were conducted with 3 biological replicates and representative gels/blots are shown.

as the membrane fraction. Western blot analysis showed that the cyclase partitions between both fractions (Extended Data Fig. 3). The association with the membrane fractions was independent of Cap2, Cap3 and PspA, providing evidence that the cyclase itself could interact with membranes (Extended Data Fig. 3).

**Conjugation in vitro**

To reconstitute conjugation in vitro, we incubated the purified proteins in a conjugation assay and analysed the reaction mixtures by sodium dodecyl sulfate–polyacrylamide gel electrophoresis (SDS–PAGE) and western blot using antibodies raised against the cyclase and PspA$^{Bsu}$. We observed conjugation of the cyclase to PspA$^{Bsu}$ in the presence of Cap2 and ATP (Fig. 3a). In addition, the cyclase formed higher-molecular-weight (MW) species in the presence of Cap2 and ATP, indicative of Cap2–cyclase thioester intermediates, as observed previously[18,19]. MS confirmed the presence of cyclase and Cap2 peptides in these bands (Extended Data Fig. 4a and Supplementary Information). The physiological relevance of these species is unclear, as they were not observed in *B. subtilis* (Fig. 2d).

To further investigate PspA conjugation to the cyclase, we created variant PspA$^{Bsu}$ proteins K147R and K220R to block conjugation at these positions. The K147R variant showed a 35% reduction in conjugation, while the K220R variant was still efficiently targeted for conjugation (Fig. 3b,c and Extended Data Fig. 4b), suggesting that K147 is a preferred

residue for conjugation but that additional unidentified lysine residues on PspA$^{Bsu}$ can also be targeted. PspA proteins show considerable sequence variability: the proteins from *B. subtilis* and *B. cereus* share only 31% sequence identity (Extended Data Fig. 4c). To confirm that Cap2 can also use PspA from *B. cereus* (PspA$^{Bce}$) as a substrate for cyclase conjugation, we purified the PspA$^{Bce}$ homologue and repeated the conjugation assays, observing conjugation of the cyclase to PspA$^{Bce}$ on five C-terminal lysine residues (Extended Data Fig. 4d). We hypothesize that interaction between Cap2 and PspA initiates conjugation of the C terminus of the cyclase to an exposed lysine residue on the C terminus of PspA, which is outward facing in PspA filaments[29].

**Influence of PspA–cyclase conjugation on CBASS activation**

To understand the relevance of conjugation during phage infection, we infected cells expressing either the full CBASS system or CBASS$^{Δcap3}$ with the phi29-like phage Goe23. Cells were collected before and 5, 15 and 30 min post infection, and the membranes purified. In the western blots, we observed that infection did not increase formation of the conjugate at any timepoint. In contrast, we observed a Cap3-dependent deconjugation of cyclase–PspA 30 min after infection (Fig. 4a,b and Extended Data Fig. 5a). We next deleted *pspA* and infected the bacteria with each of the four phages. Deletion of *pspA* had no negative effect on immunity, suggesting that conjugation to PspA is not essential for the eventual activation of the cyclase (Extended Data Fig. 5b). As the

CBASS system was overexpressed in our standard strains, we mutated the hyperactive $P_{degQ\text{-}H}$ promoter controlling CBASS expression back to the $P_{degQ}$ wild-type version to reduce gene expression levels[22]. The lower expression of CBASS prevented effective immunity in a plaque assay, regardless of the presence of PspA (Extended Data Fig. 5c). However, when we compared the initial response of these strains to phage infection in liquid medium, we found that the absence of PspA provided a subtle enhancement of protection against phage Goe21 and Goe26, but not Goe23 (Fig. 4c). This hints that conjugation of the cyclase to PspA becomes relevant under low CBASS expression, perhaps by slowing down activation of CBASS.

To explore this further, we grew *B. subtilis* cultures overexpressing CBASS and infected the cultures with phage Goe23 (MOI = 2). These cultures were collected at 0 and 30 min after infection, and intracellular nucleotides were extracted and used to activate the cognate CBASS Nuc-SAVED effector in a plasmid cleavage assay. We observed plasmid degradation only after infection and in cells with an active CBASS system (Extended Data Fig. 6). As expected, deletion of *cap3* resulted in a subtle phenotype, and deletion or mutation of *cap2* abolished immunity. To probe this further, we investigated a time course of plasmid digestion using nucleotides extracted at 30 min post infection for cells ±PspA (Fig. 4d and Extended Data Fig. 7). This revealed earlier activation of Nuc-SAVED (and thus increased cyclase activation) in the absence of PspA. We extended this approach to examine the kinetics of cyclase activation post infection over a range of time for three phages (Goe23, Goe21 and Goe26) with differing replication times. For Goe23, we observed a significant increase in plasmid cleavage activity with nucleotides extracted at 25 min post infection in the *pspA* deletion strain compared with the CBASS wild type (Fig. 4e and Extended Data Fig. 8), consistent with previous results. Nuclease activity was observed only after 35 min for Goe26 and after 45 min for Goe21, suggesting that activation of the CBASS cyclase occurs only late in infection[3], and we observed no significant effect arising from *pspA* deletion for the slower-replicating phage in this assay (Fig. 4f,g and Extended Data Fig. 8).

### Conjugation in the heterologous *Escherichia coli* host
To investigate the function of the *Bce* system further, we co-expressed His-tagged cyclase and Cap2 in *E. coli* and purified the cyclase. In contrast to previously reported observations[18,30], few higher-MW species were observed, suggesting that the final target for conjugation was missing in our heterologous system (Fig. 5a, lane 5). Thus, we co-expressed His-tagged cyclase and Cap2 together with *B. subtilis* PspA in *E. coli* and observed the formation of cyclase–PspA conjugates (Fig. 5a, lane 1, and Extended Data Fig. 10a). We identified K147 as the major target for modification by the C terminus of the cyclase, confirming our initial results (Extended Data Fig. 9a and Fig. 2b). As expected, the cyclase–PspA conjugate was not formed with the truncated cyclase (Fig. 5a, lane 4) or when the cyclase and PspA were co-expressed with

a catalytic dead mutant of Cap2 (Fig. 5a, lane 2/3). These data provide strong support for the hypothesis that PspA is the relevant acceptor protein for conjugation with the cyclase.

As expected from previous experiments, co-expression with Cap3 drastically reduced the formation of the cyclase–PspA conjugate (Fig. 5a, lane 6). We were unable to express the *Bce* Cap3 protein and substituted a Cap3 orthologue from *Cytobacillus oceanisediminis* (61% sequence identity to Cap3$^{Bce}$) to confirm the activity of Cap3 in vitro. To obtain the substrate, we purified the cyclase from the co-expression and separated monomeric cyclase from the cyclase–PspA conjugate by size exclusion chromatography (SEC) (Extended Data Fig. 9b). Incubation of the purified cyclase–PspA conjugate with Cap3$^{Coc}$ reversed conjugation, as expected (Fig. 5b and Extended Data Fig. 9c). We compared the mass of non-conjugated cyclase to that of deconjugated cyclase by intact MS and determined that they had the same mass (Extended Data Fig. 9e). In addition, we treated the two proteins with the peptidase AspN and analysed the peptides by MS, identifying the C-terminal peptide of the cyclase in both versions, which indicates that deconjugation by Cap3 is scarless (Extended Data Fig. 9d,e,f). In other words, Cap2-dependent conjugation and Cap3-dependent deconjugation can exist in equilibrium, with the cyclase shuttling between different forms.

### Activity of the *B. cereus* cyclase in vitro
We observed earlier that the *Bce* Nuc-SAVED effector (Cap4) was activated by synthetic (3′,3′,3′) cyclic triadenosine monophosphate ($cA_3$) (Extended Data Fig. 6), but note that CBASS cyclases are capable of making a variety of cyclic nucleotides with both 3′–5′ and 2′–5′ linkages[5]. We incubated the purified cyclase with two different ATP concentrations and analysed the products by high-performance liquid chromatography (HPLC) (Fig. 5c and Extended Data Fig. 10a,b). While the cyclase formed mainly $cA_3$ at low ATP concentration (0.2 mM), no $cA_3$ peak appeared when the cyclase was incubated with higher ATP concentration (5 mM). However, a minor peak observed in low ATP conditions became more prominent. To investigate which of these products activated the CBASS effector Nuc-SAVED, we performed a plasmid cleavage assay with the respective samples (Fig. 5d and Extended Data Fig. 10c). Only the sample containing mainly $cA_3$ induced plasmid cleavage through activation of Nuc-SAVED. Nuc-SAVED was not activated by the other products from the 5 mM ATP reaction, showing that $cA_3$ is the activator of the *Bce* CBASS effector. Since the 2′–5′ and 3′–5′ linked forms often co-elute on HPLC[5], we could not discriminate between the two isomeric forms of $cA_3$.

We quantified the levels of production of $cA_3$ in the low-ATP reaction samples using a standard curve generated with synthetic $cA_3$ (Extended Data Fig. 10d). The wild-type cyclase generated cyclic nucleotide at a very low level, with less than a single turnover per hour (0.33 and 0.24 $cA_3$ h$^{-1}$ calculated from two timepoints) (Fig. 5e). These data suggest that the cyclase is inactive in vitro. To investigate this further, we created a variant of the cyclase lacking residues 302–331

---

**Fig. 4 | Cyclase conjugation to PspA responds to phage infection. a**, Strains harbouring genomic integration of the CBASS operon (strains LK06 (CBASS), LK20 (CBASS$^{Δcap3}$)) were grown and infected with phage Goe23 to an MOI of 2. Cultures were grown for an additional 5, 15 or 30 min before collection. Membrane fractions were analysed by SDS–PAGE and western blot with a cyclase-specific antibody. Antibody α-Rny (against RNase Y) served as a loading control. mpi, minutes post infection. **b**, Quantification of reduction of cyclase–PspA conjugate upon infection. Data are shown relative to wild-type CBASS at timepoint 0 min. The experiment was conducted with $n = 3$ for CBASS$^{Δcap3}$ and $n = 4$ for CBASS biologically independent samples. Data are presented as mean ± s.d. Statistical analysis was performed using one-way ANOVA, followed by Tukey's multiple comparisons test (*$P = 0.0129$). **c**, Infection assay in liquid medium of strains containing CBASS components as indicated. Cultures were infected at mid-exponential growth with phage Goe23, Goe21 or Goe26 (MOI of 2). The experiment was conducted with 2 biologically independent replicates and

data are presented as mean ± s.d. **d**, Nucleotides were extracted from *B. subtilis* strains LK06 (CBASS) and LK51 (CBASS Δ*pspA*) 30 min after infection with Goe23 and used to activate the CBASS effector Nuc-SAVED using a plasmid cleavage assay. The experiment was conducted with 3 biological replicates, each with 3 technical replicates (all gels shown in Extended Data Fig. 7). Data are presented as mean ± s.d. **e,f,g**, CBASS (strain LK06) and CBASS Δ*pspA* (strain LK51) cells were infected with phage Goe23 (**e**), Goe21 (**f**) or Goe26 (**g**). Activation of Nuc-SAVED by the extracted nucleotides was assayed and quantified (exemplary agarose gel at the bottom; all replicates in Extended Data Fig. 8). All experiments were conducted with 3 biological replicates, each with 3 technical replicates. Data are presented as mean ± s.d. Statistical analysis was performed using one-way ANOVA, followed by Tukey's multiple comparisons test ($P$ values listed from left to right on the graph; Goe23 NS > 0.9999, *$P = 0.0454$, NS > 0.9999; Goe21 NS = 0.1821, NS = 0.9996; Goe26 NS > 0.9999, NS > 0.9999).

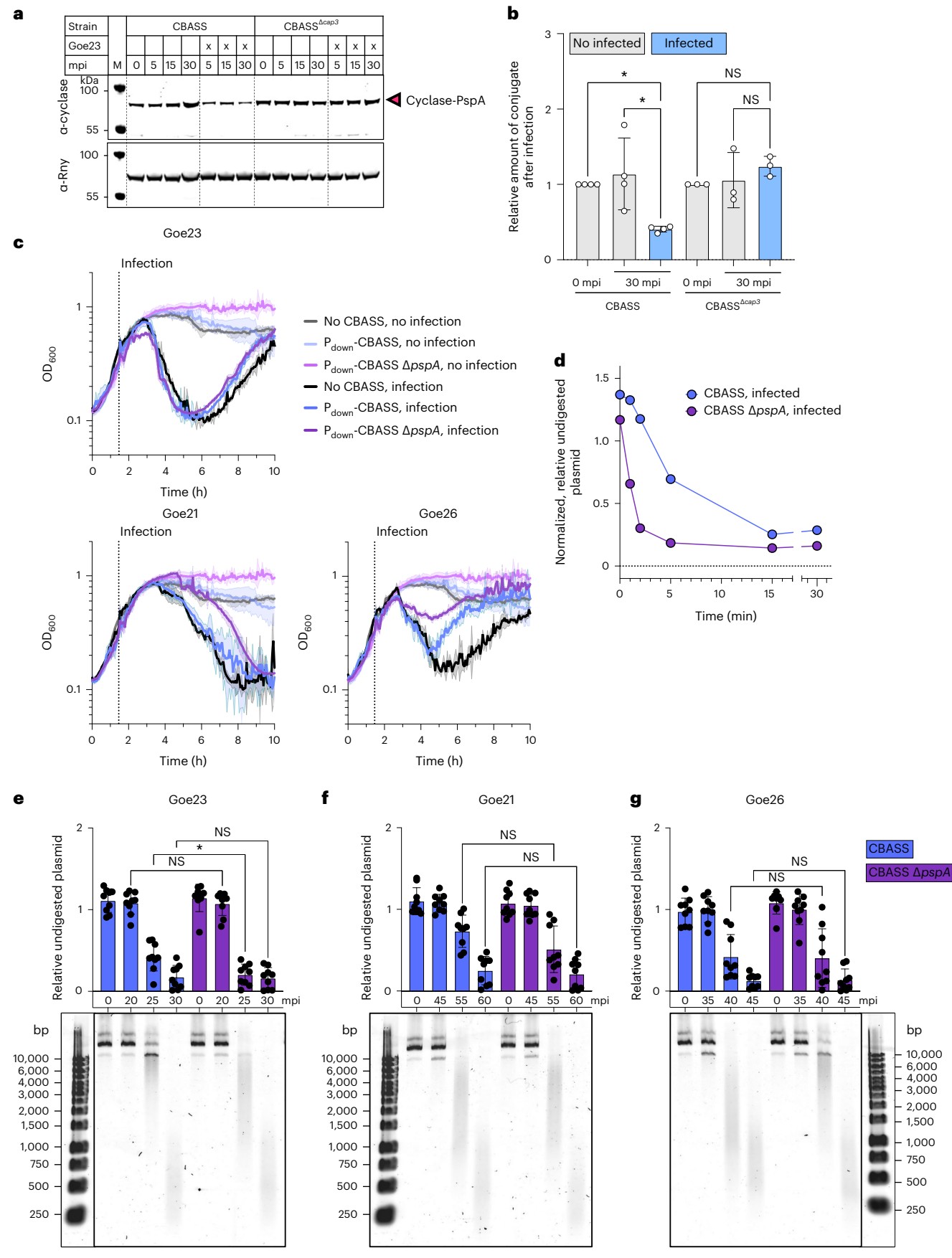

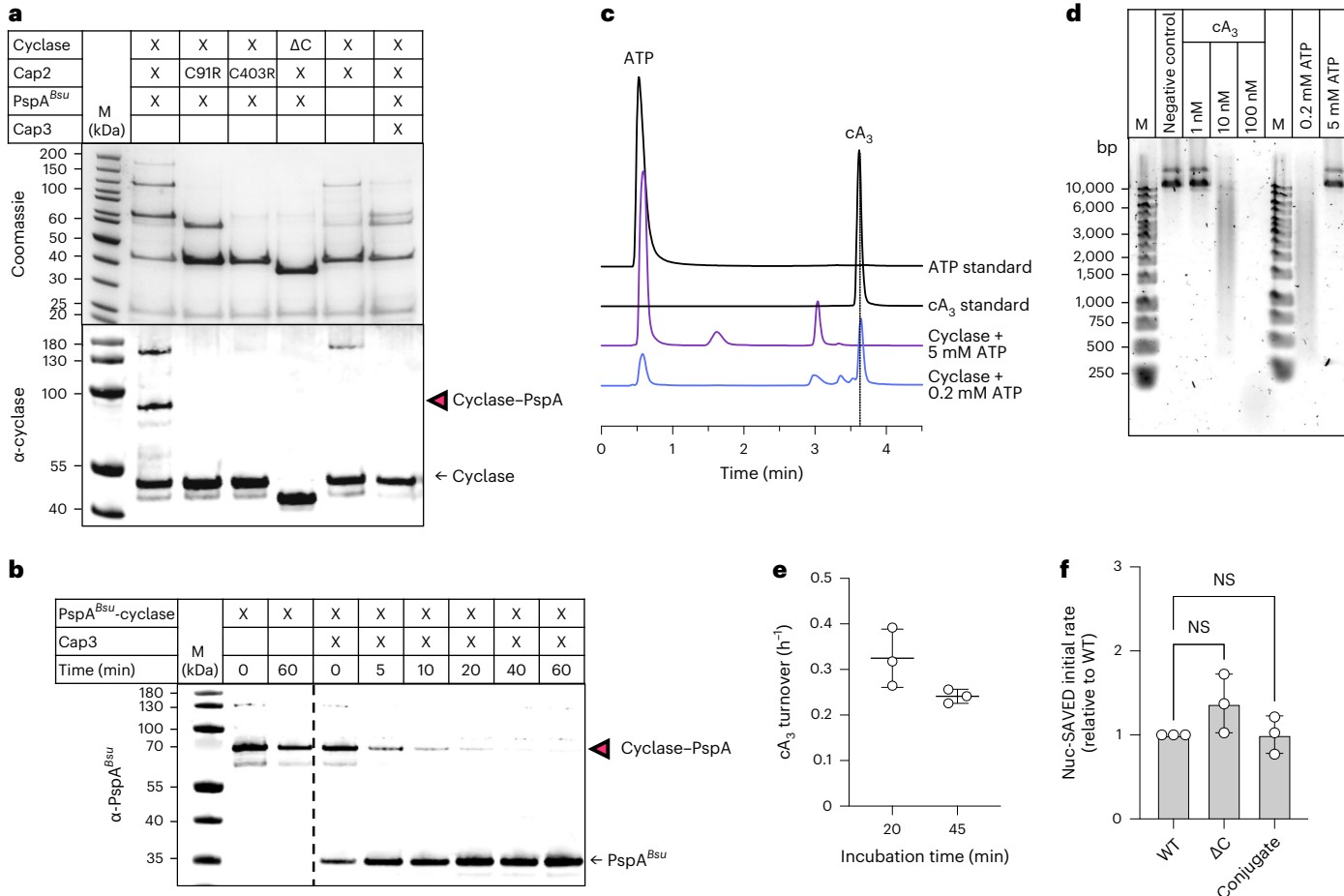

**Fig. 5 | Conjugation in the heterologous *E. coli* host. a**, Co-expression of the respective CBASS components from *Bacillus cereus* and PspA from *Bacillus subtilis* (*Bsu*) in *E. coli* and purification via His-tagged cyclase. Separation by SDS–PAGE and detection of the cyclase by western blot with a cyclase-specific antibody. The presence of PspA peptides in the conjugate band was confirmed by MS. ΔC, deletion of C terminus (aa 302–331). The experiment was conducted with 3 biological replicates and representative images from 1 replicate are shown. **b**, The cyclase–PspA conjugate was incubated together with the Cap3 orthologue from *Cytobacillus oceanisediminis* and the cleavage was followed by western blot with a PspA-specific antibody (see also Extended Data Fig. 9c). *n* = 3, a representative blot is shown. **c**, The reaction products from a cyclase reaction (50 μM cyclase, 0.2/5 mM ATP, 45 min incubation at 37 °C) were analysed by HPLC. The *y* axis is rescaled for presentation. The experiment was conducted with 3 replicates and a representative trace is shown. See Extended Data Fig. 10 for

replicates. **d**, The reaction products from **c** were tested for their ability to activate the CBASS effector Nuc-SAVED in a plasmid cleavage assay. *n* = 3, a representative gel is shown. See Extended Data Fig. 10 for replicates. **e**, The turnover number of the cyclase with respect to cA₃ production with 0.2 mM ATP as substrate was calculated from the HPLC data by measuring the area under the peak and fitting the data to a cA₃ standard curve (see Extended Data Fig. 10 for HPLC traces of replicates and linear regression of cA₃ standard). *n* = 3, data are presented as mean ± s.d. **f**, The activities of the wild-type cyclase, the truncated cyclase (1–301) and the cyclase–PspA conjugate were measured in a fluorescent DNA cleavage assay with the effector protein Nuc-SAVED. The initial rate of Nuc-SAVED was calculated and normalized to the protein amount, and set relative to the wild-type protein. *n* = 3, data are presented as mean ± s.d. Statistical analysis was performed using one-way ANOVA, followed by Dunnett's multiple comparisons test (NS = 0.1679).

that comprise the flexible tail, which cannot be conjugated, and compared the activity of the truncated protein with the wild-type cyclase and with the cyclase conjugated to the PspA protein, purified from *E. coli* (Extended Data Fig. 9b). Cyclase activity was measured by assaying the activity of the Nuc-SAVED effector in a fluorescent assay (analogous to that in ref. 31) where the release of a fluorescent signal, generated by cleavage of a 30-bp-long double-stranded DNA molecule containing a fluorescent probe and a quencher, was followed. Using this highly sensitive assay, we observed that all three cyclase species: conjugated, unconjugated and truncated, activated the Nuc-SAVED effector to the same extent, within error (Fig. 5f). Thus, our data indicate that the unconjugated, truncated and PspA-conjugated cyclase species are all inactive forms, suggesting that priming requires deconjugation from PspA by Cap3 and subsequent, as yet undetermined, steps to activate the cyclase and initiate the CBASS response.

## Discussion

Recent studies of type II CBASS systems have uncovered the important roles for Cap2-mediated conjugation and Cap3-mediated deconjugation of the cyclase, but left several mechanistic questions unanswered[18,19,30]. Our investigation of the system from *B. cereus* shows that Cap2 conjugates the cyclase to a specific acceptor protein: the major effector of the Psp response, PspA. This posttranslational modification is fully reversible by Cap3, raising the prospect of an equilibrium of conjugated and deconjugated cyclase that could represent a means to control the activation of the CBASS response. The activation of type II CBASS cyclases, which clearly requires a conjugation step, has been subjected to much speculation. In eukaryotes, cGAS is activated by DNA-induced oligomerization[32–34] and cyclase–cyclase conjugation has been proposed as the activation mechanism for a CBASS type II short system with only an E2 enzyme[19]. Previous studies of type II CBASS have revealed conjugation to a wide range of housekeeping

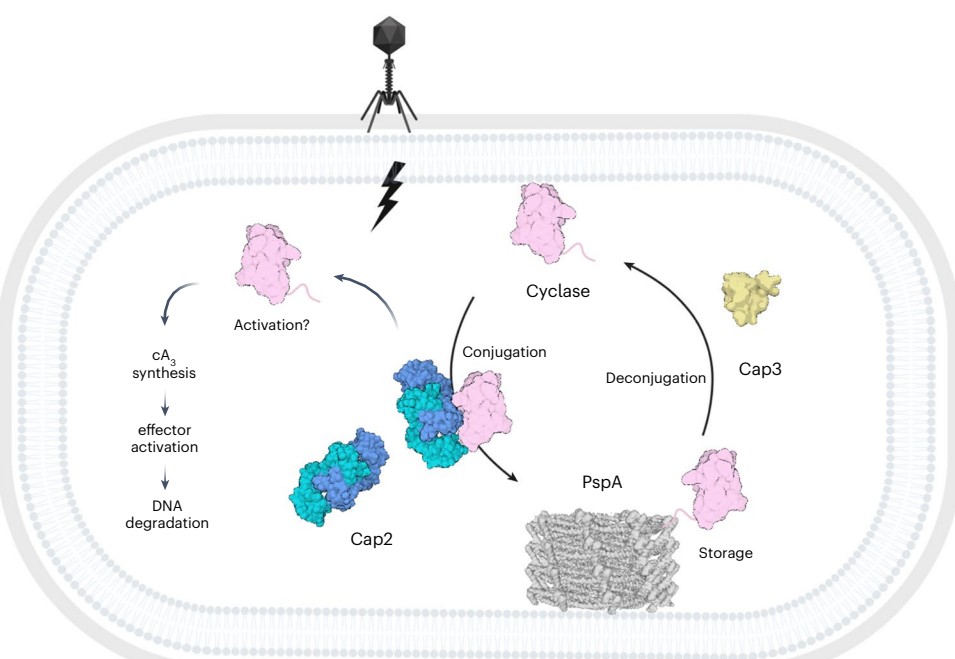

**Fig. 6 | Model of type II CBASS regulation through cyclase conjugation.**
In the absence of phage infection, Cap2 ligates the cyclase to PspA to prevent activation and induction of CBASS. The cyclase–PspA conjugate is cleaved by the isopeptidase Cap3. This equilibrium of unconjugated/conjugated cyclase is shifted towards unconjugated cyclase upon phage infection. Activation of the cyclase is Cap2-dependent and leads to degradation of double-stranded DNA through activation of the CBASS effector Nuc-SAVED, but the specific mechanism of activation remains unknown. Created with BioRender.com.

proteins, reminiscent of the background conjugation that might be observed when a protein partner is absent[30]. In contrast, by isolating the membrane-associated fraction of cyclase, our study has revealed specific conjugation to a single target protein, PspA. We hypothesize that by ligating the cyclase to a highly abundant protein such as PspA, the cyclase is distributed in the cell, preventing clustering and activation in the absence of infection. The ability of Psp systems to respond to phage infection by sensing cell envelope damage could add another level of regulation to CBASS systems, potentially regulating Cap3-mediated deconjugation and, thus, activation of CBASS defence.

The role of the Cap3 enzyme, which deconjugates cyclases from a range of partner proteins in vitro, has been a point of speculation[18,30]. In this study, the Cap3[Bce] enzyme increased immunity provided by the *Bce* CBASS system (Fig. 1b,c). In a study of an *E. coli* CBASS system, Cap3 was essential for immunity against phages T4 and T6 but not required for phages lambda and T2 (ref. 30). In contrast, when a *V. cholerae* CBASS system was tested in a heterologous *E. coli* host, Cap3 was not required for immunity against T2, T4, T5 or T6 phages[18]. In fact, deletion of Cap3 was associated with increased levels of cGAMP production by the *V. cholerae* cyclase immunoprecipitated from *E. coli* cells, leading the authors to propose a model whereby Cap3 is an antagonist of Cap2 that limits cyclase priming[18]. Finally, Cap3 was not required for antiphage immunity in a *P. aeruginosa* system[13].

The observed inhibition of the *Bce* cyclase at ATP levels typical for normally growing bacterial cells could have a regulatory function in vivo. Phage infection can alter intracellular nucleotide levels[35], a disturbance possibly affecting the activity of the cyclase and/or the Cap2 protein. It has been observed before that CBASS, in contrast to other defence systems, responds rather late during the infection cycle of the phage[3] and our observations support this idea (Fig. 4e,f,g). After 30 min post infection with phage Goe23, we detected a decrease in the amount of the cyclase–PspA conjugate in infected cells (Fig. 4a,b). On the basis of this and our observation of earlier accumulation of the second messenger in cells lacking PspA after infection with phage Goe23 (Fig. 4e and Extended Data Fig. 8), we suppose that deconjugation of the cyclase

is necessary for activation. This idea is supported by the observation that CBASS, under low expression, is impaired in the presence of PspA, but can provide some immunity against phage Goe21 and Goe26 in its absence (Fig. 4c). While we observed a direct effect on conjugation and cyclase activation with phage Goe23 in our overexpressed system, we had to downregulate CBASS expression massively to observe a PspA-dependent impairment of CBASS immunity with phage Goe21 and Goe26. This underlines the importance of cyclase conjugation to PspA in the natural host, where PspA concentration probably vastly outnumbers that of the CBASS cyclase. The rather subtle effects of PspA deletion observed in our study may be due in part to the use of artificial promoter systems for gene expression. Recent studies have revealed a role for the CapP-CapH signalling pathway, which provides a means to regulate CBASS expression in response to signals of phage infection such as DNA damage[36]. These genes are present upstream of the *B. cereus* CBASS operon studied here and are an important focus for future work.

A working model for the regulation of the *B. cereus* type II CBASS is presented in Fig. 6. In uninfected cells, newly synthesized cyclase is conjugated by Cap2 to the abundant cellular protein PspA. An equilibrium between Cap2-mediated conjugation and Cap3-mediated deconjugation may be established, which could be an effective means to prevent cyclase activation by oligomerization—a mechanism that appears conserved in both eukaryal and bacterial systems[19,32–34]. At later timepoints following phage infection, Cap3-mediated deconjugation of cyclase from PspA is observed, implying a role for Cap3 in sensing infection. It is possible that this relates to the changes in PspA organization upon phage infection[20,25], making deconjugation more favourable. The release of cyclase from its conjugation partner could then allow Cap2-mediated activation of the cyclase, possibly due to oligomerization of multiple cyclase molecules and concomitant activation, as observed for the type I CBASS system[19]. PspA is near-universal in bacteria[37], but was not observed as a conjugation partner in studies conducted in *E. coli*[18,30]. In *E. coli*, PspF forms an inhibitory complex with PspA in uninducing conditions, which may prevent interaction with

Cap2 (refs. [38],[39]). Moreover, the use of overexpression platforms and heterologous hosts, while essential for initial biochemical studies, can mask important aspects of these defence systems. Equally, alternative conjugation partners may be utilized in preference over PspA in other bacterial lineages. Finally, we have not directly observed the activated cyclase in the *B. cereus* system, so this remains a key point for future study. Ultimately, cyclases may be activated by conjugation to each other, forming covalently linked chains, as suggested for the type II short system[19].

## Methods

### Strains, media and growth conditions

*E. coli* DH5α and BL21 (DE3) or C43 (ref. [40]) were used for cloning and for the expression of recombinant proteins, respectively. All *B. subtilis* strains used in this study are derivatives of the laboratory strain 168. *B. subtilis* and *E. coli* were grown in Lysogeny Broth (LB) or in sporulation (SP) medium[40,41]. For growth assays and the in vivo interaction experiments, *B. subtilis* was cultivated in LB. The media were supplemented with ampicillin (100 µg ml⁻¹ for *E. coli*), kanamycin (50 µg ml⁻¹ for *E. coli*, 10 µg ml⁻¹ for *B. subtilis*), chloramphenicol (35 µg ml⁻¹ for *E. coli*, 5 µg ml⁻¹ for *B. subtilis*), erythromycin and lincomycin (2 and 25 µg ml⁻¹, respectively), or Zeocin (35 µg ml⁻¹) if required.

### Isolation of novel *B. subtilis* phages

The laboratory strain *B. subtilis* Δ6 (ref. [23]), a derivate of *B. subtilis* 168, was used as host for the isolation of novel phages as described previously[42]. Briefly, raw sewage from a municipal sewage treatment plant in Göttingen, Germany served as environmental phage source and was cleared by centrifugation at 5,000 *g* for 10 min, following filtration through a sterile filter (0.45 µm, Sarstedt). The host was infected by mixing 100 µl of an overnight culture with 2 ml of processed sewage water, followed by 5 min incubation at room temperature (r.t.) to allow adsorption. This bacterial suspension was then mixed with 2.5 ml of 0.4% agarose (50 °C) (Fisher BioReagents) dissolved in LB medium, spread on a LB plate and incubated overnight at 28 °C. Individual plaques were picked, serially diluted and higher dilutions were used for re-infection of the host. Cells were grown until complete lysis and the supernatant processed as described above. This re-infection was repeated to scale up the phage lysate. For genome sequencing of the novel isolates, phage DNA was isolated as previously described[43]. Sequencing was performed using an Illumina MiSeq sequencer (Illumina). Phages were classified according to nucleotide sequence homology to the closest relative as revealed by Megablast in the Geneious Prime 2022.2.1 software. The genomic sequences of the new phages and their closest relative (downloaded from NCBI on 10 March 2023) were aligned using MMseqs in the pyGenomeViz package (https://github.com/moshi4/pyGenomeViz, accessed on 10 April 2023; input: gbk files; mode: pgv-mmseqs). A list of all phages isolated in this study can be found in Supplementary Table 1.

### Phage amplification and storage

Phage lysates were generated from *B. subtilis* Δ6 using the modified double agar overlay plate technique. For this, a culture of *B. subtilis* Δ6 was precultured to late exponential growth phase in LB medium. Bacterial cells (100 µl) were mixed with 100 µl phage dilution, incubated for 5 min at r.t. to allow adsorption of the phage, mixed with 2.5 ml LB supplemented with 0.4% agarose (50 °C) and poured over a prewarmed (37 °C) LB plate. Plates were incubated overnight at 37 °C. Phages were collected the following day by adding 5 ml of LB medium directly onto the plate, incubating for at least 30 min at r.t., and collecting and filtering the resulting liquid through a 0.2 µm Nanosep filter. Lysate titres were determined by infection of *B. subtilis* Δ6 using the modified double agar overlay plate technique as described above and the resulting p.f.u. ml⁻¹ was calculated. Phage lysates were stored at 4 °C.

### Phage infection assays

Phage infection assays were performed on plates and in liquid culture. The desired *B. subtilis* strains were precultured overnight at 28 °C and used to inoculate fresh LB medium. This culture was grown until late exponential growth phase. For infection on plates, 100 µl were mixed with 2.5 ml LB supplemented with 0.4% agarose (50 °C) and poured over a prewarmed LB plate. Serial dilutions of the phage lysate were spotted onto this plate. Plates were incubated overnight at 37 °C and pictures were taken the following day. For quantification, p.f.u. ml⁻¹ was calculated from at least three biological replicates. Fold defence was calculated as the ratio between the p.f.u. ml⁻¹ obtained from wild type (no CBASS) and strains containing the defence system. For infection in liquid medium, the optical density of the culture at 600 nm (OD₆₀₀) was adjusted to 1.0. This culture was then used to inoculate fresh LB medium in a 96-well plate (Microtest Plate 96-well, Sarstedt) to OD₆₀₀ of 0.1 (1:10 dilution). Growth was tracked in an Epoch 2 microplate spectrophotometer (BioTek Instruments) at 37 °C with linear shaking at 237 cpm (4 mm) for 20 h, and OD₆₀₀ was measured at 2 min intervals.

### DNA manipulation

Transformation of *E. coli* and plasmid DNA extraction were performed using standard procedures[40]. All commercially available plasmids, restriction enzymes, T4 DNA ligase and DNA polymerases were used as recommended by the manufacturers. Chromosomal DNA of *B. subtilis* was isolated as previously described[36]. *B. subtilis* was transformed with plasmid and genomic DNA according to the two-step protocol[36].

### Construction of mutant strains by allelic replacement

Deletion of the *pspA* gene was achieved by transformation of *B. subtilis* Δ6 with a PCR product constructed using oligonucleotides to amplify DNA fragments flanking the target genes and an appropriate intervening resistance cassette as previously described[44]. The integrity of the regions flanking the integrated resistance cassette was verified by sequencing PCR products of ~1,100 bp amplified from chromosomal DNA of the resulting mutant strains. A list of all strains constructed in this study can be found in Supplementary Table 2.

### Plasmid construction and mutagenesis

The wild-type CBASS operon from *B. cereus* WPysSW2 was ordered as genomic DNA from Integrated DNA Technologies (IDT) and the desired genes were amplified using appropriate oligonucleotides that attached specific restriction sites to the fragment. Plasmids derived from pGP1460 (integration into *lacA*)[45] were linearized with ScaI for genomic integration. The integrity of the integration was confirmed by PCR and subsequent sequencing of the region.

For protein expression in *E. coli*, the genes encoding CD-NTase, Cap2, Cap3, Nuc-SAVED and PspA*Bce* were codon optimized and obtained from IDT. Each gene was subcloned into the vector pEhis-V5TEV[46], allowing expression with a cleavable N-terminal polyhistidine tag, or into pGP172 for expression with an N-terminal Strep-tag. The *pspABsu* gene was amplified from *B. subtilis* 168 chromosomal DNA using oligonucleotides that attached specific restriction sites to the fragment and cloned into the vector pEhisV5TEV. The PspA variants K147 and K220 were generated by site-directed mutagenesis of the *pspA*-pEhisV5TEV vector. For co-expressions in *E. coli*, the cyclase gene remained within the pEhisV5TEV vector to retain cleavable polyhistidine tag, while other proteins (Cap2, Cap3 and PspA*Bsu*) were expressed on pCDFduet and pACYCduet. All synthetic genes, plasmids and primers are listed in Supplementary Tables 3–5.

### Protein expression and purification

*E. coli* C43 (DE3) was transformed with the plasmid pEhisV5TEV encoding the wild-type *Bce* cyclase or a variant truncated after amino acid 301, Nuc-SAVED, Cap2, Cap3, wild-type PspA, or the variants, or pGP172 encoding Strep-Cap2. Expression of the recombinant proteins was

induced by the addition of isopropyl 1-thio-β-D-galactopyranoside (final concentration, 0.4 mM for pEhisV5TEV or 1 mM for pGP172) to exponentially growing cultures ($OD_{600}$ of 0.8) of *E. coli* carrying the relevant plasmid. The *Bce* cyclase and Cap3 were induced overnight at 25 °C. The truncated cyclase was induced for 4 h at 25 °C. The Nuc-SAVED protein and Strep-Cap2 were induced overnight at 16 °C. Cell pellets were resuspended in lysis buffer (50 mM Tris-HCl, 250 mM NaCl, 10 mM imidazole, 10% glycerol, pH 7.5) supplemented with 1 mg ml$^{-1}$ lysozyme and protease inhibitor (Roche). The cells were lysed by sonication on ice six times for 1 min at an amplitude of 12, with 1 min pause on ice in between (MSE Soniprep 150). After lysis, the crude extract was centrifuged at 117,734 × *g* for 30 min and then passed over a Ni$^{2+}$nitrilotriacetic acid column (His-trap FF crude, Cytiva). Ni$^{2+}$nitrilotriacetic acid columns were washed with wash buffer (50 mM Tris-HCl, 250 mM NaCl, 30 mM imidazole, 10% glycerol, pH 7.5) to clear unbound protein. Target protein was eluted with an imidazole gradient or biotin (5 mM). After elution, the fractions were tested for the desired protein using SDS–PAGE. To remove the 8x-His-TEV tag from the proteins, the relevant fractions were combined and the tag was removed with the TEV protease (ratio 10:1 w/w) during overnight dialysis (Biodesign Cellulose Dialysis Tubing Roll, 10 kDa) against wash buffer. The cleaved TEV moiety and the protease were removed using a fresh Ni$^{2+}$nitrilotriacetic acid column. The purified protein was concentrated in a Merck Amicon Ultra-15 Centrifugal Filter device (cut-off 10 kDa). The protein was loaded on a HiLoad 16/600 Superdex 200 pg column pre-equilibrated with gel filtration buffer (20 mM Tris-HCl, 250 mM NaCl, 10% glycerol, pH 7.5) and the fractions containing pure protein were collected and concentrated again. The protein samples were stored at −70 °C until further use.

Expression of wild-type PspA$^{Bsu}$ or the protein variants and PspA$^{Bce}$ was induced with 0.4 mM isopropyl-β-D-1-thiogalactoside (IPTG) and cells grown for 4 h at 25 °C. PspA was purified as described above, except that the Ni$^{2+}$nitrilotriacetic acid column was washed with 20 column volumes (CV) of wash buffer (containing 50 mM Tris-HCl pH 7.5, 500 mM NaCl, 30 mM imidazole and 10% glycerol), then 4 CV of 50 mM Tris-HCl pH 7.5, 500 mM NaCl, 50 mM imidazole and 10% glycerol, and then eluted directly with 6 CV of elution buffer (containing 50 mM Tris-HCl pH 7.5, 500 mM NaCl, 500 mM imidazole and 10% glycerol). The eluted peak was pooled and concentrated using an Amicon Ultra-15 (Millipore) centrifugal filter (MW cut-off 10 kDa) before loading onto an equilibrated HiTrap desalting column (Cytiva), then washed with 50 mM Tris-HCl pH 7.5, 500 mM NaCl, 30 mM imidazole and 10% glycerol. The desalted protein was collected and treated with TEV protease for 2 h at r.t. with gentle agitation. The protein was isolated from TEV protease by passing through another His-Trap FF crude column, and the unbound fraction collected and concentrated. PspA was further purified by SEC (S200 16/60, Cytiva) in buffer containing 20 mM Tris pH 7.5, 500 mM NaCl and 10% glycerol. Chosen fractions were concentrated and aliquoted, then frozen at −70 °C.

Strep-tagged proteins were expressed, lysed and ultracentrifuged as described for His-tagged proteins above in buffer W (100 mM Tris-HCl, 150 mM NaCl, pH 8.0), unless stated otherwise. The crude extract was then applied to a StrepTactin column (IBA Lifesciences) and the column washed with 100 mM Tris-HCl and 150 mM NaCl pH 8.0. The protein was eluted with a biotin gradient (5 mM). Relevant fractions were identified by SDS–PAGE and concentrated using Vivaspin turbo 15 (Sartorius) before storage as described above.

## Co-expression and purification of the cyclase$^{Bce}$–PspA$^{Bsu}$ conjugate

Co-expression strains (His-tagged cyclase$^{Bce}$, untagged PspA$^{Bsu}$ and Cap2$^{Bce}$) were generated by transformation of *E. coli* C43 (DE3) with pV5SpHISTEV containing wild-type cyclase$^{Bce}$ or the truncated variant (1–301), in addition to the dual-expression vector pCDFduet that contained PspA$^{Bsu}$ and Cap2$^{Bce}$. Cultures were induced for protein expression with 0.4 mM IPTG for 4 h at 25 °C, conditions previously identified as optimum for cyclase–PspA conjugate formation[46]. Cyclase–PspA conjugates were purified following the His-tagged protein method detailed above. As the only protein with His-tag present was the cyclase, minimal Cap2/PspA were expected unless conjugated to/interacting with cyclase. Multiple cyclase–PspA conjugation products were separated through SEC on a HiLoad 26/600 Superdex 200 pg column, at a flow rate of 1.5 ml min$^{-1}$ for increased separation of varying-MW species. Fractions containing individual conjugation species were then identified via SDS–PAGE and pooled separately before concentrating and storing at −70 °C as above.

## Pulldown of cyclase$^{Bce}$ in *E. coli*

Co-expression strains of the His-cyclase and PspA with or without additional expression of Cap2 or Cap3 were grown in 10 ml LB and induced during exponential growth phase with 0.4 mM IPTG. Expression strains were then incubated for 4 h at 25 °C and cells collected by centrifugation. His-tagged cyclase was purified and visualized on SDS–PAGE as previously described[46]. Briefly, cell pellets were resuspended in lysis buffer (50 mM Tris-HCl, 250 mM NaCl, 10 mM imidazole, 10% glycerol, pH 7.5) before being lysed by sonication on ice. The cell lysate was then centrifuged at 10,000 × *g* for 10 min before supernatant was added to Biosprint plasticware. His-tagged proteins were isolated using Nickel beads (His Mag Sepharose Ni, GE Healthcare) in the QIAGEN BioSprint machine, with wash and elution buffers containing 50 mM Tris-HCl, 500 mM NaCl, 10% glycerol (pH 7.5) and 30/500 mM imidazole, respectively. Eluted His-tagged protein and total protein-containing fractions were analysed by SDS–PAGE and western blot.

## Purification of cyclase$^{Bce}$ from *B. subtilis* and separation of cell extract

For overexpression of the cyclase in *B. subtilis*, strain LK40 was transformed with either pLK20 (Strep-cyclase), pLK21 (Strep-cyclase$^{\Delta C}$) or the empty vector control (pGP380). For comparison of the levels of cyclase conjugate, the *B. subtilis* strains LK06, LK10, LK18, LK20 and LK51 containing single genomic integrations of the CBASS operon were used. The respective strains were cultivated in LB medium until exponential growth phase was reached ($OD_{600}$~0.5). If desired, cells were infected with the phage Goe23 and incubated for another 5, 15 or 30 min. The cells were collected by centrifugation at 3,220 *g* and 4 °C, and the pellets stored at −20 °C. For separation of cytosolic and membrane fractions of the crude extract, the pellets were resuspended in buffer M (50 mM NaH$_2$PO$_4$, 50 mM Na$_2$HPO$_4$, pH 6.8) supplemented with 1 mg ml$^{-1}$ lysozyme and protease inhibitor (Roche), and opened by sonication. To remove cell debris and unbroken cells, the lysate was centrifuged at 3,220 *g* for 20 min at 4 °C. The supernatant was subjected to ultracentrifugation at 21,000 *g* for 90 min at 4 °C to pellet the membranes. The supernatant containing cytosolic proteins was removed, the pellet washed with buffer M and the membranes pelleted again in an additional centrifugation step (21,000 *g*, 60 min, 4 °C). The supernatant was discarded and the wash step repeated one more time. The resulting pellet containing the membranes was resuspended in buffer M. To achieve solubilization of the membranes, the fraction was 1:1 diluted with buffer M containing 0.5% n-Dodecyl-β-D-maltoside () (final concentration of 0.25%) and incubated overnight with rotation at 4 °C. The protein concentration of cytosolic and membrane fractions was determined, and equal amounts of the fractions were separated on an SDS gel and analysed by staining with Coomassie blue and western blot analysis. To confirm the proper separation of cytosolic and membrane fractions, membranes were also developed with α-HPr and α-Rny primary antibodies as cytosolic and membrane protein controls, respectively. The cyclase was further purified from both cytosolic and membrane fractions or the entire cell lysate using Protein A-coupled Dynabeads (Invitrogen) that were saturated with α-cyclase antibody for immunoprecipitation. The eluates were separated by SDS–PAGE

and analysed by western blot. Bands of interest were subjected to further analysis by MS. For purification of the cyclase from cell lysates without separation into cytosolic and membrane fractions, the pellets were resuspended in buffer W (100 mM Tris-HCl, 150 mM NaCl, pH 8.0) supplemented with 1 mg ml$^{-1}$ lysozyme and protease inhibitor (Roche), and opened by sonication. To remove cell debris and unbroken cells, the lysate was centrifuged at 20,000 $g$ for 10 min at 4 °C. Protein concentration of the lysates was determined and adjusted with buffer W. The cyclase was purified from the lysates using Protein A-coupled Dynabeads (Invitrogen) saturated with α-cyclase antibody. The eluates were separated by SDS–PAGE and analysed by western blot.

### Western blot analysis

Polyclonal antibodies against the *B. cereus* cyclase or the PspA protein from *B. subtilis* were produced in rabbits (by Kaneka Eurogentec), and the anti-cyclase antibody was purified with Cyanogen bromide-activated Sepharose (Merck) by immobilizing purified cyclase. The purified α-cyclase antibody was used at 1:10,000 dilution for detection and the α-PspA$^{Bsu}$ serum at 1:5,000 dilution. The α-HPr antibody was used at 1:10,000 dilution and the α-Rny antibody at 1:100,000 dilution. Western blot analysis was performed on cell extracts and purified protein to follow the modification of the cyclase protein. Samples were resolved using SDS–PAGE and transferred to PVDF membranes. Membranes were blocked for at least 1 h at r.t. in blocking buffer (1× PBS, 5% w/v milk powder, 0.025% Tween20). Incubation with the respective antibodies (α-cyclase, α-HPr, α-Rny) was carried out overnight at 4 °C. Blots were extensively washed and incubated with the α-rabbit secondary antibodies (IRDye 800CW donkey anti-rabbit IgG secondary antibody; LI-COR) at 1:20,000 dilution in blocking buffer. The specific protein bands were visualized using the LI-COR Odyssey CLx and images adjusted using the ImageJ software (http://rsb.info.nih.gov/ij/index.html). Bands were analysed and quantified with ImageJ by comparing the intensity of selected bands to a standard band on the same blot. Data were analysed and visualized using GraphPad Prism 9.5.1 (Dotmatics).

### Conjugation assay

For reconstruction of the conjugation in vitro, the cyclase (5 µM) was incubated together with Cap2 (5 µM), PspA (5/10 µM), 10 mM MgCl$_2$ and 1 mM ATP in 100 mM Tris-HCl and 150 mM NaCl (pH 8.0) for 30 min at 25 °C. The samples were separated by SDS–PAGE and analysed by Coomassie staining or western blot with an antibody raised against the cyclase.

### Mass spectrometry analysis

Protein bands were excised from the gel and prepared for MS analysis using established protocols[47]. Briefly, this included destaining with ethanol:water, reduction with dithiorethritol and alkylation with iodoacetamide. Samples were digested with either trypsin, AspN or a combination of trypsin and GluC. The peptides from the gel pieces were soaked and the eluent concentrated to 20 µl. The samples were analysed by nanoLC–MS/MS on a ThermoScientific Fusion Lumos Orbitrap mass spectrometer coupled to a ThermoScientific u3000 nanoLC. An LC was configured in trap elute format, with the Acclaim Pepmap 100 (100 µm × 2 cm) nanoViper trap and the Pepmap RSLC C18 3 µm 100A (75 µm × 15 cm) Easyspray column (both ThermoScientific). Of the sample, 5–10 µl was injected onto the trap at 15 µl min$^{-1}$ of loading buffer (0.05% trifluoroacetic acid in water) and run for three min. The trap was switched in line with the analytical column and the sample eluted at a gradient over 65 min (A = 100% water with 0.1% formic acid, B = 20% water 80% acetonitrile, 0.1% formic acid, 2% A to 3 min, linear to 40% A over 42 min, linear to 95% A over 4 min, hold for 5 min, linear back to 2% A and re-equilibrate for 10 min). The flow from the column was sprayed directly into the Easyspray orifice at a voltage of 1,700 V positive ionization. MS data were collected from 350 to 2,000 on the

orbitrap at a resolution of 120,000 for the survey scan and a cycle time of 2 s for data-dependant acquisition conditions for MSMS on the orbitrap trap at a resolution of 30,000. Both electron transfer dissociation (ETD) and higher energy collison dissociation (HCD) fragmentation techniques were used. Raw data were exported and extracted using ms convert (ProteoWizard). The data were searched using the Mascot search engine (MatrixScience) against an internally generated database of 6,700 protein sequences containing the sequences of our recombinant proteins or the *B. subtilis* specific protein database (UniProt Proteome ID UP000001570). Settings were 20 ppm on the MS and 0.1 Da on the MSMS data, with a fixed modification of carbamidomethyl on cysteine and variable oxidation of methionine. For the identification of the conjugation site on PspA, the mass of the last C-terminal tryptic peptide of the cyclase (KPGGFA) minus one water molecule (C(27) H(39) N(7) O(6) 557.296182) was set as a variable modification on lysine residues. Peptides identified as modified are shown in Supplementary Table 6.

Intact mass measurement was carried out on a Waters Xevo G2S TOF mass spectrometer with Waters Acquity LC. A volume of 10 µl of 1 µM sample was desalted online through a Waters MassPrep On-Line Desalting Cartridge (2.1 × 10 mm), eluting at 200 µl min$^{-1}$ with an increasing acetonitrile concentration (2% acetonitrile, 98% aqueous 1% formic acid to 98% acetonitrile, 2% aqueous 1% formic acid) and eluted directly into the MS with a lock mass of LeuEnk to ensure stable calibration. The spectra across the elution peak were combined and the charged ion envelope deconvoluted with MaxEnt algorithm using the peak width at half height of the most intense peak, to a resolution of 0.1 Da.

### Cyclase product quantification by HPLC

To obtain cyclase products, the reaction was set up as follows: 50 µM cyclase, 0.2 mM/5 mM ATP in cyclase buffer (50 mM HEPES, 150 mM KCl, 10 mM MgCl$_2$, 10% glycerol, pH 7.5) for 20 min/45 min. Reaction samples were filtered through a Pall Nanosep spin filter (3 kDa cut-off) by centrifugation at 10,000 $g$ for 10 min to remove protein. A volume of 5 µl of product or synthetic standard was injected onto a C18 column (Kinetex EVO P 2.1 ×50 mm, particle size of 2.6 µm) attached to a Thermo UltiMate 3000 chromatography system. Absorbance was monitored at 260 nm and 40 °C. Gradient elution was performed with solvent A (100 mM ammonium acetate) and solvent B (100% methanol plus 0.1% trifluoroacetic acid) with a flow rate of 0.3 ml min$^{-1}$ as follows: 0–0.5 min, 0% B; 0.5–3.5 min, 20% B; 3.5–5 min, 50% B; 5–10 min, 100% B. The area under the peak (mAU (milli-absorbance units) × min) was determined with the Chromeleon 6.80 software (Dionex). The activity of the cyclase was calculated by fitting the area under the peak values to those of a cA$_3$ standard curve. Data were analysed and visualized using GraphPad Prism 9.5.1 (Dotmatics).

### Cyclase product quantification by fluorescent Nuc-SAVED assay

Wild-type cyclase$^{Bce}$, the truncated variant (1–301) or the cyclase$^{Bce}$–PspA$^{Bsu}$ conjugate were incubated with ATP under standardized conditions for comparison of product formation. Cyclase (20 µM) was incubated with 250 µM ATP in cyclase buffer (50 mM HEPES, 150 mM KCl, 10 mM MgCl$_2$, 10% glycerol, pH 7.5) for 1 h, unless stated otherwise. Reactions were quenched with 10 mM EDTA and cyclase denatured at 95 °C for 5 min. For rapid quantification of the relative cA$_3$ produced by the cyclase variants, a dilution of the cyclase reaction was incubated with the CBASS effector Nuc-SAVED. Method and conditions were adapted from an analogous previously described NucC assay[31]. The assay contained 50 mM Tris-HCl (pH 7.5), 20 mM NaCl, 10 mM MgCl$_2$, 10% (v/v) glycerol, 100 nM FAM:Iowa Black double-stranded DNA substrate, 100 nM Nuc-SAVED and 500× dilution of denatured cyclase reaction or 25 nM synthetic cA$_3$, unless otherwise stated. The absorbance was measured with a FluoStar Omega plate reader

(BMG Labtech) using fluorescence detection (ex/em 485/520 nm) in black non-binding half-area 96-well plates (Corning). Fluorescence was measured at 30 s intervals at 37 °C. The reaction was initially incubated in the absence of cA$_3$/cyclase product for 10 min to measure a baseline. The measurement was stopped, synthetic cA$_3$/cyclase product added and the measurement continued for 50 min. A standard curve was generated for Nuc-SAVED with varying synthetic 3′5′ cA$_3$ concentrations (0.1 nM–1 µM) and the initial rate of reaction used to generate a standard curve of rate vs [cA$_3$]. For the comparison of the different cyclase variants, the cyclase reaction sample was analysed by SDS–PAGE additionally to normalize the protein amount to the initial rate, as the conjugate sample was less pure than the wild type and the truncated cyclase. Bands were analysed and quantified with ImageJ by comparing the intensities of selected bands on the same gel. Data were analysed and visualized using GraphPad Prism 9.5.1 (Dotmatics).

### Extraction of nucleotides

For the extraction of nucleotides, *B. subtilis* cultures carrying genomic integrations of the respective CBASS version were grown until exponential growth phase at 37 °C. If indicated, the cultures were infected with phage Goe21, Goe23 or Goe26 to a multiplicity of infection (MOI) of 2 and grown for additional time as indicated. A volume of 50 ml of the cultures was collected immediately and the pellets resuspended in extraction solution (methanol:water:acetonitrile, 2:2:1). The cells were lysed and the proteins denatured by vortexing for 20 s and incubation at 95 °C for 10 min. The samples were stored at −70 °C overnight. After thawing, the denatured proteins were separated by centrifugation at 21,000 *g* and the protein pellet used to determine the protein amount in each sample. The supernatants containing the cell-free extracts were dried at 40 °C in a Speed-Vac. The dried pellets were resuspended in 60 µl of water and filtered through a Pall Nanosep spin filter (3 kDa cut-off) by centrifugation at 10,000 *g* for 10 min. The filtrate was tested for the CBASS-specific nucleotide in a plasmid cleavage assay.

### Plasmid cleavage assay

The assay contained 20 mM HEPES (pH 7.5), 10 mM NaCl, 10 mM MgCl$_2$, 10% (v/v) glycerol, 75 nM plasmid DNA, 1 µM CBASS effector protein Nuc-SAVED and 1.5 µl of the extracted nucleotides or the diluted cyclase reaction product (1:1,000/1:10,000) per 15 µl reaction sample or synthetic cA$_3$, unless stated otherwise. The reaction was started by the addition of the nucleotide extracts or the synthetic standard, and the samples incubated at 37 °C for 30 min, or 10 min in the case of the cyclase reaction products or, in the case of the time course experiment, for the stated amount of time. The samples were mixed with DNA loading dye (TriTrack, ThermoScientific) and separated on a 1% agarose gel (run in 1× TBE buffer). The DNA was stained with SYBR-Green (1:50,000, in 1× TBE) and the gels imaged on a Typhoon FLA 7000 fluorescence imaging system (Cytiva). The band of the undigested supercoiled plasmid was quantified with ImageJ by comparing the intensity of selected bands to a standard band on the same gel. If indicated, relative values were normalized to the amount of protein extracted in parallel to the nucleotides from the same cultures. Data were analysed and visualized using GraphPad Prism 9.5.1 (Dotmatics).

### Reporting summary

Further information on research design is available in the Nature Portfolio Reporting Summary linked to this article.

## Data availability

Supplementary Information is available for this paper. Raw data are provided in Supplementary Tables 1–6. Mass spectrometry data are available via figshare at https://doi.org/10.6084/m9.figshare.25341769 (ref. 48). The *Bacillus subtilis* complete proteome dataset (UniProt Proteome ID UP000001570) is available at https://www.uniprot.org/. Source data are provided with this paper.

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

## Acknowledgements

This work was funded by a European Research Council Advanced Grant (grant number 101018608) to M.F.W. L.K. was funded by an EMBO postdoctoral fellowship (grant number ALTF 234-2022). L.G.-M. was funded by the UKRI Biotechnology and Biological Sciences Research Council (BBSRC) (grant number BB/T00875x/1). We thank J. Stülke (University of Göttingen) for generously providing *Bacillus subtilis* strains, plasmids and antibodies α-HPr and α-Rny; S. Synowsky and S. McMahon for the MS analysis; K. Kohm and V. Lutz for the initial isolation of the new phage isolates; and F. Commichau for providing training in phage biology. For the purpose of open access, the author has applied a CC BY public copyright licence to any accepted author manuscript version arising from this submission.

## Author contributions

L.K. and M.F.W. conceived and designed the study. L.K., L.G.-M. and S.G. performed the experiments. L.K., L.G.-M. and M.F.W. analysed the data. S.S. performed the MS analysis. R.H. isolated the phages. M.F.W. acquired funding. L.K. wrote the original draft of the manuscript. All authors reviewed and edited the manuscript.

## Competing interests

The authors declare no competing interests.

## Additional information

**Extended data** is available for this paper at https://doi.org/10.1038/s41564-024-01670-5.

**Correspondence and requests for materials** should be addressed to Larissa Krüger or Malcolm F. White.

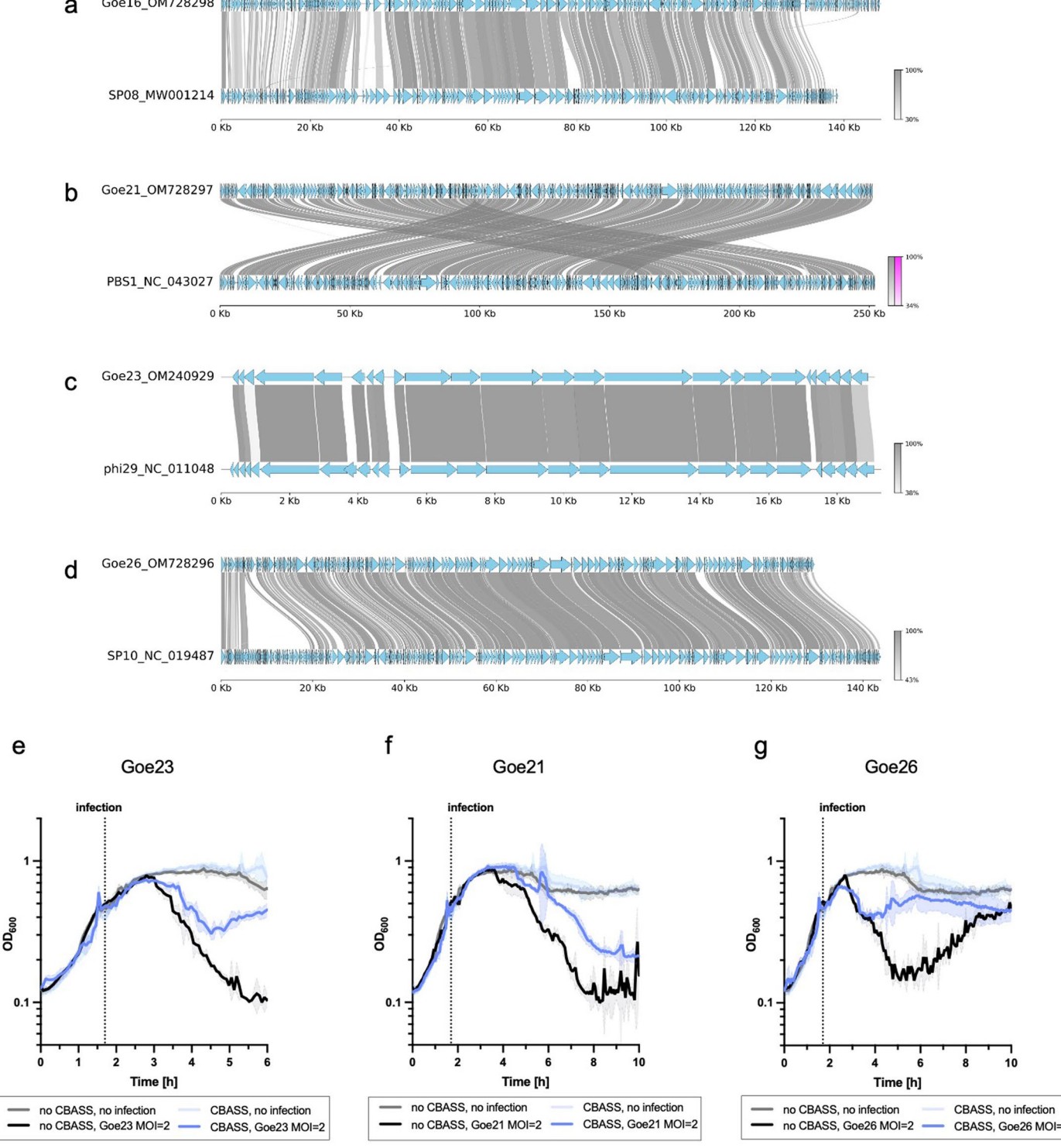

**Extended Data Fig. 1 | Genome alignments of four new *Bacillus subtilis* phages with their closest relative and infection assay in liquid medium.** The genomic sequences of the novel *Bacillus subtilis* phages Goe16 (**a**), Goe21 (**b**), Goe23 (**c**) and Goe26 (**d**) were compared to the genome sequence of their closest relative as revealed by Megablast. Genome sequences were aligned with MMseqs within the pyGenomeViz package. blue: CDS; grey: normal link; magenta: inverted link. Colour shades represent sequence identities (%). **e-f**, The CBASS operon of *Bacillus cereus* WPySW2 was integrated into *Bacillus subtilis* Δ6. The growth of wildtype (no CBASS, strain LK18) and cells expressing CBASS (strain LK06) was monitored until cultures reached mid-exponential growth phase. Cells were then infected with phage Goe23 (**e**), Goe21 (**f**), or Goe26 (**g**) to an MOI of 2. The experiment was conducted with two biologically independent replicates with two technical replicates each and data are presented as mean values ± SD.

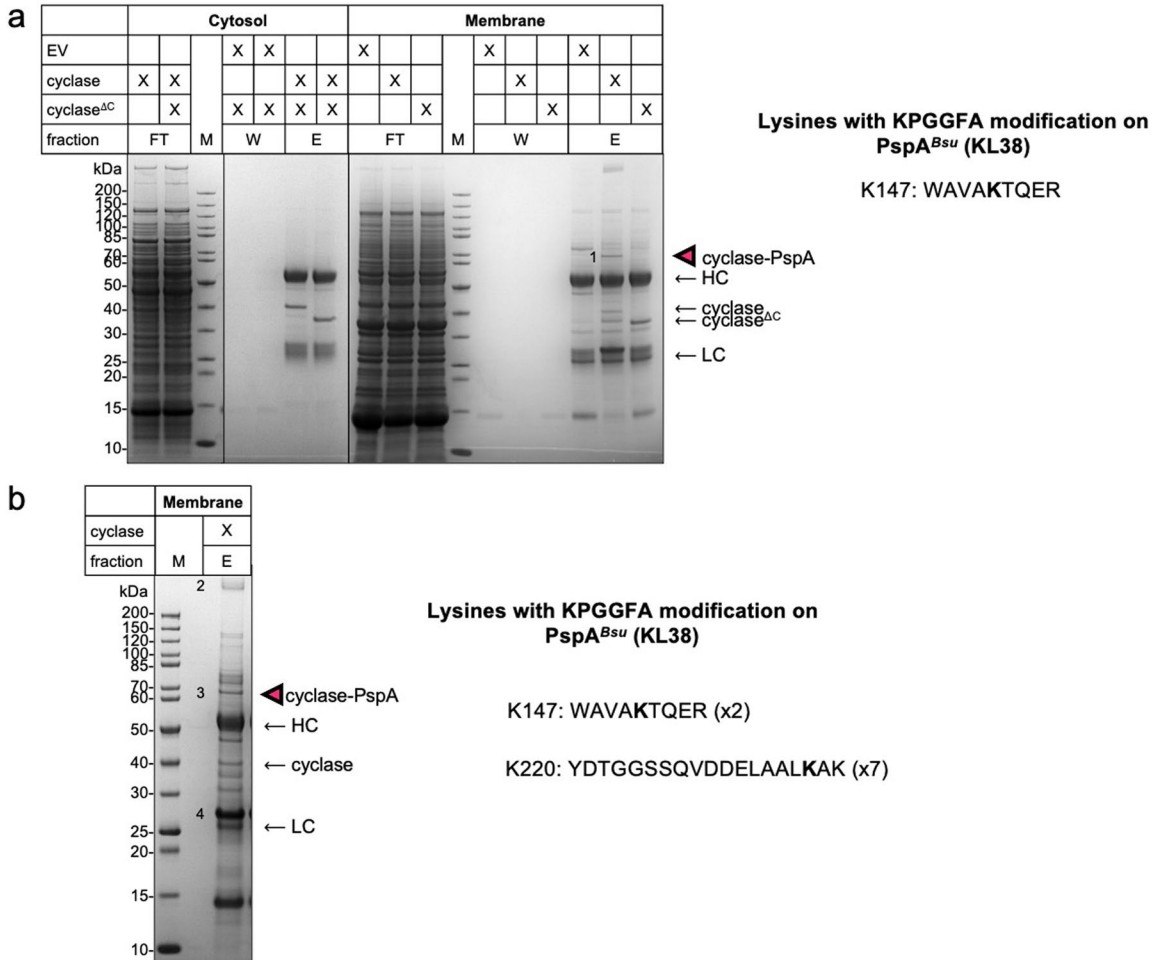

**Extended Data Fig. 2 | The cyclase is conjugated to the phage shock protein A *in vivo*. a**, Purification of the overexpressed cyclase (*Bce*) from *B. subtilis* cytosolic and membrane fractions using Protein A coupled Dynabeads and a cyclase specific antibody. Abbreviations: EV: empty vector; FT: flow-through; W: wash fraction; E: elution fraction; HC: heavy chain; LC: light chain. Band 1 was excised from the SDS gel, digested with trypsin and analyzed by MS.

The mass of the last C-terminal tryptic peptide of the cyclase (KPGGFA) was set as a variable modification in the peptide search. **b**, Replicate of a. Bands were analysed as before. Abbreviations: PSMs: peptide sequence matches. Excised gel bands numbered 1–4, results of MS analysis can be found in Supplementary Information.

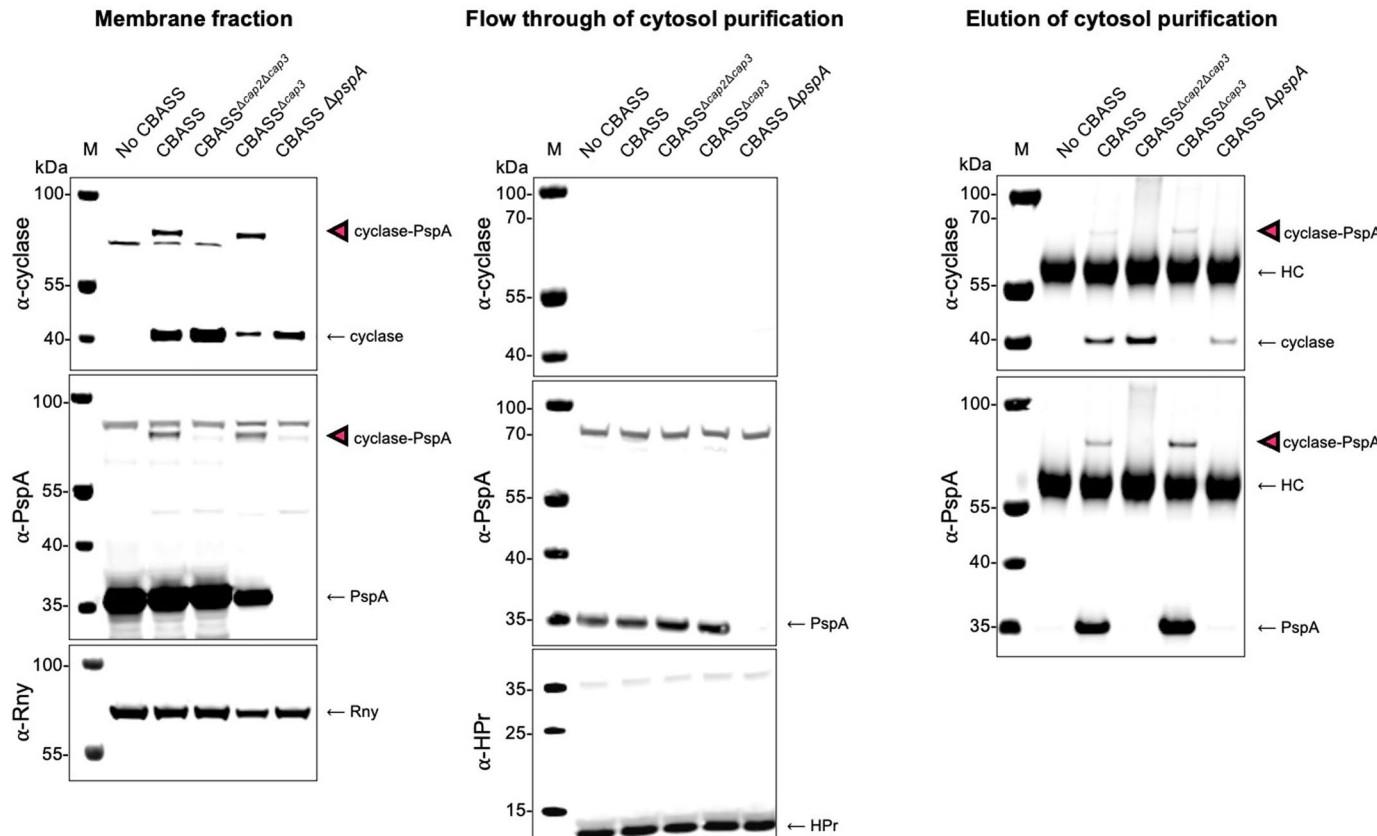

**Extended Data Fig. 3 | Cyclase conjugation is Cap2 dependent, reversible by Cap3, and PspA is the sole target.** *B. subtilis* wild type (no CBASS) and strains expressing genomic integrations of the *B. cereus* CBASS operon were grown and harvested at early exponential growth phase. Cell lysates were separated into cytosol and membranes. Cytosolic fractions were concentrated by purifying the cyclase using Protein A coupled beads and a cyclase specific antibody. Membrane fractions, the flow through fractions of the cytosol concentration (loading controls for purification), and the elution fractions were separated by SDS-PAGE and analysed by Western blot with a cyclase- or PspA-specific antibody. Antibodies raised against the RNase Y protein and the phospho-carrier protein HPr served as a loading control for the membrane fraction and cytosol purification, respectively. Abbreviations: HC: heavy chain. The experiment was conducted with two biological replicates and representative images from one replicate are shown.

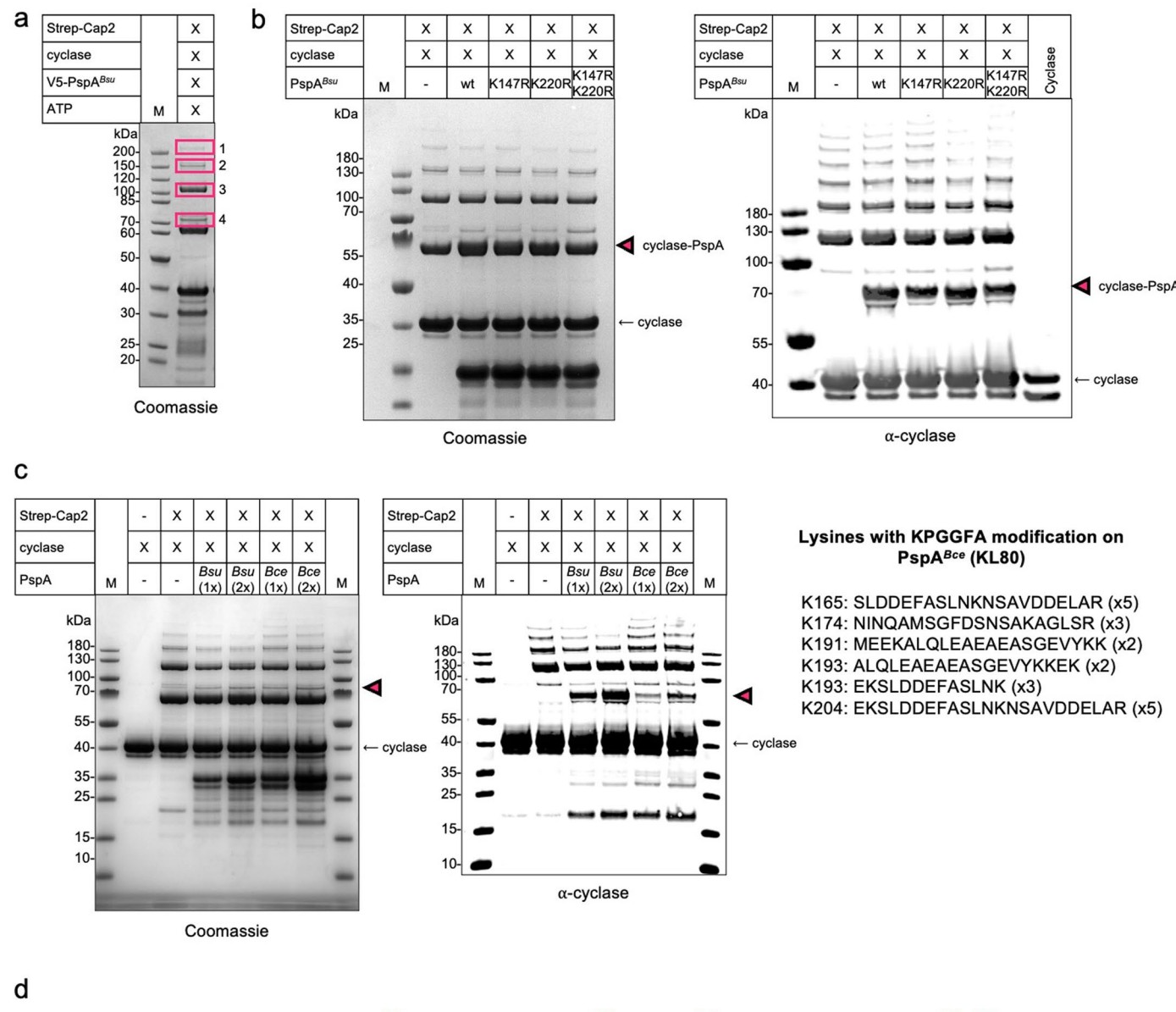

**Lysines with KPGGFA modification on PspA^Bce (KL80)**

K165: SLDDEFASLNKNSAVDDELAR (x5)
K174: NINQAMSGFDSNSAKAGLSR (x3)
K191: MEEKALQLEAEAEASGEVYKK (x2)
K193: ALQLEAEAEASGEVYKKEK (x2)
K193: EKSLDDEFASLNK (x3)
K204: EKSLDDEFASLNKNSAVDDELAR (x5)

**Extended Data Fig. 4 | See next page for caption.**

**Extended Data Fig. 4 | Cap2 conjugates the cyclase to PspA. a**, *In vitro* conjugation of the cyclase to PspA. Reaction mixture contained 5 μM Strep-Cap2, 5 μM cyclase and 10 μM wild type PspA$^{Bsu}$ (*Bacillus subtilis*), 10 mM MgCl$_2$, 1 mM ATP, 100 mM Tris-HCl pH 8, 150 mM NaCl. Samples were incubated for 30 min at 25 °C, separated by SDS-PAGE. Bands (1-4) were excised from the SDS gel, digested with trypsin and analyzed by MS. Results of the MS analyis can be found in Supplementary Information. The experiment was conducted with three replicates and a representative gel is shown. **b**, *In vitro* conjugation of the cyclase to PspA$^{Bsu}$ wild type and variant proteins. Reaction mixture contained 5 μM Strep-Cap2, 5 μM cyclase and 10 μM wild type PspA$^{Bsu}$ or variants as indicated, 10 mM MgCl$_2$, 1 mM ATP, 100 mM Tris-HCl pH 8, 150 mM NaCl. Samples were incubated for 30 min at 25 °C, separated by SDS-PAGE (top) and analysed by Western blot (bottom) with a cyclase-specific antibody. Western blot shows the full blot image

of the same blot from Fig. 3b. Strep-Cap2: 67.8 kDa; cyclase: 37.8 kDa; PspA: 25.1 kDa. **c**, *In vitro* conjugation assay of the cyclase as described for **a** with the two PspA homologs from *B. subtilis* (*Bsu*) and *B. cereus* (*Bce*). Reaction mixture contained 5 μM Strep-Cap2, 5 μM cyclase and 5 μM or 10 μM of the respective PspA homolog. Samples were analysed by SDS-PAGE and Western blot with a cyclase-specific antibody. The cyclase-PspA$^{Bce}$ conjugate band (2x) was excised from the SDS gel, digested with trypsin and analyzed by MS. The experiment was conducted with three replicates and a representative gel is shown. **d**, Sequence alignment of PspA homologs from *B. subtilis* 168, *Bacillus cereus* WPySW2, and *Escherichia coli* K12 was created with Geneious 2022.2.1 and visualized with Jalview 2.11.2.5. Lysine residues are highlighted in blue. The conjugation target residues of *B. subtilis* PspA and *B. cereus* PspA are marked with magenta circles.

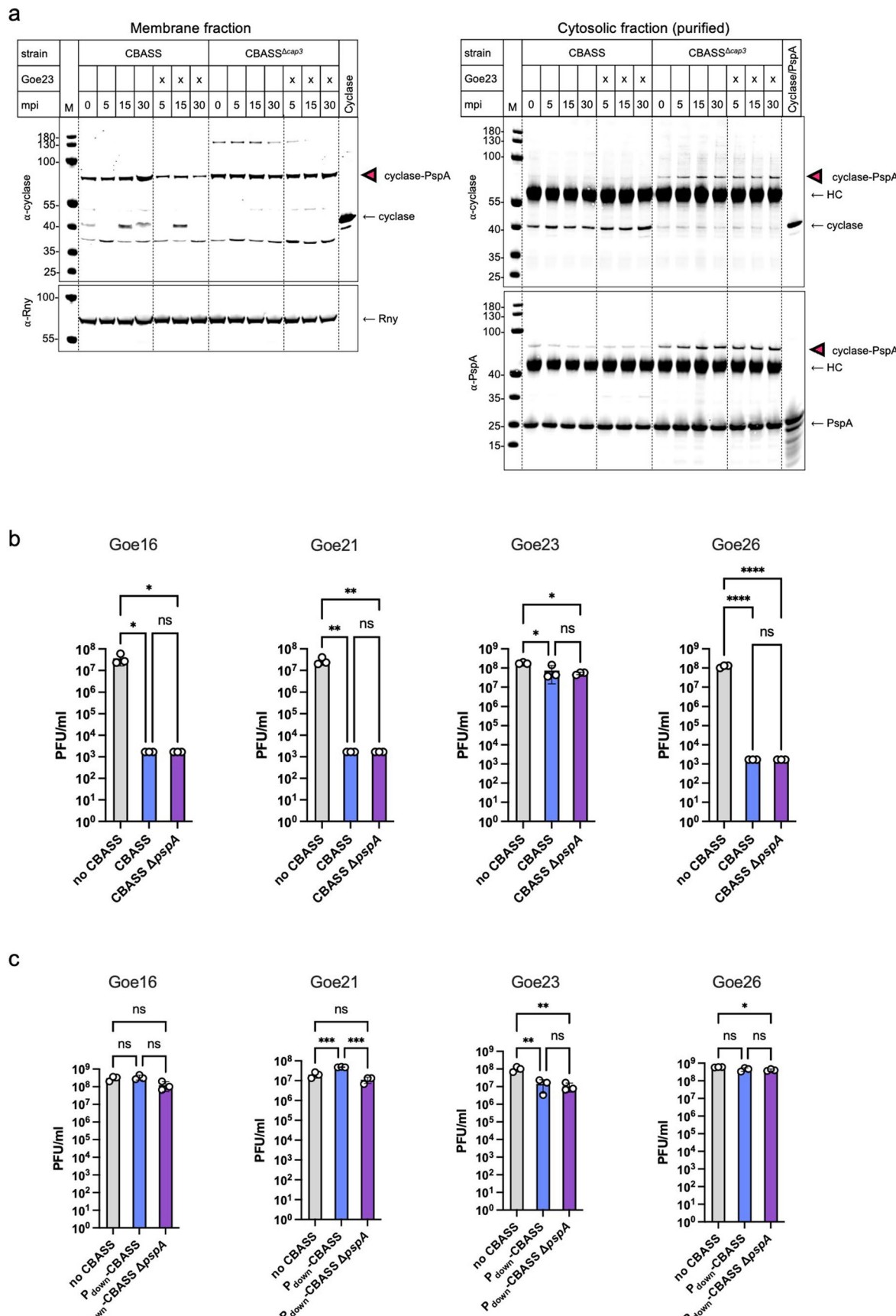

**Extended Data Fig. 5 | See next page for caption.**

**Extended Data Fig. 5 | Effect of conjugation and *pspA* deletion on phage immunity. a**, Strains harbouring genomic integration of the CBASS operon (strains LK06 (CBASS), LK20 (CBASS$^{\Delta cap3}$)) were grown until exponential growth phase and, if indicated, infected with phage Goe23 to an MOI of 2. Cultures were grown for an additional 5, 15, or 30 min before harvesting. Cell lysates were separated into cytosolic and membrane fractions. Fractions were adjusted in protein content, cytosolic fraction purified using Protein A-coupled beads and a cyclase specific antibody, and separated by SDS-PAGE, and the conjugate and monomeric cyclase and PspA$^{Bsu}$ were detected by Western blot with a cyclase-/PspA$^{Bsu}$-specific antibody. A specific antibody raised against the RNase Y protein/PspA served as a loading control. Abbreviations: mpi: minutes post infection. The blot of the membrane fraction is the uncut blot shown in Fig. 4a. **b**, Infection assay of *B. subtilis* strains containing CBASS components as indicated. Plaque forming units (PFU/ml) of plaque assays were calculated from three biological replicates. Strains used: LK06 (CBASS), LK18 (no CBASS), and LK51 (CBASS Δ*pspA*). No CBASS and CBASS data has been replotted from Fig. 1c. The plaque assays were performed with three biological replicates and data are presented as mean values ± SD. Statistical analysis was performed using a one-way ANOVA, followed by Tukey's multiple comparisons test (P values listed as appearing on the graph left to right, Goe16: *P = 0.0446, ns > 0.9999; Goe21: **P = 0.0037, ns > 0.9999; Goe23: *P = 0.03, *P = 0.0129, ns > 0.7481; Goe26: ****P < 0.0001, ns > 0.9999). **c**, Strains used: LK18 (no CBASS), LK63 (P$_{down}$-CBASS). And LK77 (P$_{down}$-CBASS Δ*pspA*). Abbreviations: P$_{down}$: Promoter with mutation in −10 region leading to lower expression of downstream gene (Stanley et al.[22]). The plaque assays were performed with three biological replicates and data are presented as mean values ± SD. Statistical analysis was performed using a one-way ANOVA, followed by Tukey's multiple comparisons test (P values listed as appearing on the graph left to right, Goe16: ns = 0.82, ns = 0.117, ns = 0.0518; Goe21: ***P = 0.0008, ns = 0.1193, ***P = 0.002; Goe23: **P = 0.005, **P = 0.004, ns = 0.967; Goe26: ns = 0.0837, *P = 0.0394, ns = 0.8222). Detailed information on the genotypes of all strains can be found in Supplementary Table 2.

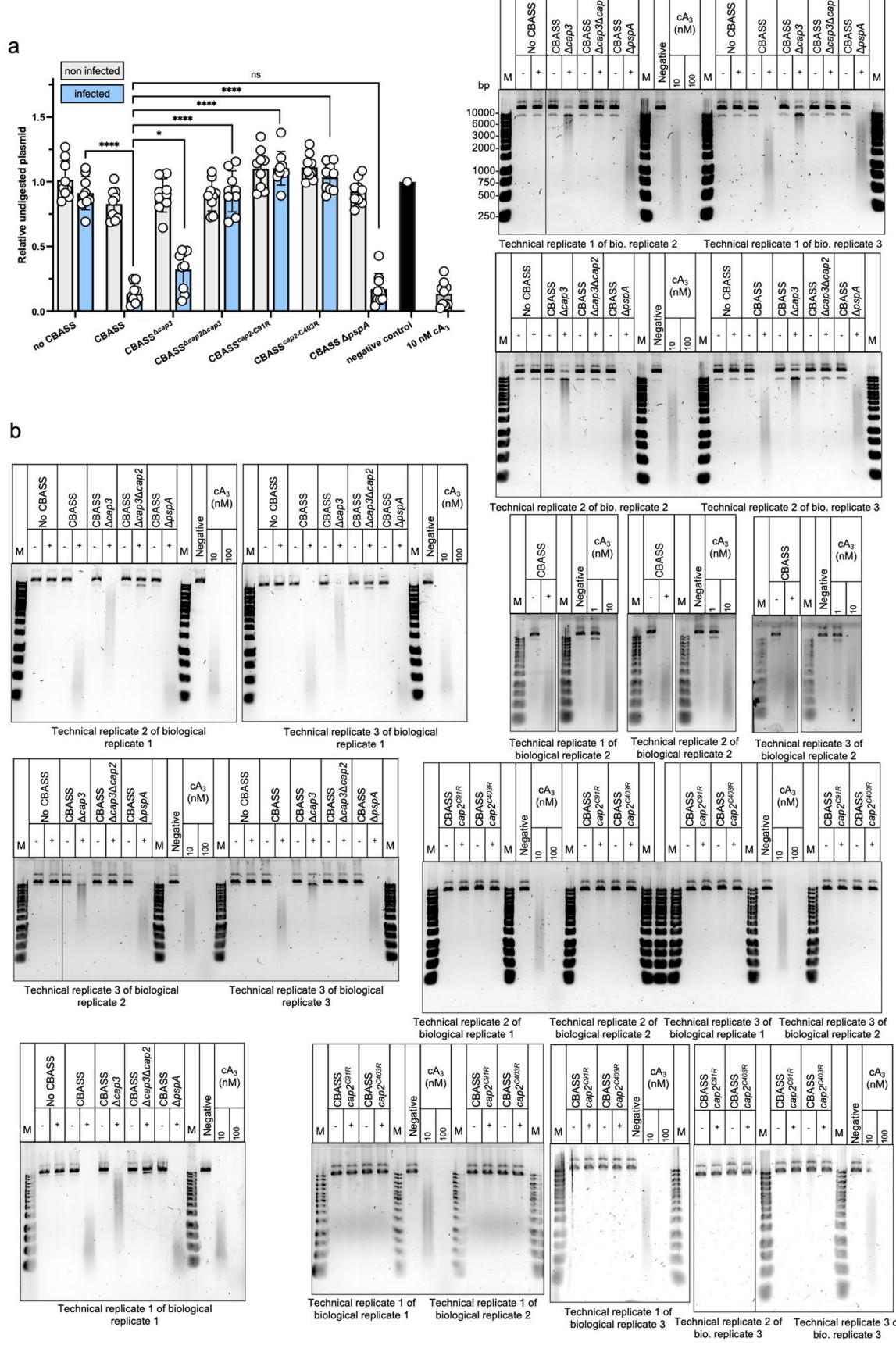

**Extended Data Fig. 6 | See next page for caption.**

**Extended Data Fig. 6 | Production of CBASS cyclic nucleotides infected cells.**
**a**, *B. subtilis* wild type (no CBASS) and cells carrying CBASS versions as indicated were grown up to early exponential growth phase. Cultures were then either infected with the phage Goe23 (+) to an MOI of 2 or not infected (−) and incubated for 30 min and harvested. Nucleotides were extracted and tested for their ability to activate the CBASS effector Nuc-SAVED in a plasmid cleavage assay. Synthetic $cA_3$ served as a control. Plasmid cleavage was assessed using a 1 % agarose gel.

Strains used: LK06 (CBASS), LK10 (CBASS$^{\Delta cap2 \Delta cap3}$), LK18 (no CBASS), LK20 (CBASS$^{\Delta cap3}$), LK51 (CBASS $\Delta pspA$), LK64 (CBASS$^{cap2\text{-}C91R}$), LK65 (CBASS$^{cap2\text{-}C403R}$). All experiments were conducted with three biological replicates with three technical replicates each. Data are presented as mean values ± SD. Statistical analysis was performed using a one-way ANOVA, followed by Tukey's multiple comparisons test (****$P < 0.0001$, *$P = 0.0477$, ns = 0.9972). **b**, Whole agarose gels from quantification shown in **a**.

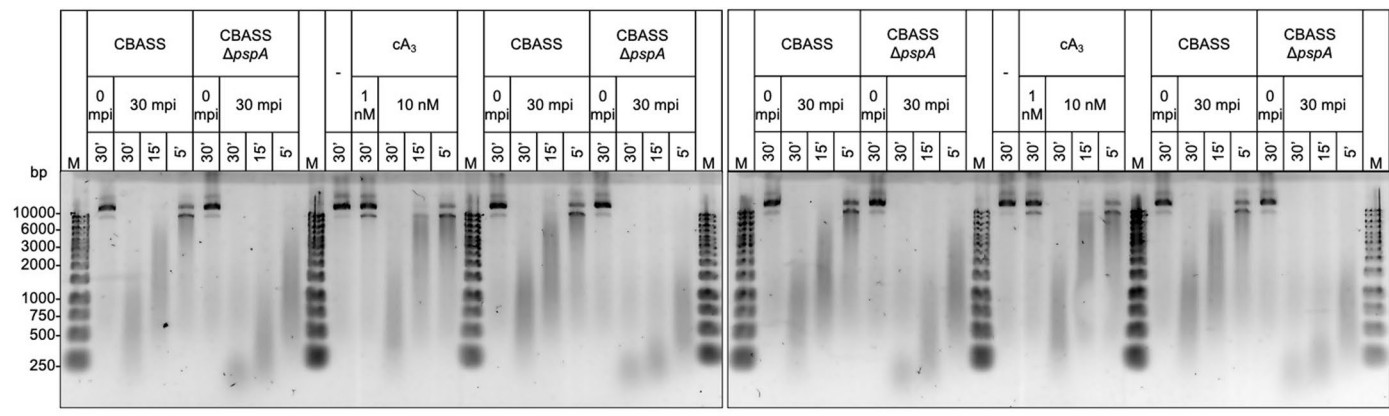

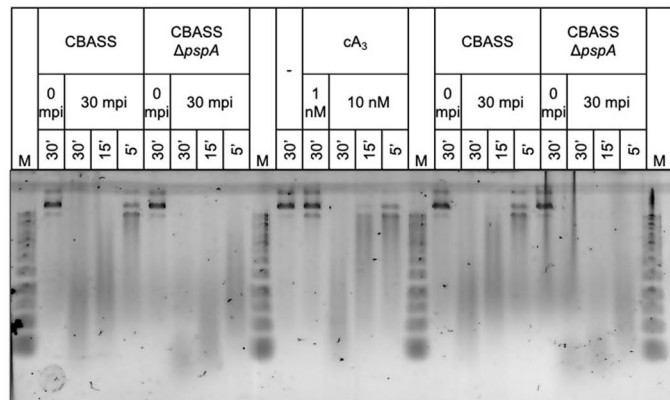

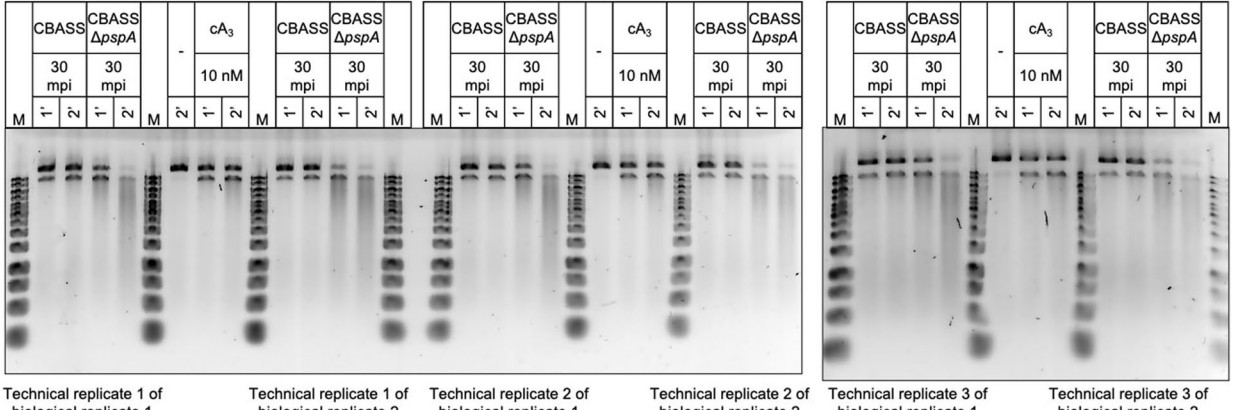

**Extended Data Fig. 7 | Increased second messenger production in the absence of PspA.** Nucleotides were extracted from *B. subtilis* cells carrying CBASS versions as indicated 30 min after infection with phage Goe23 (MOI = 2). Nucleotides were tested for their ability to activate the CBASS effector Nuc-SAVED in a time course experiment (1, 2, 5, 15, 30 min) using a plasmid cleavage assay. The cleavage activity of Nuc-SAVED after incubation with the nucleotide extracts was assessed using a 1% agarose gel. The experiment was conducted with three biological replicates with three technical replicates each. Strains used: LK06 (CBASS) and LK51 (CBASS Δ*pspA*). Whole agarose gels of quantification shown in Fig. 4d. Detailed information on the genotypes of all strains can be found in Supplementary Table 2. Abbreviation: mpi: minutes post infection.

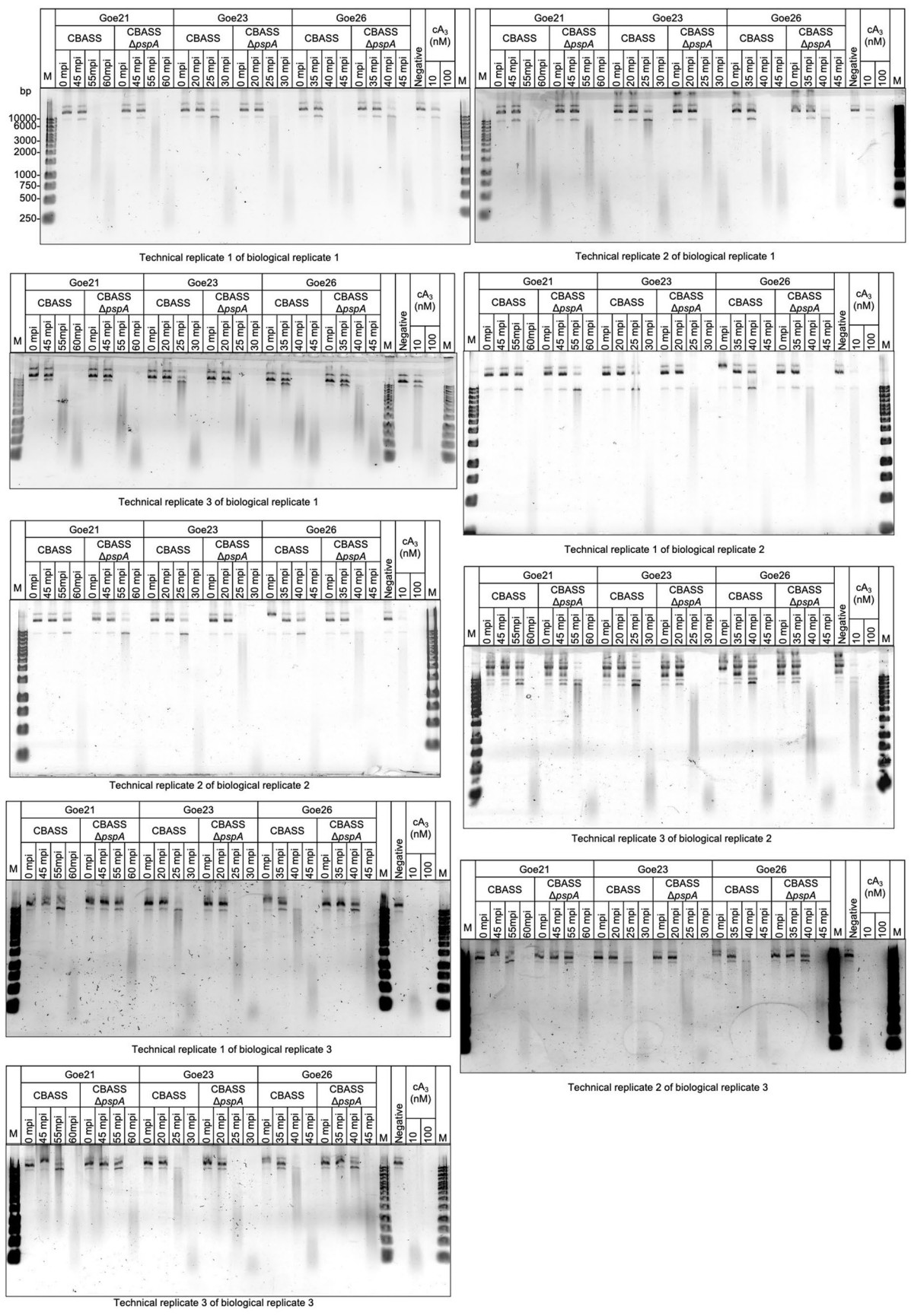

**Extended Data Fig. 8 | See next page for caption.**

**Extended Data Fig. 8 | Cyclase activation after infection with phage Goe21, Goe23, or Goe26.** Nucleotides were extracted from *B. subtilis* strains LK06 (CBASS) and LK51 (CBASS Δ*pspA*) infected with phage Goe21, Goe23, or Goe26 (MOI = 2) after the indicated time points (mpi: minutes post infection). Nucleotides were tested for their ability to activate the CBASS effector Nuc-SAVED using a plasmid cleavage assay. The cleavage activity of Nuc-SAVED after incubation with the nucleotide extracts was assessed using a 1% agarose gel. The experiment was conducted with three biological replicates with three technical replicates each. Whole agarose gels from quantification shown in Fig. 4e, f, g.

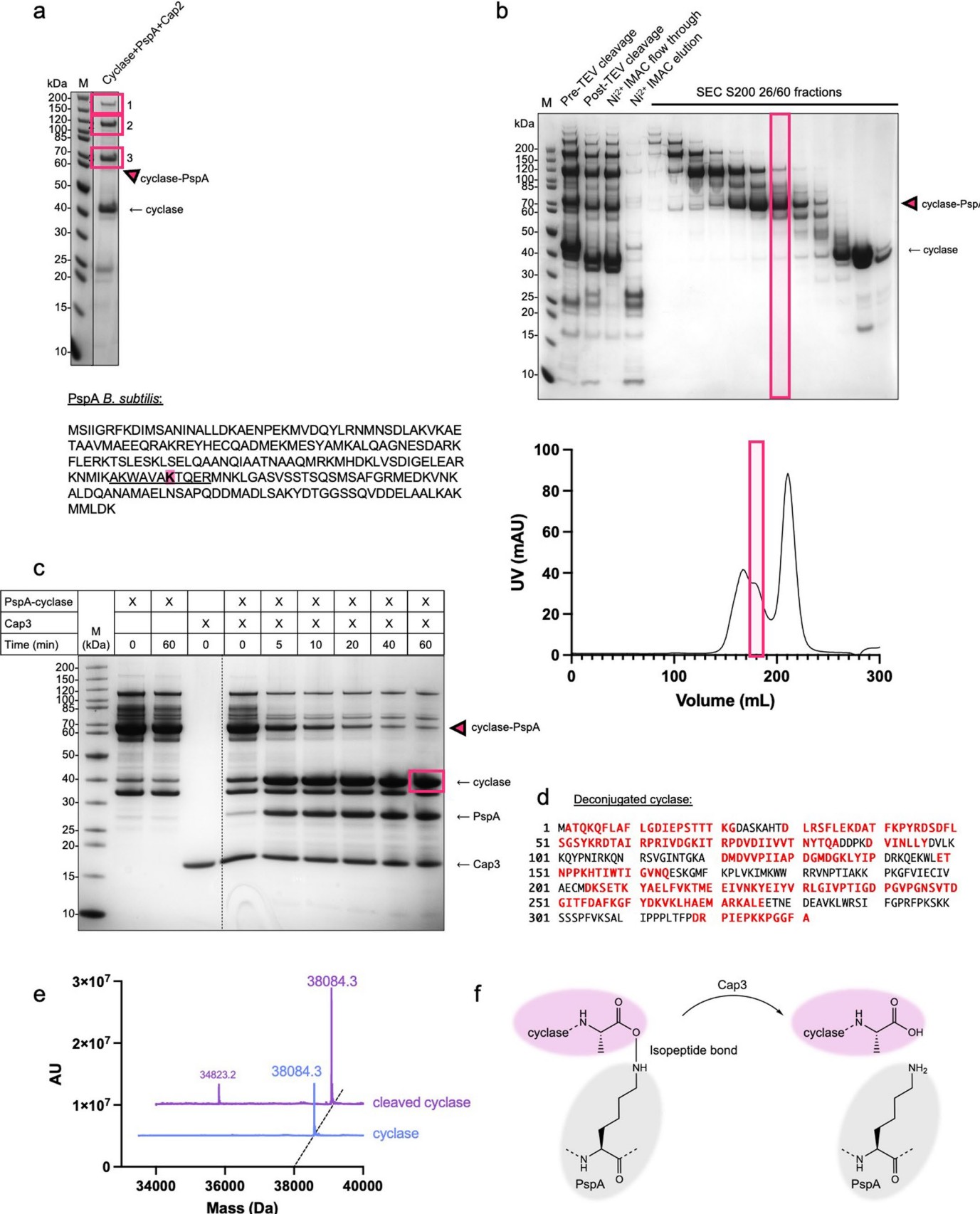

**a**

PspA *B. subtilis*:

MSIIGRFKDIMSANINALLDKAENPEKMVDQYLRNMNSDLAKVKAE
TAAVMAEEQRAKREYHECQADMEKMESYAMKALQAGNESDARK
FLERKTSLESKLSELQAANQIAATNAAQMRKMHDKLVSDIGELEAR
KNMIKAKWAVAKTQERMNKLGASVSSTSQSMSAFGRMEDKVNK
ALDQANAMAELNSAPQDDMADLSAKYDTGGSSQVDDELAALKAK
MMLDK

**d** Deconjugated cyclase:

```
  1 MATQKQFLAF LGDIEPSTTT KGDASKAHTD LRSFLEKDAT FKPYRDSDFL
 51 SGSYKRDTAI RPRIVDGKIT RPDVDIIVVT NYTQADDPKD VINLLYDVLK
101 KQYPNIRKQN RSVGINTGKA DMDVVPIIAP DGMDGKLYIP DRKQEKWLET
151 NPPKHTIWTI GVNQESKGMF KPLVKIMKWW RRVNPTIAKK PKGFVIECIV
201 AECMDKSETK YAELFVKTME EIVNKYEIYV RLGIVPTIGD PGVPGNSVTD
251 GITFDAFKGF YDKVKLHAEM ARKALEETNE DEAVKLWRSI FGPRFPKSKK
301 SSSPFVKSAL IPPPLTFPDR PIEPKKPGGF A
```

**Extended Data Fig. 9 | See next page for caption.**

**Extended Data Fig. 9 | Conjugation in the heterologous *E. coli* host and cleavage of the conjugate by Cap3. a**, The presence of cyclase and PspA peptides in the three conjugate bands was confirmed by MS and the tryptic peptide (WAVAKTQER/AKWAVAKTQER) containing the modified lysine residues (highlighted in magenta) is underlined. The experiment was conducted three times and a representative gel image is shown. **b**, Co-expression of the CBASS cyclase (*Bce*), Cap2 (*Bce*) and PspA (*Bsu*) in *E. coli* and purification of the cyclase-PspA conjugate by size exclusion chromatography. The experiment was conducted three times and a representative gel image is shown. **c**, the cyclase-PspA conjugate was incubated together with the Cap3 ortholog from *Cytobacillus*

*oceanisediminis* and the cleavage was followed by SDS-PAGE (see also Fig. 5b). **d**, the band of the deconjugated cyclase was excised from the SDS gel (magenta box in **c**), digested with the peptidase AspN and the peptides were analysed by MS. Peptides identified in the peptide search are shown in bold red. The overall protein sequence coverage was 61%. The presence of the C-terminal peptide including the C-terminal Ala residue involved in conjugation indicates that the cleavage of the cyclase-PspA conjugate by Cap3 leaves no scar on the cyclase. **e**, analysis of the size of the cyclase after cleavage. For this, the last sample (60 min) of the conjugate cleavage assay from **c** was analysed by intact MS. **f**, cleavage of the cyclase-PspA conjugate by Cap3 leads to unaltered cyclase monomers.

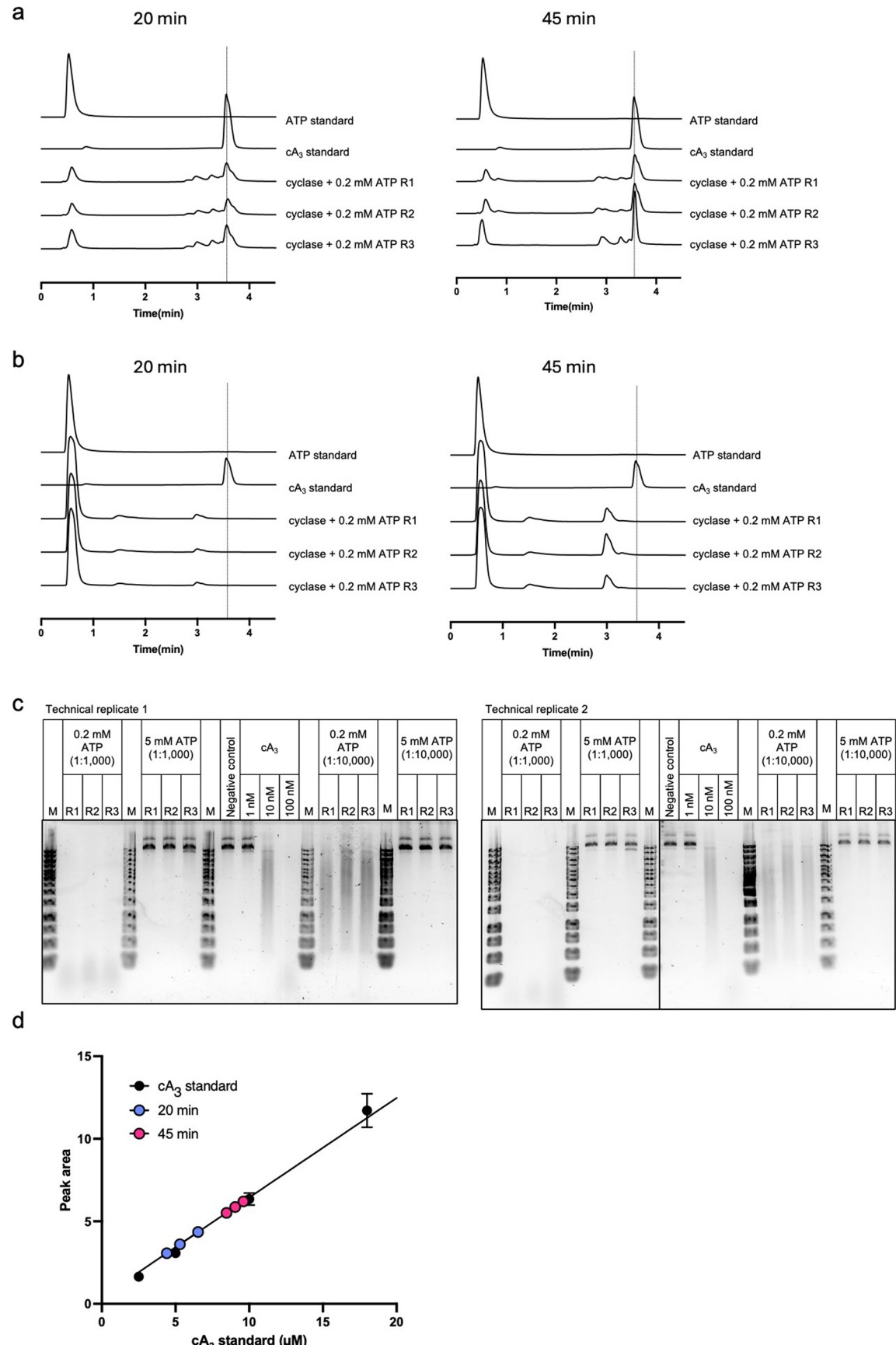

**Extended Data Fig. 10 | See next page for caption.**

**Extended Data Fig. 10 | Analysis of cyclase products. a**, reaction products from a cyclase reaction (50 μM cyclase, 0.2 mM ATP, 20 min/45 min incubation at 37 °C) were analysed by HPLC. The traces of all three replicates are shown. Synthetic ATP and $cA_3$ (cyclic triadenosine monophosphate) served as standard. **b**, same as **a** with the difference that the reaction mixture contained 5 mM ATP. **c**, The reaction samples from a and b (45 min) were tested for their ability to activate the CBASS effector Nuc-SAVED in a plasmid cleavage assay and plasmid cleavage assessed using a 1% agarose gel. The experiment was conducted with three biological replicates with two technical replicates each **d**, a dilution series of synthetic $cA_3$ was analysed by HPLC and the peak area of the $cA_3$ peak used to create a standard curve. The amounts of $cA_3$ in the reaction products from **a** were calculated by fitting to the standard curve. The experiment was performed with three replicates and data are presented as mean values ± SD. Abbreviations: Replicate 1 (R1), 2 (R2), and 3 (R3).

Larissa Krüger

# Reporting Summary

## Statistics

For all statistical analyses, confirm that the following items are present in the figure legend, table legend, main text, or Methods section.

| n/a | Confirmed | |
|---|---|---|
| ☐ | ☒ | The exact sample size (*n*) for each experimental group/condition, given as a discrete number and unit of measurement |
| ☐ | ☒ | A statement on whether measurements were taken from distinct samples or whether the same sample was measured repeatedly |
| ☐ | ☒ | The statistical test(s) used AND whether they are one- or two-sided<br>*Only common tests should be described solely by name; describe more complex techniques in the Methods section.* |
| ☒ | ☐ | A description of all covariates tested |
| ☒ | ☐ | A description of any assumptions or corrections, such as tests of normality and adjustment for multiple comparisons |
| ☐ | ☒ | A full description of the statistical parameters including central tendency (e.g. means) or other basic estimates (e.g. regression coefficient) AND variation (e.g. standard deviation) or associated estimates of uncertainty (e.g. confidence intervals) |
| ☐ | ☒ | For null hypothesis testing, the test statistic (e.g. *F*, *t*, *r*) with confidence intervals, effect sizes, degrees of freedom and *P* value noted<br>*Give P values as exact values whenever suitable.* |
| ☒ | ☐ | For Bayesian analysis, information on the choice of priors and Markov chain Monte Carlo settings |
| ☒ | ☐ | For hierarchical and complex designs, identification of the appropriate level for tests and full reporting of outcomes |
| ☒ | ☐ | Estimates of effect sizes (e.g. Cohen's *d*, Pearson's *r*), indicating how they were calculated |

*Our web collection on statistics for biologists contains articles on many of the points above.*

## Software and code

Policy information about availability of computer code

| Data collection | Phosphorimage and fluorescence scan data were collected using GE Healthcare ImageQuant TL 7.0<br>HPLC data were collected using Chromeleon 6.8 Chromatography Data System software (ThermoFisher). Western blots were visualized using the Licor Odyssey CLx Image Studio 5.x software. |
|---|---|
| Data analysis | GraphPad Prism 9.5.1 was used for enzymatic data analysis and creation of graphs. Geneious Prime 2022.2.1 (Biomatters) was used for sequence analysis. ImageJ 1.53k was used for quantification of gel images |

For manuscripts utilizing custom algorithms or software that are central to the research but not yet described in published literature, software must be made available to editors and reviewers. We strongly encourage code deposition in a community repository (e.g. GitHub). See the Nature Portfolio guidelines for submitting code & software for further information.

## Data

Policy information about availability of data

All manuscripts must include a data availability statement. This statement should provide the following information, where applicable:
- Accession codes, unique identifiers, or web links for publicly available datasets
- A description of any restrictions on data availability
- For clinical datasets or third party data, please ensure that the statement adheres to our policy

Supplementary information is available for this paper. Raw data are provided in the supplementary tables and files. Mass spectrometry data are available on

# Research involving human participants, their data, or biological material

Policy information about studies with human participants or human data. See also policy information about sex, gender (identity/presentation), and sexual orientation and race, ethnicity and racism.

| | |
|---|---|
| Reporting on sex and gender | n/a |
| Reporting on race, ethnicity, or other socially relevant groupings | n/a |
| Population characteristics | n/a |
| Recruitment | n/a |
| Ethics oversight | n/a |

Note that full information on the approval of the study protocol must also be provided in the manuscript.

# Field-specific reporting

Please select the one below that is the best fit for your research. If you are not sure, read the appropriate sections before making your selection.

☒ Life sciences  ☐ Behavioural & social sciences  ☐ Ecological, evolutionary & environmental sciences

For a reference copy of the document with all sections, see nature.com/documents/nr-reporting-summary-flat.pdf

# Life sciences study design

All studies must disclose on these points even when the disclosure is negative.

| | |
|---|---|
| Sample size | technical triplicates and biological duplicates or triplicates (as stated) were obtained for all measurements. This represents the accepted norm for a biochemical study, allowing standard deviation and mean to be calculated. |
| Data exclusions | no data was excluded |
| Replication | Biological and technical replicates were carried out as indicated in the figure legends. |
| Randomization | Randomization was not applied to the small biochemical datasets studied, in keeping with established norms for this type of study. |
| Blinding | Blinding was not applied to the small biochemical datasets studied, in keeping with established norms for this type of study. |

# Reporting for specific materials, systems and methods

We require information from authors about some types of materials, experimental systems and methods used in many studies. Here, indicate whether each material, system or method listed is relevant to your study. If you are not sure if a list item applies to your research, read the appropriate section before selecting a response.

## Materials & experimental systems

| n/a | Involved in the study |
|---|---|
| ☐ | ☒ Antibodies |
| ☒ | ☐ Eukaryotic cell lines |
| ☒ | ☐ Palaeontology and archaeology |
| ☒ | ☐ Animals and other organisms |
| ☒ | ☐ Clinical data |
| ☒ | ☐ Dual use research of concern |
| ☒ | ☐ Plants |

## Methods

| n/a | Involved in the study |
|---|---|
| ☒ | ☐ ChIP-seq |
| ☒ | ☐ Flow cytometry |
| ☒ | ☐ MRI-based neuroimaging |

## Antibodies

| | |
|---|---|
| Antibodies used | anti-cyclase (Bacillus cereus), anti-PspA (Bacilus subtilis), anti-Rny (Bacillus subtilis), anti-HPr (Bacillus subtilis), IRDye® 800CW Donkey anti-Rabbit IgG Secondary Antibody (LI-COR Biosciences; P/N: 926-32213) |
| Validation | The specificity of the antibodies was tested in Western blots against the respective purified protein and, if applicable, against cell lysate of Bacillus subtilis. |

