## [Peer Review File · Nature Microbiology]

Peer Review Information

Journal: Nature Microbiology

Manuscript Title: Reversible conjugation of a CBASS nucleotide cyclase regulates bacterial immune response to phage infection

Corresponding author name(s): Professor Malcolm White

Reviewer Comments & Decisions:

Decision Letter, initial version:

Message: 18th September 2023

Dear Professor White,

Thank you for your patience while your manuscript "Reversible conjugation of a CBASS nucleotide cyclase regulates immune response to phage infection" was under peer-review at Nature Microbiology. It has now been seen by 3 referees with broad phage defense system expertise, whose comments you will find at the end of this email. Although they find your work of some potential interest, they have raised a number of concerns that will need to be addressed before we can consider publication of the work in Nature Microbiology.

In particular, you will see that Reviewers #1 and #2 request a number of additional experiments which should be straightforward to address and would strengthen the manuscript. From an editorial perspective, the most important consideration is that your revised manuscript indicates that this is a more general phenomenon across phage, rather than just with Goe23. Finally, Reviewer #3 requested that the manuscript be revised with an eye to our more general microbiology audience, which we also agree would strengthen the paper.

Should further experimental data allow you to address these criticisms, we would be happy to look at a revised manuscript.

2Please include a data availability statement as a separate section after Methods but before references, under the heading "Data Availability". This section should inform readers about the availability of the data used to support the conclusions of your study. This information includes accession codes to public repositories (data banks for protein, DNA or RNA sequences, microarray, proteomics data etc...), references to source data published alongside the paper, unique identifiers such as URLs to data repository entries, or data set DOIs, and any other statement about data availability. At a minimum, you should include the following statement: "The data that support the findings of this study are available from the corresponding author upon request", mentioning any restrictions on availability. If DOIs are provided, we also strongly encourage including these in the Reference list (authors, title, publisher (repository name), identifier, year). For more guidance on how to write this section please see: <http://www.nature.com/authors/policies/data/data-availability-statements-data-citations.pdf>

* If you have not done so already we suggest that you begin to revise your manuscript so that it conforms to our Article format instructions at <http://www.nature.com/nmicrobiol/info/final-submission>. Refer also to any guidelines provided in this letter.

When submitting the revised version of your manuscript, please pay close attention to our [href="https://www.nature.com/nature-portfolio/editorial-policies/image-integrity">Digital Image Integrity Guidelines](https://www.nature.com/nature-portfolio/editorial-policies/image-integrity). and to the following points below:

Note: This url links to your confidential homepage and associated information about manuscripts you may have submitted or be reviewing for us. If you wish to forward this e-mail to co-authors, please delete this link to your homepage first.

Nature Microbiology is committed to improving transparency in authorship. As part of our efforts in this direction, we are now requesting that all authors identified as 'corresponding author' on published papers create and link their Open Researcher and Contributor Identifier (ORCID) with their account on the Manuscript Tracking System (MTS), prior to acceptance. This applies to primary research papers only. ORCID helps the scientific community achieve unambiguous attribution of all scholarly contributions. You can create and link your ORCID from the home page of the MTS by clicking on 'Modify my Springer Nature account'. For more information please visit please visit www.springernature.com/orcid.

If you wish to submit a suitably revised manuscript we would hope to receive it within 6 months. If you cannot send it within this time, please let us know. We will be happy to consider your revision, even if a similar study has been accepted for publication at Nature Microbiology or published elsewhere (up to a maximum of 6 months).

Yours sincerely,

Reviewer Comments:

Reviewer #1 (Remarks to the Author):

In this study Krüger et al. describe the reversible conjugation of a CD-NTase (cyclase) to the ubiquitous phage shock protein (PspA) and propose a model by which this conjugation plays a role in the sequestration of the cyclase to limit spurious activation of the Type II CBASS system in the absence of phage infection. They first show the genetic and catalytic requirements of the *B. cereus* CBASS for protection against phage predation in the heterologous *B. subtilis* host and further show the unexpected finding that this cyclase is conjugated specifically to the *B. subtilis* PspA protein in a Cap2 dependent manner. The Cap3 enzyme is also shown to scarlessly proteolyze the cyclase-PspA conjugate in vitro and in vivo, and this activity occurs late during infection with phage Groe23. The cyclase in vitro appears to be in a state of reduced activity, regardless of its conjugated state, in vitro and inactive in vivo in the absence of phage. This suggests that a critical condition to activate the cyclase during infection remains to be discovered, and the authors suggest based on in vitro cyclase assays this could be related to the abundance of intracellular nucleotides (e.g., ATP). Heroic efforts were made to demonstrate the utility of the PspA-cyclase conjugate in vivo, but the physiological role of this interaction remains enigmatic, with no

3conclusive evidence demonstrating this interaction between the host PspA and the CBASS cyclase is necessary for defense against phage by the Type-II CBASS. The expansion of these findings to include additional phage infection experiments (e.g., Groe26) validating the delayed activation hypothesis, demonstration of cyclase conjugation to diverse PspA substrates (not only *B. subtilis* PspA), as well as testing the CBASS antiphage activity under control of its native promoter (*B. subtilis* host is fine) would greatly improve the significance of these findings.

I want to commend the authors for their hard work, and I implore them to please read the following comments with the knowledge that they are meant to be helpful and written with a voice of curiosity and encouragement. There is no malice intended by any language that follows and in no way are they meant to be combative, rude, or demeaning. It is likely that some of these ideas and observations presented may reflect my own misinterpretation of data or text, despite my best efforts to be thorough. Please know that I respect your time, effort, and diligence in preparing this manuscript and thank you for reporting these interesting results.

Primary Comments:

The characteristics of both plaquing and planktonic growth of *B. subtilis* Δ6 + CBASS challenged with Goe23 are dramatically different from the other phages tested (Figs. 1B & 1C, Ext Data Fig 2). The Goe23 plaquing phenotype and the crash in OD (~3-5h) could be indications of slow activation of CBASS that is specific to this phage, as these are not seen in similar challenges with the three other phage. Because the delayed time to activation (~30 min) of this CBASS is an important conclusion of this study, it is important to demonstrate this is not a phenomenon specific to Goe23 infection alone (Figs. 4b,d,e and Ext. Figs. 6 and 7) (i.e., is delayed Cap3 deconjugation/activation of the cyclase only happening with Goe23). Additional temporal experiments (Similar to those performed in Ext. Figs. 6 and 7) should be performed using at least one additional phage. Perhaps Goe26 might be the most reasonable follow-up phage to use as both planktonic and plaquing data are currently presented, which could ease the burden of these experiments.

Lines 253-254, "The low sequence conservation between PspA homologs may explain why Cap2 requires flexibility in target recognition (Extended Data Fig. 5c)". The in vitro assay is a neat tool and I commend the authors for the monumental task of purifying all these components and establishing a condition that recapitulates some of the aspects observed in their in vivo conditions. However, the in vitro condition likely has some substantial artifactual activity where the Cap2 enzyme appears more promiscuous in its conjugation of the cyclase to alternative substrates/residues (e.g., high MW complexes of Cap2-cyclase in Fig. 3a not seen in vivo, Figs. 2a, 2d, Extended Data 3a, Extended Data Fig. 9a). However, in vivo results demonstrate cyclase-specific peptides are found only on residues K147 and K220 (Fig. 2b, Lines 176-177, Lines 332-333). To suggest that there is flexibility in target recognition by Cap2 in vivo activity to facilitate its ability to work with diverse PspA substrates (i.e., function in diverse bacterial hosts), the authors need to perform similar purifications and Western blots as those presented in Fig. 2d using strains containing the K147R and K220R variants of the *B. subtilis* PspA and demonstrate whether comparable alternative Lys residues on PspA are truly utilized as substrates in vivo. It is also important to assess the capacity of Cap2 to conjugate the cyclase to *B. cereus* PspA, which is presumably the "natural" substrate, and determine the locations of preferred Lys residues

for cyclase conjugation.

In Fig. 4d, use of either of the Cap2 inactive Cys->Arg variants (Figs. 1b,c, 5a) is important to demonstrate the utility of conjugation and would be more convincing than the double CBASS Δ Cap2 Δ Cap3 currently presented (i.e., the variants are a precision tool for addressing the role of conjugation while the double mutant is a hammer). If it is impractical to repeat these experiments with the cap2 variants, it is important to justify the use of the double knockout mutant in lieu of these superior strains (this is also applicable to Fig. 2d, but more critical to address in Fig. 4d where cyclase activity is explored in vivo).

In conjunction with the previous comments, would the authors please double check that in generating the double cap2 cap3 deletion strain (e.g., Fig. 4d) an unintended catalytically inactive chimeric cyclase has not been created (i.e., the native cyclase stop codon is preserved) and the Nuc-SAVED effector is still likely to be functional (i.e., start codon is preserved, a putative ribosome binding site exists, and transcription of the effector is unlikely to be interrupted.. basically check for any obvious indications of polar effects on the effector). There is overlap between the final nucleotide of the cyclase stop codon and the first nucleotide of the cap2 start codon and the Nuc-SAVED effector overlaps with the last four nucleotides of cap3. Generation of unintended mutations in either the cyclase and/or Nuc-SAVED could have significant consequences for the interpretation of cyclase activity and phage challenges assays reported anywhere the CBASS Δ Cap2 Δ Cap3 strain is used.

Lines 317-320, there is discussion that the lack of the native host context may be responsible for the absence of a clear phenotype for the PspA-cyclase conjugation. The use of the *B. cereus* CBASS "...expressed under a strong, constitutive *B. subtilis* promoter (PdegQ)..." (Lines 86 – 87) might be masking a PspA phenotype. The discovery of the specific PspA-cyclase conjugate is a remarkable finding, demonstration of its relevance to the initiation and deployment of the CBASS in vivo is important to bolster the significance of this discovery. I would strongly suggest the authors generate two *B. subtilis* strains (pspA+ and pspA-) where the CBASS is regulated by the native promoter from *B. cereus*, rather than PdegQ. Evaluating the antiphage activity of these two strains against the four Groe phage in plaquing assay and planktonic challenges is likely the best chance to reveal a cyclase-PspA phenotype that would leave no doubt as to its relevance to bacterial immunity.

Secondary Comments:

Line 150, previously in Fig. 1 the cyclase single amino acid variant A331E alone lost all ability to prevent phage invasion in the population (presumably no longer a Cap2 substrate), I was surprised to see that this variant was never used again in lieu of the C-terminal-truncated cyclase in later experiments (e.g., Fig. 2). Was there a notable reason for using this more substantially mutated gene as opposed to the single point mutation? Substitution of the truncated C-terminal cyclase for the single A331E variant requires demonstration that the response to phage challenge is the same between the two cyclase mutants (e.g., Fig. 1, does the truncated cyclase also fail to prevent phage infection).

Lines 215-216, It is hard to evaluate if the Westerns in Extended Data Fig. 4 show that

5more cyclase is associated with the membrane fraction compared to the cytosolic fraction without a reference protein sample for relative intensity comparisons within and across blots (e.g., purified cyclase of a known concentration run alongside the two fractions to compare intensities)?

Lines 323-328, please comment on how similar or divergent the Bce Cap2 is from the *V. cholerae*, *E. coli*, and *P. aeruginosa* enzymes previously tested. It seems that the Bce Cap2 studied here may have a unique mechanism to interact with PspA that is not present in these other Cap2 orthologs. Perhaps the utilization of PspA as a substrate could be predicted in other Cap2 enzymes from such a comparison?

Line 326-330, Does the heterologous *E. coli* host not encode a native *pspA*, is the abundance of this native PspA too low to detect HMW cyclase conjugates, or is the native *E. coli* PspA an inferior Cap2 substrate? Also, now that the experiments are being performed in a new host, please explicitly state which *pspA* allele is being over-expressed in *E. coli*.

Line 422-424, "...to prevent activation of CBASS in the absence of phage ... CBASS uses its protein modification machinery to conjugate the cyclase to the highly conserved PspA protein..." I am unconvinced that conjugation of the cyclase to PspA is preventing spurious activation of this CBASS in the absence of phage. It appears the cyclase is in an inactive form in the absence of phage in a manner independent of its post-translational conjugation to PspA. This is evident in Fig. 2a where there is over-expression of the cyclase and constitutive expression of the Nuc-SAVED effector (i.e., massive amounts of un-conjugated cyclase and effector, but no killing). Additionally, the inactivity or limited activity of purified cyclase in Fig. 5f (conjugated and unconjugated) also suggests the cyclase is primarily activated by an unknown factor (as stated in Line 412). The presence or absence of PspA also does not appear to influence the ability of CBASS to effectively limit phage predation (Fig. 4c). Therefore, based on the data in Figs. 4d,e and that mentioned above, it would be more appropriate to say the conjugation of the cyclase to PspA could be a means to slow cyclase activation during infection (rather than prevent activation in the absence of phage).

Additional Comments / Observations

[These do not require responses; I only want to bring them to the authors' attention and are all marked with an *]

*In general, the absence of asterisks for relevant p-values appears to be a systemic oversight throughout the manuscript (e.g., Fig. 1C and Fig. 4), including Extended Data Figs.

*In general, consistency across all figures with the usage of white and magenta triangles for identification of monomeric proteins and cyclase-PspA conjugates would be helpful. That being said, I found the most useful application of triangles was in Fig. 3a where only the magenta triangle was common across blots and included the name cyclase-PspA while the individual monomeric proteins were called out with arrows by name (rather than white triangles). This simplified and uniform identification scheme would greatly simplify interpretation of many of the blots and gels.

*Line 98, Fig. 1c – Error bars are difficult to see, and it appears a formatting error occurred

in the histogram for Goe21 CBASS Δ Cap3. Also check universally that sample labels are formatted inconsistently across similar experiments (extra spaces in names, capitalized gene names, font size, etc.) (e.g., Figs. 1C & 4D).

*Line 89, italicize the prophage-like element "skin".

*Lines 125-126, It would be more correct to adjust language from "induced point mutations in relevant residues", to something like, "introduced point mutations to produce catalytically inactive variants". Keeping genes as things to mutate and proteins as variants.

*Lines 126-127, rework the sentence beginning with, "The absence of cap2 and cap3..." as it is seemingly contradictory to the data and statements which immediately follow re: cap3.

Line 180, Fig. 2a it is not immediately apparent what the five "" refer to on the gel (analyzed by MS?). Additionally, the gel is very small (e.g., MWs are overwritten on one another) and increasing the size of this key piece of evidence would be helpful. In Fig. 2c, consider identifying the HC as done for Extended Data Fig. 4.

Line 240, missing description of significance ascribed to a single "" shown in Figure 3c.

*Lines 262-264, this may be semantics but to my eye, Groe23 appears to induce the WORST CBASS response of the four phages tested (Fig. 2b) leading to the least direct reduction in total plaques and delayed immunity in liquid medium.

*Line 287, please indicate that Groe23 was used to induce the cyclase activity in extracts used in Fig. 4d

*Line 289, is Fig. 4e the same data presented in Extended Data Fig. 8b? It is not clear how these two experiments are different, as Fig. 4e looks to display the mean values of individual data points in Extended Data Fig. 8b. Additionally, it appears that STD is missing in Fig. 4e.

*Lines 298-301, please provide a reference or justification for exploring cA3 (rather than another nucleotide moiety) as the presumed product of the cyclase and activator of Nuc-SAVED effector.

*Lines 307-309, it does not appear CBASS Δ Cap2 Δ Cap3 is used in all these figures referenced here, please double check (e.g., no Fig. 1d in the manuscript).

*Lines 315-316, "This supports the idea that the cyclase needs to be deconjugated prior to activation...". I suggest amending this language to be less strict for the requirement of deconjugation ("needs") for cyclase activation, as the Δ Cap3 strain still provides defense against phage (Fig. 1), the cyclase in this strain still synthesizes CA3 in a phage-dependent manner (Fig. 4d), and conjugated cyclase is later shown to activate Nuc-SAVED at the same rate as the full-length monomeric cyclase in vitro (Fig. 5f). Something along the lines of deconjugation "enhances in vivo cyclase activity".

*Line 349, reference to Fig. 5c is incorrect, should be reference to Extended Data Figs. 9c,d,e.

*Line 400-401, "This data suggests that the wild type Bce cyclase is inactive in this unconjugated form in vitro." Please remove the unconjugated form portion of the statement, as the data only suggests the wild type cyclase catalytic activity is very slow in vitro (Fig. 5e). The activity relative to the conjugation status of the cyclase is not tested until later (Fig. 5f) where it is shown to not play a significant role in the in vitro activity of the enzyme.

*Line 979, Extended Data Fig. 2 title does not mention Goe26 and as the figure compliments the solid agar derived plaquing assays in Fig. 1, I suggest exchanging "in vivo" for "in liquid medium", or an alternative title like, "CBASS protects planktonic *B. subtilis* Δ6 from Goe23 and Goe26."

*Line 982, references to growth curves in Extended Date Fig. 2 are incorrect.

*Line 986, Extended Data Fig. 3. In Extended Data Fig 3a, it is unclear what strain "EV" refers to – presumably this is the empty vector control? This legend is also missing descriptions of triangular markers. In extended Data Fig. 3b the PspA primary sequence is redundant (Fig. 2b), please consider removing it from this location as it provides little information relevant to the interpretation of data found in this figure.

*Line 997, Extended Data Fig. 4, consider adding the white triangles (or better yet, arrows with names) highlighting the monomeric forms of the indicated proteins for all Western blots.

*Line 1021, the Western blots in Extended Data Fig. 5b and in Fig. 3b appear to be the same, if so please mention this here. I noticed they looked very similar while trying to eyeball the intensities of the cyclase-PspA conjugates across PspA variants.

*Line 1026, for ease of interpretation please highlight the residues in the PspA alignment that correspond to *B. subtilis* PspA K147 and K220 in Extended Data Fig. 5c.

*Line 1033, please indicate what R1 and R2 are in the gel legends of Extended Data Fig. 6a mean (presumably these are two biological replicates).

*Line 1043, please indicate what +/- symbolize in Extended Data Fig. 7

*Line 1054, none of the panels in Extended Data Figure 8 are referenced in the body of the MS that I can find. Additionally, more information is required in the figure legend to understand the experiments depicted in the gels. (e.g., the use of R1 and R2 in the first-row of the gel legends, "30" in the second-row, and the third-row numerals 1, 2, 5, 15, and 30).

*Line 1066, Extended Data Figure 9a, please number the band boxes on the gel that were analyzed by MS.

*Line 1083, "...conjugate cleavage assay from b was analysed by intact mass

spectrometry." This statement should refer to Extended Data 9c rather than b.

Reviewer #2 (Remarks to the Author):

The work from Krüger et al. describes the regulation of the CBASS system from *B. cereus* in *B. subtilis*. After a clear and up-to-date introduction, the authors demonstrate with strong proofs that the nucleotide cyclase is conjugated by the protein Cap2 to the ubiquitous membrane protein PspA. This sequestration of the cyclase to the membrane prevents non-relevant activation of the CBASS system. The protein Cap3 has been shown to be necessary for the deconjugation of the Cylclase from PspA. These findings present a real advancement in the understanding of the regulation of phage defense system in bacteria, and especially for CBASS. This work presents a strong interest for every research community with interests in phage defense systems. The paper is well written and made with solid science. The conclusions drawn by the authors are clear, careful and rely on clear proof and serious studies. The M&M, as well as the legends are nicely detailed.

All experiments related to the main message of the articles are flawless and well made, but I noticed some points of the study or the discussion could be improved by some additional information/discussions.

Overall, I'll recommend the publication of this article after a round of minor revision and say that the authors accomplished an incredible work.

Here are the main questions/critics/suggestions that I have regarding the manuscript:

- Taxonomy of the four new phages L90-95 + Supp table 1: You did a great job to make taxonomic attributions to these new phages, unfortunately they are now obsolete (see <https://doi.org/10.1007/s00705-022-05694-2>). I'll strongly suggest you to update their names and taxonomic distribution (forget myoviridae et al families and go deeper in classification if you can (see <https://doi.org/10.3390/v9040070> for an up-to-date guide)). I'll also recommend adding to Table S1 a column saying if the phage is virulent or temperate, since this information is important regarding the phenotype of lysis plaques with Geo23.

- L111-113: While it's really intriguing that CBASS does not provide the same resistance to *B. subtilis*, if the infection is made on agar or in liquid medium, I realized that you used two different temperatures for these tests (37°C and 30°C respectively). It'll be better to see the plate assay from Fig.1 and Fig. S2 made at the same temperature for Geo23, as such parameter can significantly the outcome of the infection.

- It's really intriguing that infection by the phage Geo21 can be stopped by the CBASS system, since this phage belongs to the Jumbo-phage family. Some studies suggest that the phage genome is shielded early during the infection process, explaining its resistance against the CRISPR-Cas system (<https://doi.org/10.1038/s41586-019-1786-y>). The observation that the CBASS system is efficient even against Jumbo-phages suggest that the effector protein (Nuc-*SAVED*) is able to get in the shell-structure to degrade both phage and bacterial DNA. Confirming this fact can be done easily, and the information could really improve the impact of this study on our understanding of the CBASS system. But, since it's not about the main message of this study, I am not requiring it as a major review. Confirmation of degradation of phage/bacterial DNA in infected cells could be assessed with

9a qPCR assay; or by microscopy with any DNA binding agent like DAPI.

- Figure 1: The difference in the sensitivity of the different phages to CBASS is really intriguing, and some elements brought by the authors (L113-123) may explain why Geo23 is poorly targeted by the CBASS system. An easy explanation for these differences could also be the eclipse period of each phage (time required to perform a lytic cycle after DNA injection). This paper and some others have clearly shown that CBASS is a system becoming active quite late during the infection process. I think it would be useful if the authors provide information about the eclipse periods of the different phages used in this study, in a way to rule out that Geo23 is almost insensitive to CBASS just because it lyses the bacteria faster than the CBASS system can kill it.

- The model proposed in Fig.6 and during the final discussion (L430-435) is nice and fits the paper data. Nonetheless, I am wondering if the authors could expand their model regarding the cyclase-Cap2 interaction. In Fig. 3a, it seems clear that Cap2-Cyclase complexes are "fused together", creating the multiple high molecular weight species observed on the gel. Do the authors think that this *in vitro* assay could be a clue that the cyclase is activated by oligomerization, of Cap2-cyclase complex first, and then by an unknown mechanism, Cap2 are removed from the cyclase oligomer. The need for high ATP concentration to form these complexes (Fig 3a.) and low ATP concentration for cyclase activation could be part of this mechanism too. I think more discussions are needed about this in the discussion.

- L212-216 and Fig. S4 : it is not crystal clear for me how the authors can make a comparison of the quantity of cyclase in the cytoplasm and along the membrane, since they are performing a pull-down for the cytoplasmic fraction, but not for the membrane one. I would like to see explanation about this comparison clarified somewhere.

Minor text modifications & typos:

General remark about the figures : the "*" representing p-values are missing in all figures of the paper.

In the M&M, for IPTG induction, it is not always clear for how long and at which temperature the induction has been made, and if the culture were shaken. Also, some description of centrifuge speed in rpm, while g are more convenient for transposition to another model of centrifuge.

L31-33 : The sentences sound weird, even if I get the main idea. Talking about p bacterial defense system for eukaryotic sound out of topic.

L109 : please add « in our experimental conditions »

L115 : Are any of these four phages close to each others?

L116 : Author may add (if available) the identity of the gene coding for the shell-protein in Geo21's genome, and information about it in this paragraph.

L121 : two "the" in the sentence -> the the indicator.

L140-150 : Please explain why the cyclase could still be in the membrane fraction with its C-terminal deleted? Do you think the isopeptide bond is made with other proteins with other residue in the cyclase, as you showed L216-218?

L261-263 : I think precision about the "lytic phenotype" should be made. Are you talking about its ability to bypass the CBASS resistance, or about the size/clarity of the lytic plaques.

L224-225 : Please remind the reader that this binding has already been seen before in

10

other works, as you state in the introduction (ref 22,23).

L230-231 : I disagree on this point, the light band/smear observed in the Δ Cap3 & Δ pspA might suggest that these species exist, but in a lower quantity. Unless you have proteomics data to refute my point.

L251-253 : And the proteomics data didn't reveal conjugation on other residues of the protein? Precision should be made about this in the text.

L271-272 : Does the Δ pspA have any growth defect compared to a wild type?

L665 : the word « Infection » does not belong to this sentence.

L747 : I think that "his-tagged protein" should be replaced by "bacterial pellet"

L746-750: the sentence about resuspension seems to be repeated.

L750 : Could you please provide the parameters of the sonication, and bring precision if the sample was kept on ice or not.

L1020-1021 : Left and right should be replaced by top and bottom.

Reviewer #3 (Remarks to the Author):

In the study, Krüger et al. explore mechanistic aspects of the CBASS anti phage defense system in *Bacillus cereus*.

The experiments are generally well designed and follow logic progression. Expressing the CBASS system in *Bacillus subtilis* and *E. coli*, they find that the CBASS component Cap2 conjugates the CBASS cyclase to the abundant protein PspA. They find that Cap3 is required to cleave the cyclase-PspA conjugation, to release the now active cyclase, which can then produce the cyclic nucleotide signal to launch the CBASS cascade.

General comments:

Overall, the manuscript appears rushed and is difficult to follow with many errors as exemplified below.

The methods section is particularly difficult to read. This distracts from the overall findings and makes it very difficult for the reader to follow.

The authors should rewrite the manuscript to better guide the reader. In its current form, the paper is targeted specifically for the CBASS field and not a broader microbiology audience.

The introduction fails to properly describe the CBASS system, including the key role of the cyclic nucleotides. The authors do not mention that it is an abortive infection system. An introductory figure with an overview of the CBASS system would greatly improve the manuscript.

There are issues with the font in many of the figures, e.g. Fig. 4B.

The methodology is generally not explained sufficiently for others to be able to reproduce the data. The plasmid cleavage assay and the nucleotide extraction procedures are particularly unclear.

Several key terms are not defined: eg. CRISPR, ESCRT, Nuc-*SAVED* (not introduced until

line 381).

The nomenclature is inconsistent. Throughout the manuscript, standard nomenclature for naming genes and proteins are confused. E.g. Fig. 1 “ Δ Cap3”.

The section with the extended data figures appears rushed. Nonexistent subfigures are referred to in the figure legend, e.g. Extended Fig 2, where the figure legend refers to sub Fig c, which does not exist.

Specific comments:

The authors introduce four new phages. They rely on specific time points after infection to determine the effect on CBASS, but the timing of phage lifecycles is not determined.

Therefore it is unclear at which point in the phage lifecycle the tests are performed.

Fig. 1. The significance values are not plotted.

Fig. 4a. Why not show the free cyclase, to show that it gets decoupled and not e.g. degraded?

Fig. 4d. and Extended Fig. 7b appear to follow a similar setup, but the non-infected controls are dissimilar and it is unclear from the figure legend why this would be.

Fig. 4e. Why no SD?

Fig. 6. The figure would benefit from adding more text in connection with the arrows.

Line 23: “after infection by the isopeptidase Cap3”. This sentence is confusing.

Line 30: “The best-known examples for bacterial defence systems are the adaptive and innate immune system of eukaryotes”. This is confusing.

Line 44-46: The references are missing.

Line 64: “To increase our understanding of CBASS regulation”. This implies testing how the system is regulated in its native host. The statement should be reworded to better reflect the focus of the study, which is more mechanistic in nature.

Line 87: The authors introduce the Δ 6 strain. They explain that it lacks 5 prophages. What is the 6th deletion?

Line 146: “A-coupled beads”. A is not explained.

Line 194: The names of strains are introduced. These names are not used consistently and seem out of place here.

Line 263: “showed the most lytic phenotype (Fig. 1b), supposedly inducing the most drastic CBASS

response and, thus, making it an ideal candidate to study our system in vivo”. CBASS affects the Goe23 plaque phenotype but does not affect the PFU, so it is unclear why it is the ideal candidate. Why not choose one of the phages where CBASS reduces PFU?

Line 263: “the most lytic phenotype”. I suggest “most clear plaques”.

Line 286: “infection drop assay”. I suggest “plaque assay”.

Line 288: It is not explained what cA3 is.

Line 301: “This confirmed that the cyclase is inactive in the absence of phage”. This is a very strong statement based on the data. The assay does not measure cyclase activity directly and relies on heterologous expression of the cyclase, followed by indirect measurement on the effector.

Line 308: The authors refer to Fig. 1d, which is missing.

Line 309: “Cap2 is essential for activation”. Why was the cap2 mutant not tested in this assay?

12

Line 322: This section seems out of place with the flow of the story.

Line 436: The authors discuss whether PspA could sense initial phage injection-mediated membrane stress and activate the CBASS system, but in line 301 they conclude that the system senses phage infection around 30 min post infection, similar to what has been observed before. They should discuss this in relation to their comment on PspA.

Line 438: "survival program". CBASS is an abortive infection suicide program, not a survival program.

Author Rebuttal to Initial comments

We are grateful to the reviewers who provided extensive, constructive and helpful comments for our manuscript. We have addressed every point made by the three reviewers below. In the revised manuscript we include a substantial amount of additional experimental data to meet the expectations of the reviewers. Specifically these include:

- **Repetition of conjugation assays with PspA from *Bacillus cereus***
 - This new data confirms that Cap2 is able to use both PspA homologs (PspA^{Bsu} and PspA^{Bce}) as substrates for cyclase conjugation. We have to note, however, that purification of the PspA^{Bce} protein was challenging and the protein is not very stable. In addition, we identified conjugation target residues on PspA^{Bce}. This new data is discussed in the revised version of the manuscript and can be found in Extended Data Fig. 5c.
- **Addition of Cap2 catalytic mutants to infection assays (Fig 1b, c) and effector activation assays with extracts from CBASS uninfected and infected cells (Fig. 4d, Ext. Data Fig. 7)**
 - These data provide evidence that the catalytic Cap2 mutants and the deletion mutant behave in the same way regarding their ability to activate the cyclase (Fig. 4d, Extended Data Fig. 7) and provide immunity (Fig. 1b, c).
- **Analysis of cyclase activity in response to infection with different phages by nucleotide extraction of infected cultures and effector activation assays (Fig. 4e, f, g)**
 - We extracted nucleotides over a broader range of time after infection with phage Goe23 and studied Nuc-SAVED activation in the plasmid cleavage assay. This revealed a significant increase of plasmid cleavage activity with nucleotides extracted 25 minutes post infection in the *pspA* deletion strain compared to the CBASS wild type (Fig. 4e, Extended Data Fig. 9). Repeating this experiment with phage Goe21 and Goe26, however, did not show significant differences between the CBASS and CBASS Δ *pspA* strains. These observations are discussed and put in context.

- **Infection experiments with strains expressing CBASS from a downregulated promoter. This strain was constructed by mutating the hyperactive PdegQ promoter regulating expression of the CBASS operon in *B. subtilis* back to wild type analogous to Stanley *et al*, 2005.**
 - This showed that low expression of CBASS strongly reduces immunity in *B. subtilis* against infection with phage Goe21, Goe23, and Goe26. Interestingly, deletion of *pspA* in such cells provided some protection against phage Goe21 and Goe26 in liquid culture. This new data can be found in Fig. 4c and Extended Data Fig. 6c and supports the idea that conjugation to PspA does affect immunity and phenotypes may be masked in the overexpressed system.

Malcolm White & Larissa Krüger

12 December 2023

Reviewer #1 (Remarks to the Author):

In this study Krüger et al. describe the reversible conjugation of a CD-NTase (cyclase) to the ubiquitous phage shock protein (PspA) and propose a model by which this conjugation plays a role in the sequestration of the cyclase to limit spurious activation of the Type II CBASS system in the absence of phage infection. They first show the genetic and catalytic requirements of the *B. cereus* CBASS for protection against phage predation in the heterologous *B. subtilis* host and further show the unexpected finding that this cyclase is conjugated specifically to the *B. subtilis* PspA protein in a Cap2 dependent manner. The Cap3 enzyme is also shown to scarlessly proteolyze the cyclase-PspA conjugate in vitro and in vivo, and this activity occurs late during infection with phage Groe23. The cyclase in vitro appears to be in a state of reduced activity, regardless of its conjugated state, in vitro and inactive in vivo in the absence of phage. This suggests that a critical condition to activate the cyclase during infection remains to be discovered, and the authors suggest based on in vitro cyclase assays this could be related to the abundance of intracellular nucleotides (e.g., ATP). Heroic efforts were made to demonstrate the utility of the PspA-cyclase conjugate in vivo, but the physiological role of this interaction remains enigmatic, with no conclusive evidence demonstrating this interaction between the host PspA and the CBASS cyclase is necessary for defense against phage by the Type-II CBASS. The expansion of these findings to include additional phage infection experiments (e.g., Groe26) validating the delayed activation hypothesis, demonstration of cyclase conjugation to diverse PspA substrates (not only *B. subtilis* PspA), as well as testing the CBASS antiphage activity under control of its native promoter (*B. subtilis* host is

14fine) would greatly improve the significance of these findings.

I want to commend the authors for their hard work, and I implore them to please read the following comments with the knowledge that they are meant to be helpful and written with a voice of curiosity and encouragement. There is no malice intended by any language that follows and in no way are they meant to be combative, rude, or demeaning. It is likely that some of these ideas and observations presented may reflect my own misinterpretation of data or text, despite my best efforts to be thorough. Please know that I respect your time, effort, and diligence in preparing this manuscript and thank you for reporting these interesting results.

Response: We thank the reviewer for the kind words about our work.

Primary Comments:

The characteristics of both plaquing and planktonic growth of *B. subtilis* Δ6 + CBASS challenged with Goe23 are dramatically different from the other phages tested (Figs. 1B & 1C, Ext Data Fig 2). The Goe23 plaquing phenotype and the crash in OD (~3-5h) could be indications of slow activation of CBASS that is specific to this phage, as these are not seen in similar challenges with the three other phage. Because the delayed time to activation (~ 30 min) of this CBASS is an important conclusion of this study, it is important to demonstrate this is not a phenomenon specific to Goe23 infection alone (Figs. 4b,d,e and Ext. Figs. 6 and 7) (i.e., is delayed Cap3 deconjugation/activation of the cyclase only happening with Goe23). Additional temporal experiments (Similar to those performed in Ext. Figs. 6 and 7) should be performed using at least one additional phage. Perhaps Goe26 might be the most reasonable follow-up phage to use as both planktonic and plaquing data are currently presented, which could ease the burden of these experiments.

Response: We thank the reviewer for the suggestions to include other phages. The revised version of the manuscript contains additional effector activation experiments with nucleotides extracted from CBASS-expressing cells infected with Goe21 and Goe26 (Fig. 4f, g, supporting previous Ext. Fig. 7). We extracted nucleotides from cells infected with phage Goe21, Goe23 or Goe26 after different time points post infection and measured their ability to activate the cognate CBASS effector Nuc-SAVED in a plasmid cleavage assay. These experiments confirm that the cyclase is activated after infection with all three phages. Interestingly, we detected CBASS nucleotides only 25 min, 35 min and 45 min after infection with phage Goe23, Goe26 and Goe21, respectively. This late activation of CBASS during the infection cycle is in agreement with our previous observations. We also observed a significant increase of plasmid cleavage activity with nucleotides extracted 25 minutes post infection with phage Goe23 in the *pspA* deletion strain compared to the CBASS wild type (Fig. 4e, Extended Data Fig. 9).

Lines 253-254, “The low sequence conservation between PspA homologs may explain why Cap2 requires flexibility in target recognition (Extended Data Fig. 5c)”. The *in vitro* assay is a neat tool and I commend the authors for the monumental task of purifying all these components and establishing a condition that recapitulates some of the aspects observed in their *in vivo* conditions. However, the *in vitro* condition likely has some substantial artifactual activity where the Cap2 enzyme appears more promiscuous in its conjugation of the cyclase to alternative substrates/residues (e.g., high MW complexes of Cap2-cyclase in Fig. 3a not seen *in vivo*, Figs. 2a, 2d, Extended Data 3a, Extended Data Fig. 9a). However, *in vivo* results demonstrate cyclase-specific peptides are found only on residues K147 and K220 (Fig. 2b, Lines 176-177, Lines 332-333). To suggest that there is flexibility in target recognition by Cap2 *in vivo* activity to facilitate its ability to work with diverse PspA substrates (i.e., function in diverse bacterial hosts), the authors need to perform similar purifications and Western blots as those presented in Fig. 2d using strains containing the K147R and K220R variants of the *B. subtilis* PspA and demonstrate whether comparable alternative Lys residues on PspA are truly utilized as substrates *in vivo*. It is also important to assess the capacity of Cap2 to conjugate the cyclase to *B. cereus* PspA, which is presumably the “natural” substrate, and determine the locations of preferred Lys residues for cyclase conjugation.

Response: We realize that the determination of the target lysine residue on PspA is limited by several points. Firstly, as mentioned by the reviewer, we do not know the importance of the two lysine residue *in vivo*. And secondly, we likely miss modified lysine residues in our MS analysis, due to the generation of peptides unsuitable for detection. To address these points, we analysed the conjugation of the cyclase to *B. cereus* PspA in our *in vitro* conjugation assay. Although the *Bce* homolog was not very stable, we observed conjugation of the cyclase to PspA^{Bce}. This shows that Cap2 is capable of using both PspA homologs as substrates for conjugation. This new data can be found in Extended Data Fig. 5d.

In Fig. 4d, use of either of the Cap2 inactive Cys->Arg variants (Figs. 1b,c, 5a) is important to demonstrate the utility of conjugation and would be more convincing than the double CBASSΔCap2ΔCap3 currently presented (i.e., the variants are a precision tool for addressing the role of conjugation while the double mutant is a hammer). If it is impractical to repeat these experiments with the cap2 variants, it is important to justify the use of the double knockout mutant in lieu of these superior strains (this is also applicable to Fig. 2d, but more critical to address in Fig. 4d where cyclase activity is explored *in vivo*).

Response: We thank the reviewer for pointing out this issue. We repeated the experiment with the catalytic mutants of Cap2 and this data can be found in the revised version of the manuscript. Plasmid cleavage cannot be observed with both mutants, same as the Cap2-Cap3 double deletion mutant, indicating that the cyclase is not activated in such cells. On another note, we added the Cap2-Cap3

double deletion mutant to the infection assays of Fig. 1b,c to complement both observations.

In conjunction with the previous comments, would the authors please double check that in generating the double cap2 cap3 deletion strain (e.g., Fig. 4d) an unintended catalytically inactive chimeric cyclase has not been created (i.e., the native cyclase stop codon is preserved) and the Nuc-SAVED effector is still likely to be functional (i.e., start codon is preserved, a putative ribosome binding site exists, and transcription of the effector is unlikely to be interrupted.. basically check for any obvious indications of polar effects on the effector). There is overlap between the final nucleotide of the cyclase stop codon and the first nucleotide of the cap2 start codon and the Nuc-SAVED effector overlaps with the last four nucleotides of cap3. Generation of unintended mutations in either the cyclase and/or Nuc-SAVED could have significant consequences for the interpretation of cyclase activity and phage challenges assays reported anywhere the CBASS Δ Cap2 Δ Cap3 strain is used.

Response: We agree that it is essential to confirm that the sequence of any of the constructs used is flawless. As described in the supplementary table, the double cap2 cap3 mutant has been generated as follows: amplification of the cyclase gene, cloning into the vector pGP1460, amplification of the effector gene with a forward primer adding a RBS, and cloning into the vector downstream of the cyclase gene (pLK04). The plasmid has been sequenced and we can confirm that both genes are complete from start to stop.

Lines 317-320, there is discussion that the lack of the native host context may be responsible for the absence of a clear phenotype for the PspA-cyclase conjugation. The use of the *B. cereus* CBASS "...expressed under a strong, constitutive *B. subtilis* promoter (PdegQ)..." (Lines 86 – 87) might be masking a PspA phenotype. The discovery of the specific PspA-cyclase conjugate is a remarkable finding, demonstration of its relevance to the initiation and deployment of the CBASS in vivo is important to bolster the significance of this discovery. I would strongly suggest the authors generate two *B. subtilis* strains (pspA⁺ and pspA⁻) where the CBASS is regulated by the native promoter from *B. cereus*, rather than PdegQ. Evaluating the antiphage activity of these two strains against the four Groe phage in plaquing assay and planktonic challenges is likely the best chance to reveal a cyclase-PspA phenotype that would leave no doubt as to its relevance to bacterial immunity.

Response: We agree that the overexpression of CBASS could mask the regulatory function of the cyclase conjugation. To address this issue, we mutated the hyperactive PdegQ_H promoter (very strong) back to the wild type PdegQ promoter with very low activity (as demonstrated by Stanley 2005). With such low expression, CBASS was unable to confirm notable immunity against any of the phages tested in this study in a plaque assay (Extended Data Fig. 6c). Interestingly, when we compared the initial response to phage infection in liquid medium, we found that in the absence of PspA, low CBASS expression provided

17some protection against phage Goe21 and Goe26, but not Goe23 (Fig. 4c). This indicates that under low CBASS expression, presence of PspA is inhibitory for CBASS immunity, likely due to conjugation of the majority of the cyclase to PspA. These new data can be found in Fig. 4c and Ext. Fig. 6c.

Secondary Comments:

Line 150, previously in Fig. 1 the cyclase single amino acid variant A331E alone lost all ability to prevent phage invasion in the population (presumably no longer a Cap2 substrate), I was surprised to see that this variant was never used again in lieu of the C-terminal-truncated cyclase in later experiments (e.g., Fig. 2). Was there a notable reason for using this more substantially mutated gene as opposed to the single point mutation? Substitution of the truncated C-terminal cyclase for the single A331E variant requires demonstration that the response to phage challenge is the same between the two cyclase mutants (e.g., Fig. 1, does the truncated cyclase also fail to prevent phage infection).

Response: The bulk of the work presented was carried out with the truncated variant, both *in vitro* and *in vivo*, before the A331E variant was tested, so the reason for this was essentially technical. We confirmed that the truncated cyclase had a very similar basal cyclase activity to the wild-type protein, so we are reasonably confident that the two variants are functionally comparable. Due to the requirement for extensive further experimentation to address the key points raised by reviewers we did not carry out further tests to compare the two “tail” variants.

Lines 215-216, It is hard to evaluate if the Westerns in Extended Data Fig. 4 show that more cyclase is associated with the membrane fraction compared to the cytosolic fraction without a reference protein sample for relative intensity comparisons within and across blots (e.g., purified cyclase of a known concentration run alongside the two fractions to compare intensities)?

Response: We agree and removed the statement about quantitative distribution from the text.

Lines 323-328, please comment on how similar or divergent the Bce Cap2 is from the *V. cholerae*, *E. coli*, and *P. aeruginosa* enzymes previously tested. It seems that the Bce Cap2 studied here may have a unique mechanism to interact with PspA that is not present in these other Cap2 orthologs. Perhaps the utilization of PspA as a substrate could be predicted in other Cap2 enzymes from such a comparison?

Response: We thank the reviewer for bringing up this interesting point. We compared the Bce Cap2 protein with the homologs and the sequence conservation is indeed not as high. The *Vibrio* and *E. coli* proteins share 51% sequence identity (72% similarity), while the Bce protein shares only 15-17%

18sequence identity (35-36% similarity) with either of them. This is not entirely unexpected as *Bacillus cereus* is more distant in the tree. It would be interesting to investigate the interaction between Cap2 and PspA further to understand whether interacting residues in Cap2 are conserved. We added the following sentence to the manuscript: “The Bce Cap2 protein is phylogenetically more distant and shares only around 35% amino acid sequence similarity with these homologs.”.

Line 326-330, Does the heterologous *E. coli* host not encode a native *pspA*, is the abundance of this native PspA too low to detect HMW cyclase conjugates, or is the native *E. coli* PspA an inferior Cap2 substrate? Also, now that the experiments are being performed in a new host, please explicitly state which *pspA* allele is being over-expressed in *E. coli*.

Response: Even though *E. coli* and *B. subtilis* both encode for PspA homologs, the context and the Psp system work very differently. In *E. coli*, PspA forms an inhibitory complex with PspF, thereby preventing induction of the Psp response. This seems to make PspA unavailable as a substrate of Cap2. In *B. subtilis* on the other hand, PspA is not associated with another protein of the Psp system in uninducing conditions. We have added a clarifying sentence to our discussion. We also explicitly state now which *pspA* allele is overexpressed in *E. coli*.

Line 422-424, “...to prevent activation of CBASS in the absence of phage ... CBASS uses its protein modification machinery to conjugate the cyclase to the highly conserved PspA protein...” I am unconvinced that conjugation of the cyclase to PspA is preventing spurious activation of this CBASS in the absence of phage. It appears the cyclase is in an inactive form in the absence of phage in a manner independent of its post-translational conjugation to PspA. This is evident in Fig. 2a where there is over-expression of the cyclase and constitutive expression of the Nuc-SAVED effector (i.e., massive amounts of un-conjugated cyclase and effector, but no killing). Additionally, the inactivity or limited activity of purified cyclase in Fig. 5f (conjugated and unconjugated) also suggests the cyclase is primarily activated by an unknown factor (as stated in Line 412). The presence or absence of PspA also does not appear to influence the ability of CBASS to effectively limit phage predation (Fig. 4c). Therefore, based on the data in Figs. 4d,e and that mentioned above, it would be more appropriate to say the conjugation of the cyclase to PspA could be a means to slow cyclase activation during infection (rather than prevent activation in the absence of phage).

Response: We agree and have reworded the relevant text as suggested.

Additional Comments / Observations

[These do not require responses; I only want to bring them to the authors' attention and are all marked

with an *]

*In general, the absence of asterisks for relevant p-values appears to be a systemic oversight throughout the manuscript (e.g., Fig. 1C and Fig. 4), including Extended Data Figs.

Response: We apologize for the missing asterisks. They were apparently removed during the conversion to pdf. We will make sure this won't happen again.

*In general, consistency across all figures with the usage of white and magenta triangles for identification of monomeric proteins and cyclase-PspA conjugates would be helpful. That being said, I found the most useful application of triangles was in Fig. 3a where only the magenta triangle was common across blots and included the name cyclase-PspA while the individual monomeric proteins were called out with arrows by name (rather than white triangles). This simplified and uniform identification scheme would greatly simplify interpretation of many of the blots and gels.

Response: We agree and introduced a more consistent labelling for the conjugate and the monomeric proteins.

*Line 98, Fig. 1c – Error bars are difficult to see, and it appears a formatting error occurred in the histogram for Goe21 CBASSΔCap3. Also check universally that sample labels are formatted inconsistently across similar experiments (extra spaces in names, capitalized gene names, font size, etc.) (e.g., Figs. 1C & 4D).

Response: We thank the reviewer for drawing our attention to this issue and we fixed it in the revised version of the manuscript.

*Line 89, italicize the prophage-like element “skin”.

Response: This has been changed in the manuscript.

*Lines 125-126, It would be more correct to adjust language from “induced point mutations in relevant residues”, to something like, “introduced point mutations to produce catalytically inactive variants”. Keeping genes as things to mutate and proteins as variants.

Response: We changed our wording in this sentence.

*Lines 126-127, rework the sentence beginning with, “The absence of cap2 and cap3...” as it is seemingly contradictory to the data and statements which immediately follow re: cap3.

Response: We changed our wording in this sentence.

Line 180, Fig. 2a it is not immediately apparent what the five “” refer to on the gel (analyzed by MS?). Additionally, the gel is very small (e.g., MWs are overwritten on one another) and increasing the size of this key piece of evidence would be helpful. In Fig. 2c, consider identifying the HC as done for Extended Data Fig. 4.

Response: We thank the reviewer for pointing out this issue and we changed the figure accordingly.

Line 240, missing description of significance ascribed to a single “” shown in Figure 3c.

Response: The description has been added to the figure legend.

*Lines 262-264, this may be semantics but to my eye, Groe23 appears to induce the WORST CBASS response of the four phages tested (Fig. 2b) leading to the least direct reduction in total plaques and delayed immunity in liquid medium.

Response: We now include another phage in the analysis to support our findings.

*Line 287, please indicate that Groe23 was used to induce the cyclase activity in extracts used in Fig. 4d

Response: Such statement has been included in the figure legend.

*Line 289, is Fig. 4e the same data presented in Extended Data Fig. 8b? It is not clear how these two experiments are different, as Fig. 4e looks to display the mean values of individual data points in Extended Data Fig. 8b. Additionally, it appears that STD is missing in Fig. 4e.

Response: Fig. 4e and Extended Data Fig. 8b show indeed the results of the same experiments. We agree that it is repetitive and removed Extended Data Fig. 8b. We apologize for the missing SD in Fig.4e. The error bars, same as the ** in some figures were apparently removed during the conversion to pdf. We will make sure this won't happen again.

*Lines 298-301, please provide a reference or justification for exploring cA3 (rather than another nucleotide moiety) as the presumed product of the cyclase and activator of Nuc-SAVED effector.

Response: We show later in the manuscript that cA3 is the nucleotide activating the Nuc-SAVED effector.

*Lines 307-309, it does not appear CBASS Δ Cap2 Δ Cap3 is used in all these figures referenced here, please double check (e.g., no Fig. 1d in the manuscript).

Response: We double checked and included this mutant now also in Fig. 1bc.

*Lines 315-316, “This supports the idea that the cyclase needs to be deconjugated prior to activation...”. I suggest amending this language to be less strict for the requirement of deconjugation (“needs”) for cyclase activation, as the Δ Cap3 strain still provides defense against phage (Fig. 1), the cyclase in this strain still synthesizes CA3 in a phage-dependent manner (Fig. 4d), and conjugated cyclase is later shown to activate Nuc-SAVED at the same rate as the full-length monomeric cyclase in vitro (Fig. 5f). Something along the lines of deconjugation “enhances in vivo cyclase activity”.

*Line 349, reference to Fig. 5c is incorrect, should be reference to Extended Data Figs. 9c,d,e.

Response: This has been changed in the manuscript.

*Line 400-401, “This data suggests that the wild type Bce cyclase is inactive in this unconjugated form in vitro.” Please remove the unconjugated form portion of the statement, as the data only suggests the wild type cyclase catalytic activity is very slow in vitro (Fig. 5e). The activity relative to the conjugation status of the cyclase is not tested until later (Fig. 5f) where it is shown to not play a significant role in the in vitro activity of the enzyme.

Response: Thank you for pointing this out. We changed our wording in this sentence.

*Line 979, Extended Data Fig. 2 title does not mention Goe26 and as the figure compliments the solid agar derived plaquing assays in Fig. 1, I suggest exchanging “in vivo” for “in liquid medium”, or an alternative title like, “CBASS protects planktonic *B. subtilis* Δ 6 from Goe23 and Goe26.”

Response: The title of the figure legend has been changed in the revised version of the manuscript.

*Line 982, references to growth curves in Extended Date Fig. 2 are incorrect.

Response: This has been changed.

*Line 986, Extended Data Fig. 3. In Extended Data Fig 3a, it is unclear what strain “EV” refers to – presumably this is the empty vector control? This legend is also missing descriptions of triangular markers. In extended Data Fig. 3b the PspA primary sequence is redundant (Fig. 2b), please consider removing it from this location as it provides little information relevant to the interpretation of data found in this figure.

Response: We included now a definition for EV and removed the sequence from the figure.

*Line 997, Extended Data Fig. 4, consider adding the white triangles (or better yet, arrows with names) highlighting the monomeric forms of the indicated proteins for all Western blots.

Response: We agree and introduced a more consistent labelling for the conjugate and the monomeric proteins in all figures.

*Line 1021, the Western blots in Extended Data Fig. 5b and in Fig. 3b appear to be the same, if so please mention this here. I noticed they looked very similar while trying to eyeball the intensities of the cyclase-PspA conjugates across PspA variants.

Response: The Western blots indeed show the same experiment. We now state this in the figure legend.

*Line 1026, for ease of interpretation please highlight the residues in the PspA alignment that correspond to *B. subtilis* PspA K147 and K220 in Extended Data Fig. 5c.

Response: These residues are now highlighted.

*Line 1033, please indicate what R1 and R2 are in the gel legends of Extended Data Fig. 6a mean (presumably these are two biological replicates).

Response: The description has been added to the figure legend.

*Line 1043, please indicate what +/- symbolize in Extended Data Fig. 7

Response: Such description has been added.

*Line 1054, none of the panels in Extended Data Figure 8 are referenced in the body of the MS that I can

find. Additionally, more information is required in the figure legend to understand the experiments depicted in the gels. (e.g., the use of R1 and R2 in the first-row of the gel legends, “30” in the second-row, and the third-row numerals 1, 2, 5, 15, and 30).

Response: Extended Data Fig. 8 is now referred to in the main body of the text. Descriptions have also been added to the figure legend.

*Line 1066, Extended Data Figure 9a, please number the band boxes on the gel that were analyzed by MS.

Response: The boxes have been numbered.

*Line 1083, “...conjugate cleavage assay from b was analysed by intact mass spectrometry.” This statement should refer to Extended Data 9c rather than b.

Response: This has been changed.

Reviewer #2 (Remarks to the Author):

The work from Krüger et al. describes the regulation of the CBASS system from *B. cereus* in *B. subtilis*. After a clear and up-to-date introduction, the authors demonstrate with strong proofs that the nucleotide cyclase is conjugated by the protein Cap2 to the ubiquitous membrane protein PspA. This sequestration of the cyclase to the membrane prevents non-relevant activation of the CBASS system. The protein Cap3 has been shown to be necessary for the deconjugation of the Cyclase from PspA. These findings present a real advancement in the understanding of the regulation of phage defense system in bacteria, and especially for CBASS. This work presents a strong interest for every research community with interests in phage defense systems. The paper is well written and made with solid science. The conclusions drawn by the authors are clear, careful and rely on clear proof and serious studies. The M&M, as well as the legends are nicely detailed.

All experiments related to the main message of the articles are flawless and well made, but I noticed some points of the study or the discussion could be improved by some additional information/discussions.

Overall, I'll recommend the publication of this article after a round of minor revision and say that the authors accomplished an incredible work.

Response: We thank the reviewer for the kind words about our work.

Here are the main questions/critics/suggestions that I have regarding the manuscript:

- Taxonomy of the four new phages L90-95 + Supp table 1: You did a great job to make taxonomic attributions to these new phages, unfortunately they are now obsolete (see <https://doi.org/10.1007/s00705-022-05694-2>). I'll strongly suggest you to update their names and taxonomic distribution (forget Myoviridae et al families and go deeper in classification if you can (see <https://doi.org/10.3390/v9040070> for an up-to-date guide)). I'll also recommend adding to Table S1 a column saying if the phage is virulent or temperate, since this information is important regarding the phenotype of lysis plaques with Geo23.

Response: We are grateful to the reviewer for drawing our attention to the issue of the outdated taxonomic distributions. We have updated the names and classifications and this information can be found in Table S1. We also added another column indicating the lifestyle of the respective phage.

- L111-113: While it's really intriguing that CBASS does not provide the same resistance to *B. subtilis*, if the infection is made on agar or in liquid medium, I realized that you used two different temperatures for these tests (37°C and 30°C respectively). It'll be better to see the plate assay from Fig.1 and Fig. S2 made at the same temperature for Geo23, as such parameter can significantly affect the outcome of the infection.

Response: We thank the reviewer for drawing our attention to this issue. We have to admit that the 30 °C conditions were wrongly described in the method section. The experiments in liquid medium were, same as the plate assays, done at 37°C. We updated our methods accordingly. We also repeated the experiments with phage Goe23, Goe21, and Goe26 and added the new data to Extended Data Fig. 2.

- It's really intriguing that infection by the phage Geo21 can be stopped by the CBASS system, since this phage belongs to the Jumbo-phage family. Some studies suggest that the phage genome is shielded early during the infection process, explaining its resistance against the CRISPR-Cas system (<https://doi.org/10.1038/s41586-019-1786-y>). The observation that the CBASS system is efficient even against Jumbo-phages suggest that the effector protein (Nuc-*SAVED*) is able to get in the shell-structure to degrade both phage and bacterial DNA. Confirming this fact can be done easily, and the information could really improve the impact of this study on our understanding of the CBASS system. But, since it's

25not about the main message of this study, I am not requiring it as a major review. Confirmation of degradation of phage/bacterial DNA in infected cells could be assessed with a qPCR assay; or by microscopy with any DNA binding agent like DAPI.

Response: We agree that the ability of CBASS to provide resistance against a jumbo phage is interesting and provides opportunity to understand the mechanism of CBASS further. It is quite possible that degradation of the host chromosome, outside the phage shell, is sufficient to confer immunity. We have now added some discussion of that point, but we feel that further experimentation is beyond the scope of the current paper

- Figure 1: The difference in the sensitivity of the different phages to CBASS is really intriguing, and some elements brought by the authors (L113-123) may explain why Geo23 is poorly targeted by the CBASS system. An easy explanation for these differences could also be the eclipse period of each phage (time required to perform a lytic cycle after DNA injection). This paper and some others have clearly shown that CBASS is a system becoming active quite late during the infection process. I think it would be useful if the authors provide information about the eclipse periods of the different phages used in this study, in a way to rule out that Geo23 is almost insensitive to CBASS just because it lyses the bacteria faster than the CBASS system can kill it.

Response: We thank the reviewer for raising this interesting point. A similar issue has been addressed by reviewer 1. In the revised manuscript, we include new experimental data on the activation of the cyclase after infection with different phages (Fig. 4 e, f, g). We extracted nucleotides from cells infected with phage Goe21, Goe23 or Goe26 after different time points post infection and measured their ability to activate the cognate CBASS effector Nuc-SAVED in a plasmid cleavage assay. These experiments confirm that the cyclase is activated after infection with all three phages. Interestingly, we detected CBASS nucleotides only 25 min, 35 min and 45 min after infection with phage Goe23, Goe26 and Goe21, respectively. Comparing this to the infection kinetics of wild type bacteria (named “no CBASS” in the figure) shown in Extended Data Fig. 2, it provides further evidence that CBASS is activated late in the infection cycle. At the moment we do not have enough experimental evidence to confidently argue whether Goe23 is only minorly affected by CBASS because of its short infection cycle, or because it encodes anti-defence proteins.

- The model proposed in Fig.6 and during the final discussion (L430-435) is nice and fits the paper data. Nonetheless, I am wondering if the authors could expand their model regarding the cyclase-Cap2 interaction. In Fig. 3a, it seems clear that Cap2-Cyclase complexes are “fused together”, creating the multiples high molecular weight species observed on the gel. Do the authors think that this in vitro assay could be a clue that the cyclase is activated by oligomerization, of Cap2-cyclase complex first, and then

26by an unknown mechanism, Cap2 are removed from the cyclase oligomer. The need for high ATP concentration to form these complexes (Fig 3a.) and low ATP concentration for cyclase activation could be part of this mechanism too. I think more discussions are needed about this in the discussion.

Response: The Cap2-cyclase complexes observed in vitro are reminiscent of previous studies and, as referee 1 points out, could be an in vitro artifact. Importantly, there is no evidence for these complexes in any of the in vivo experiments. Nonetheless, it is true that the final conjugation partner may well be another cyclase molecule, leading to clustering and activation. We have expanded the discussion to cover these points more fully, as requested.

- L212-216 and Fig. S4 : it is not crystal clear for me how the authors can make a comparison of the quantity of cyclase in the cytoplasm and along the membrane, since there are performing a pull-down for the cytoplasmic fraction, but not for the membrane one. I would like to see explanation about this comparison clarified somewhere.

Response: We agree and in response to this comment and similar comments from reviewer 1 we have trimmed down our wording in this part to reflect the difficulties in accurate quantitation.

Minor text modifications & typos:

General remark about the figures : the “*” representing p-values are missing in all figures of the paper.

Response: We apologize for the missing asterisks. They were apparently removed during the conversion to pdf. We will make sure this won't happen again.

In the M&M, for IPTG induction, it is not always clear for how long and at which temperature the induction has been made, and if the culture were shacked. Also, some description of centrifuge speed in rpm, while g are more convenient for transposition to another model of centrifuge.

Response: We agree with the reviewer and changed our methods accordingly.

L31-33 : The sentences sound weird, even if I get the main idea. Talking about p bacterial defense system for eukaryotic sound out of topic.

Response: We completely agree and changed the sentence.

L109 : please add « in our experimental conditions »

Response: We added this as requested.

L115 : Are any of these four phages close to each others?

Response: Phage Goe16 and Goe26 both belong to the *Herelleviridae* family and *Spounavirinae* subfamily but do not share a high sequence identity (see Table S1 for more detail).

L116 : Author may add (if available) the identity of the gene coding for the shell-protein in Geo21's genome, and information about it in this paragraph.

Response: We agree that this is interesting to look at, however, due to space limitations, we do not believe this to be relevant enough to include in the text. The sequences of all phage genomes are annotated and available for further analyses.

L121 : two “the” in the sentence -> the the indicator.

Response: We corrected this mistake in the manuscript.

L140-150 : Please explain why the cyclase could still be in the membrane fraction with its C-terminal deleted? Do you think the isopeptide bond is made with other proteins with other residue in the cyclase, as you showed L216-218?

Response: Our data shows that the interaction of the cyclase with the membrane does not involve the C-terminus of the cyclase and that conjugation to PspA is not required for membrane association. We clarified this by changing the sentence as follows “We observed this for the full-length cyclase as well as a truncated mutant, in which the flexible C-terminal tail had been deleted, indicating that the C-terminus of the cyclase is not involved in the association with the membrane”.

L261-263 : I think precision about the “lytic phenotype” should be made. Are you talking about its ability to bypass the CBASS resistance, or about the size/clarity of the lytic plaques.

Response: We agree with the reviewer that this term can be confusing in this context and changed it to “most clear plaques”.

L224-225 : Please remind the reader that this binding has already been seen before in other works, as you state in the introduction (ref 22,23).

Response: Thanks for the suggest. We mention in L272 that the cyclase-Cap2 intermediates have previously been observed by other groups and we provide the references too.

L230-231 : I disagree on this point, the light band/smear observed in the Δ Cap3 & Δ pspA might suggest that these species exist, but in a lower quantity. Unless you have proteomics data to refute my point.

Response: We thank the reviewer for their opinion and we do not have proteomics data to rule out that such species are, in low quantity, formed *in vivo*. We changed our wording to “The physiological relevance of these species is unclear, as such prominent bands were not observed in *B. subtilis*”.

L251-253 : And the proteomics data didn’t reveal conjugation on other residues of the protein? Precision should be made about this in the text.

Response: We agree with the reviewer that this sentence is a bit misleading. We observed the modification only on two lysine residues K147 and K220 on PspA. For clarity, we changed this part in the revised version of the manuscript to “This indicates that K147 is a preferred residue for conjugation, while residue K220 is less favoured. As the double mutant is still conjugated to the cyclase, additional unidentified lysine residues on PspA may be targeted for conjugation.”.

L271-272 : Does the Δ pspA have any growth defect compared to a wild type?

Response: We did not observe a growth defect of the Δ pspA mutant in the tested conditions. The growth behaviour of a Δ pspA mutant can now also be found in Fig. 4c.

L665 : the word « Infection » does not belong to this sentence.

Response: We thank the reviewer for pointing this out and we corrected this sentence.

L747 : I think that “his-tagged protein” should be replaced by “bacterial pellet”
L746-750: the sentence about resuspension seems to be repeated.

Response: We thank the reviewer for pointing this out and we corrected this part.

L750 : Could you please provide the parameters of the sonication, and bring precision if the sample was kept on ice or not.

Response: We added this information to the method section.

L1020-1021 : Left and right should be replaced by top and bottom.

Response: We corrected this in the figure legend.

Reviewer #3 (Remarks to the Author):

In the study, Krüger et al. explore mechanistic aspects of the CBASS anti phage defense system in *Bacillus cereus*.

The experiments are generally well designed and follow logic progression. Expressing the CBASS system in *Bacillus subtilis* and *E. coli*, they find that the CBASS component Cap2 conjugates the CBASS cyclase to the abundant protein PspA. They find that Cap3 is required to cleave the cyclase-PspA conjugation, to release the now active cyclase, which can then produce the cyclic nucleotide signal to launch the CBASS cascade.

General comments:

Overall, the manuscript appears rushed and is difficult to follow with many errors as exemplified

below. The methods section is particularly difficult to read. This distracts from the overall findings and makes it very difficult for the reader to follow.

Response: Sorry that the manuscript seemed rushed. We have corrected all the errors identified by the referees as well as others we spotted ourselves, and have tried to clarify the methods section as requested.

The authors should rewrite the manuscript to better guide the reader. In its current form, the paper is targeted specifically for the CBASS field and not a broader microbiology audience.

The introduction fails to properly describe the CBASS system, including the key role of the cyclic nucleotides. The authors do not mention that it is an abortive infection system. An introductory figure with an overview of the CBASS system would greatly improve the manuscript.

Response: Sorry that due to space pressures we were not able to provide a more fulsome introduction to this topic. We have now extended the first paragraph to give more background information for readers not familiar with the field: “ Thus, the eukaryotic cGAS enzyme of the cGAS-STING pathway is a homolog of the bacterial CD-NTase (cyclic nucleotidyl-transferase), which is part of CBASS (cyclic-oligonucleotide-based antiphage signalling system) ^{1,2}. Both pathways function via the detection of viral infection and activation of a nucleotide cyclase which generates a cyclic nucleotide second messenger that in turn activates an antiviral downstream response. cGAMP is used as the second messenger by both the eukaryotic and some bacterial defence systems, but recent studies have revealed a large diversity of bacterial CD-NTase enzymes that are all related to cGAS and synthesize a diverse array of cyclic nucleotides in response to phage infection ¹⁻⁴. These signals in turn activate a wide range of downstream effectors including nucleases, hydrolases and membrane disruption proteins which modulate the bacterial immune response ⁵⁻⁸. “ We hope this gives readers new to the field the information they need to gain an overview of the topic. An overview of the components of the CBASS system can be found in Fig. 1a.

There are issues with the font in many of the figures, e.g. Fig. 4B.

Response: We thank the reviewer for pointing this out and we assume that they are referring to gene names not being italicized. We changed this.

The methodology is generally not explained sufficiently for others to be able to reproduce the data. The plasmid cleavage assay and the nucleotide extraction procedures are particularly unclear.

Response: We have revised the methods accordingly and hope they are now easier to follow.

Several key terms are not defined: eg. CRISPR, ESCRT, Nuc-SAVED (not introduced until line 381).

Response: We added definitions for CRISPR and ESCRT. Nuc-SAVED is described in the introduction when we introduce the individual components of the Bce CBASS system.

The nomenclature is inconsistent. Throughout the manuscript, standard nomenclature for naming genes and proteins are confused. E.g. Fig. 1 “ Δ Cap3”.

Response: We agree with the reviewer and have addressed this issue in the revised manuscript.

The section with the extended data figures appears rushed. Nonexistent subfigures are referred to in the figure legend, e.g. Extended Fig 2, where the figure legend refers to sub Fig c, which does not exist.

Response: We thank the reviewer for drawing our attention to the mistake in the figure legend and changed this in the revised manuscript. We carefully checked all figure legends again.

Specific comments:

The authors introduce four new phages. They rely on specific time points after infection to determine de effect on CBASS, but the timing of phage lifecycles is not determined. Therefore it is unclear at which point in the phage lifecycle the tests are performed.

Response: We agree that this is not mentioned in the manuscript. However, we want to focus on the CBASS response. For this reason, we repeated the nucleotide extractions of CBASS-expressing cells infected with different phages to show that the time it takes to activate the cyclase differs depending on the phage. We would also like to draw the reviewer’s attention to Ext. Fig. 2, in which we show the infection of wild type cells with the relevant phages.

Fig .1. The significance values are not plotted.

Response: We apologize for the missing asterisks, this was an error when we converted the manuscript to pdf. We made sure all figures are complete in the revised manuscript.

Fig. 4a. Why not show the free cyclase, to show that it gets decoupled and not e.g. degraded?

Response: We are not showing the monomeric cyclase in this figure because we noticed that monomeric cyclase is not consistently associated with the membrane after late exponential growth phase. We are showing the whole blot now in Extended Data Fig. 6a together with the cytosolic fractions, which clearly show that the cyclase is not degraded after phage infection.

Fig. 4d. and Extended Fig. 7b appear to follow a similar setup, but the non-infected controls are dissimilar and it is unclear from the figure legend why this would be.

Response: Extended Fig. 7b showed the same data as Fig. 4d, but normalised. We agree that this may be redundant and removed Extended Fig. 7b from the revised version of the manuscript.

Fig. 4e. Why no SD?

Response: We apologize for the missing SD in the figures. This was an error when we converted the manuscript to pdf. We made sure all figures are complete in the revised manuscript.

Fig. 6. The figure would benefit from adding more text in connection with the arrows.

Response: We agree and modified the figure.

Line 23: “after infection by the isopeptidase Cap3”. This sentence is confusing.

Response: We agree and changed the sentence in the revised version of the manuscript.

Line 30: “The best-known examples for bacterial defence systems are the adaptive and innate immune system of eukaryotes”. This is confusing.

Response: Apologies, we agree and changed our wording.

Line 44-46: The references are missing.

Line 64: “To increase our understanding of CBASS regulation”. This implies testing how the system is regulated in its native host. The statement should be reworded to better reflect the focus of the study, which is more mechanistic in nature.

Response: We agree and changed this section.

Line 87: The authors introduce the $\Delta 6$ strain. They explain that it lacks 5 prophages. What is the 6th deletion?

Response: The sixth deletion is the deletion of the polyketide synthase (pks) operon and we added this description when introducing the $\Delta 6$ strain in the manuscript.

Line 146: “A-coupled beads”. A is not explained.

Response: The phrase is “protein A-coupled beads”. We have now capitalised “Protein” to make this clearer

Line 194: The names of strains are introduced. These names are not used consistently and seem out of place here.

Response: We added the names of the strains here to make it easier for the reader to find them in the list of strains in the supplementary data. We agree that we did not do this consistently in the other figure legends and added them to all of them in the revised manuscript.

Line 263: “showed the most lytic phenotype (Fig. 1b), supposedly inducing the most drastic CBASS response and, thus, making it an ideal candidate to study our system in vivo”. CBASS affects the Goe23 plaque phenotype but does not affect the PFU, so it is unclear why it is the ideal candidate. Why not choose one of the phages where CBASS reduces PFU?

Response: We have rephrased this sentence and now include data on two further phase. Further details are provided in the response to Referee 1.

Line 263: “the most lytic phenotype”. I suggest “most clear plaques”.

Response: This has been modified as suggested.

Line 286: “infection drop assay”. I suggest “plaque assay”.

Response: This has been modified as suggested.

Line 288: It is not explained what cA3 is.

Response: We agree and added a description for cA₃ when we first mention it.

Line 301: “This confirmed that the cyclase is inactive in the absence of phage”. This is a very strong statement based on the data. The assay does not measure cyclase activity directly and relies on heterologous expression of the cyclase, followed by indirect measurement on the effector.

Response: We agree and tuned down the wording in the revised version of the manuscript.

Line 308: The authors refer to Fig. 1d, which is missing.

Response: We thank the reviewer for pointing out this mistake to us and it has been corrected.

Line 309: “Cap2 is essential for activation”. Why was the cap2 mutant not tested in this assay?

Response: We thank the reviewer for pointing out this issue. We repeated the experiment with the catalytic mutants of Cap2 and this data can be found in the revised version of the manuscript. Plasmid cleavage cannot be observed with both mutants, same as the Cap2-Cap3 double deletion mutant, indicating that the cyclase is not activated in such cells. On another note, we added the Cap2-Cap3 double deletion mutant to the infection assays of Fig. 1b,c to complement both observations.

Line 322: This section seems out of place with the flow of the story.

Response: We acknowledge the reviewer's comment but we believe that presenting the *in vivo* data in *Bacillus subtilis*, then the confirmation *in vitro* and at last the *E. coli* *in vivo* data is the best order.

Line 436: The authors discuss whether PspA could sense initial phage injection-mediated membrane stress and activate the CBASS system, but in line 301 they conclude that the system senses phage infection around 30 min post infection, similar to what has been observed before. They should discuss this in relation to their comment on PspA.

Response: As the reviewer pointed out, the conclusion in line 301 (previous version of the manuscript) is based on the experimental results presented in this part, while line 436 is part of the discussion that aims to put the results into context.

Line 438: "survival program". CBASS is an abortive infection suicide program, not a survival program.

Response: We have removed this phrase.

Decision Letter, first revision:

Message: Our ref: NMICROBIOL-23071808A

21st February 2024

Dear Malcolm,

Thank you for your patience as we've prepared the guidelines for final submission of your Nature Microbiology manuscript, "Reversible conjugation of a CBASS nucleotide cyclase regulates immune response to phage infection" (NMICROBIOL-23071808A). Please carefully follow the step-by-step instructions provided in the attached file, and add a response in each row of the table to indicate the changes that you have made. Please also check and comment on any additional marked-up edits we have proposed within the text. Ensuring that each point is addressed will help to ensure that your revised manuscript can be swiftly handed over to our production team.

36We would like to start working on your revised paper, with all of the requested files and forms, as soon as possible (preferably within two weeks). Please get in contact with us if you anticipate delays.

In recognition of the time and expertise our reviewers provide to Nature Microbiology's editorial process, we would like to formally acknowledge their contribution to the external peer review of your manuscript entitled "Reversible conjugation of a CBASS nucleotide cyclase regulates immune response to phage infection". For those reviewers who give their assent, we will be publishing their names alongside the published article.

Nature Microbiology offers a Transparent Peer Review option for new original research manuscripts submitted after December 1st, 2019. As part of this initiative, we encourage our authors to support increased transparency into the peer review process by agreeing to have the reviewer comments, author rebuttal letters, and editorial decision letters published as a Supplementary item. When you submit your final files please clearly state in your cover letter whether or not you would like to participate in this initiative. Please note that failure to state your preference will result in delays in accepting your manuscript for publication.

Cover suggestions

COVER ARTWORK: We welcome submissions of artwork for consideration for our cover. For more information, please see our guide for cover artwork.

Nature Microbiology has now transitioned to a unified Rights Collection system which will allow our Author Services team to quickly and easily collect the rights and permissions required to publish your work. Approximately 10 days after your paper is formally accepted, you will receive an email in providing you with a link to complete the grant of rights. If your paper is eligible for Open Access, our Author Services team will also be in touch regarding any additional information that may be required to arrange payment for your article.

Please note that *Nature Microbiology* is a Transformative Journal (TJ). Authors may publish their research with us through the traditional subscription access route or make their paper immediately open access through payment of an article-processing charge (APC). Authors will not be required to make a final decision about access to their article until it has been

accepted. Find out more about Transformative Journals

Best regards,

Reviewer #1:

Remarks to the Author:

I would like to commend the authors for their willingness to address this reviewer's loquacious, voluminous, and extensive comments. Of note, their demonstration of "native" PspA-bce as a substrate for cyclase conjugation and the associated Lys residues, the exploration of *pspA* +/- strains with reduced CBASS expression revealing an in vivo PspA phenotype during phage challenge, further exploration of Cap2 catalytic cysteine variants, and more broadly assessing the temporal activation of CBASS in response to challenge by diverse phages all lead me to enthusiastically support the authors claims as presented. The authors clearly took great pains to polish the figures and text, which greatly eased evaluation of this revised manuscript. I have only minor comments to suggest the authors consider. The authors should be very proud of this substantial and valuable body of work.

Minor Comments:

Lines 23-25, final abstract sentence is missing something / confusing

Lines 138-140, please indicate which catalytic mutants are associated with the E1 and E2 domains of Cap2

Fig 2a, I could not find reference to the asterisks on the gel, consider a different unique

38identifier for the different MW bands being discussed (e.g., a-d)

Lines 401-417, the cyclase product of unknown ATP composition which elutes around 3 minutes is very intriguing – being most prominent in the 5 mM ATP condition. Upon seeing this again, in the revised manuscript, I'm too tempted to whisper in the authors' ears that this could be a very interesting molecule to pursue (in a different study, of course!). Could this represent an inhibitor of Cap3 activity or an antagonist of CA3 activation of Nuc-SAVED? Identification of this molecule and MS quantification of its abundance pre- and post-infection or inclusion in the established in vitro Cap3 deconjugation and plasmid cleavage assays might reveal some hidden role for this curious nucleotide product.

Extended Data Figures 3a-b, these gels look to have the "LC" labeled on different bands

Extended Data Figure 3 Legend Title, "conjugates" to "conjugated"

Reference 20, an updated version of this biorxiv has just been published in a peer-reviewed journal, PMID: 38172623 (I am not an author of this study)

Line 747-748 (Method: Phage infection assays), "...600 nm (OD600) was adjusted to 1.0 and the cells were used to inoculate a 96 well plate..." Is there a further dilution from OD600 of 1.0 that is performed? All liquid phage challenges appear to start around 0.1 OD600.

Reviewer #2:

Remarks to the Author:

In my opinion Krüger et al. did a great job during this revision round. They perfectly answered the critics and comments made by me, but also by the other referees. The additional experiments that they performed in the last months really helped. The overall quality of the manuscript clearly increased and I think it is now ready to be published in this journal.

Reviewer #3:

Remarks to the Author:

Krüger et al. have made a great effort in addressing the comments and suggestions and they have made extensive revisions to their manuscript. In particular, key concepts are now introduced to a broader audience, and the addition of the native promotor data nicely captures some of the nuances of how effective the CBASS system may be in connection with PspA, in the native *Bacillus cereus* strain.

I still highly recommend that the authors measure the timing of the phage life cycles/ latent period, as also suggested by another reviewer. This would allow the reader to compare the timing of CBASS activation in this system, compared to the first reports on CBASS.

Additionally, I have the following minor comments:

Title: Swapping the word "regulates" for "activate" or "trigger" would better reflect the conclusions.

The final sentence of the abstract is unclear.

There are still nomenclature issues, e.g. lines 342 and 1082 (*Escherichia coli*).

Line 46: The sentence implies that CRISPR-Cas is exclusively activated by viral DNA/RNA. This statement should be reworded to reflect that this is one of many mechanisms that activate CRISPR-Cas. Ref. 13 is not appropriate for this statement.

Lines 473-474. This statement is very bold and is not supported by the data since the experiments are not carried out in the native system under non-induced conditions. I suggest deleting this statement.

Reviewer #4:

Remarks to the Author:

Krüger et al. have made a great effort in addressing the comments and suggestions and they have made extensive revisions to their manuscript. In particular, key concepts are now introduced to a broader audience, and the addition of the native promotor data nicely captures some of the nuances of how effective the CBASS system may be in connection with PspA, in the native *Bacillus cereus* strain.

I still highly recommend that the authors measure the timing of the phage life cycles/ latent period, as also suggested by another reviewer. This would allow the reader to compare the timing of CBASS activation in this system, compared to the first reports on CBASS.

Additionally, I have the following minor comments:

Title: Swapping the word "regulates" for "activate" or "trigger" would better reflect the conclusions.

The final sentence of the abstract is unclear.

There are still nomenclature issues, e.g. lines 342 and 1082 (*Escherichia coli*).

Line 46: The sentence implies that CRISPR-Cas is exclusively activated by viral DNA/RNA. This statement should be reworded to reflect that this is one of many mechanisms that activate CRISPR-Cas. Ref. 13 is not appropriate for this statement.

Lines 473-474. This statement is very bold and is not supported by the data since the experiments are not carried out in the native system under non-induced conditions. I suggest deleting this statement.

Reviewer #5:

Remarks to the Author:

Krüger et al. have made a great effort in addressing the comments and suggestions and they have made extensive revisions to their manuscript. In particular, key concepts are now introduced to a broader audience, and the addition of the native promotor data nicely captures some of the nuances of how effective the CBASS system may be in connection with PspA, in the native *Bacillus cereus* strain.

I still highly recommend that the authors measure the timing of the phage life cycles/

40latent period, as also suggested by another reviewer. This would allow the reader to compare the timing of CBASS activation in this system, compared to the first reports on CBASS.

Additionally, I have the following minor comments:

Title: Swapping the word "regulates" for "activate" or "trigger" would better reflect the conclusions.

The final sentence of the abstract is unclear.

There are still nomenclature issues, e.g. lines 342 and 1082 (*Escherichia coli*).

Line 46: The sentence implies that CRISPR-Cas is exclusively activated by viral DNA/RNA.

This statement should be reworded to reflect that this is one of many mechanisms that activate CRISPR-Cas. Ref. 13 is not appropriate for this statement.

Lines 473-474. This statement is very bold and is not supported by the data since the experiments are not carried out in the native system under non-induced conditions. I suggest deleting this statement.

Final Decision Letter:

Message: 7th March 2024

Dear Malcolm,

I am pleased to accept your Article "Reversible conjugation of a CBASS nucleotide cyclase regulates bacterial immune response to phage infection" for publication in *Nature Microbiology*. Thank you for having chosen to submit your work to us and many congratulations.

Over the next few weeks, your paper will be copyedited to ensure that it conforms to *Nature Microbiology* style. We look particularly carefully at the titles of all papers to ensure that they are relatively brief and understandable.

You may wish to make your media relations office aware of your accepted publication, in case they consider it appropriate to organize some internal or external publicity. Once your paper has been scheduled you will receive an email confirming the publication details. This is normally 3-4 working days in advance of publication. If you need additional notice of the date and time of publication, please let the production team know when you receive the

41proof of your article to ensure there is sufficient time to coordinate. Further information on our embargo policies can be found here:
<https://www.nature.com/authors/policies/embargo.html>

Please note that *Nature Microbiology* is a Transformative Journal (TJ). Authors may publish their research with us through the traditional subscription access route or make their paper immediately open access through payment of an article-processing charge (APC). Authors will not be required to make a final decision about access to their article until it has been accepted. Find out more about Transformative Journals

We welcome the submission of potential cover material (including a short caption of around 40 words) related to your manuscript; suggestions should be sent to Nature Microbiology as electronic files (the image should be 300 dpi at 210 x 297 mm in either TIFF or JPEG format). Please note that such pictures should be selected more for their aesthetic appeal than for their scientific content, and that colour images work better than black and white or

grayscale images. Please do not try to design a cover with the Nature Microbiology logo etc., and please do not submit composites of images related to your work. I am sure you will understand that we cannot make any promise as to whether any of your suggestions might be selected for the cover of the journal.

With kind regards,